# Exploring 400 Gbps/$\lambda$ and beyond with AI-accelerated silicon photonic slow-light technology

Changhao Han[1,2,7], Qipeng Yang[1,7], Jun Qin[3,7], Yan Zhou[4,7], Zhao Zheng[1], Yunhao Zhang[5], Haoren Wang[5], Yu Sun[3], Junde Lu[3], Yimeng Wang[1], Zhangfeng Ge[4], Yichen Wu[1], Lei Wang[5], Zhixue He[5], Shaohua Yu[1,5], Weiwei Hu[1], Chao Peng[1,5,6], Haowen Shu[1,6]✉, John E. Bowers[2]✉ & Xingjun Wang[1,4,5,6]✉

Silicon photonics is a promising platform for the extensive deployment of optical interconnections, with the feasibility of low-cost and large-scale production at the wafer level. However, the intrinsic efficiency-bandwidth trade-off and nonlinear distortions of pure silicon modulators result in the transmission limits, which raises concerns about the prospects of silicon photonics for ultrahigh-speed scenarios. Here, we propose an artificial intelligence (AI)-accelerated silicon photonic slow-light technology to explore 400 Gbps/$\lambda$ and beyond transmission. By utilizing the artificial neural network, we achieve a data capacity of 3.2 Tbps based on an 8-channel wavelength-division-multiplexed silicon slow-light modulator chip with a thermal-insensitive structure, leading to an on-chip data-rate density of 1.6 Tb/s/mm². The demonstration of single-lane 400 Gbps PAM-4 transmission reveals the great potential of standard silicon photonic platforms for next-generation optical interfaces. Our approach increases the transmission rate of silicon photonics significantly and is expected to construct a self-optimizing positive feedback loop with computing centers through AI technology.

With the rapid growth of global data volume in recent years, high-speed optical interconnection is seen as one promising approach to revolutionize high-performance computing centers[1-3]. Among the reported optoelectronic integration technologies, silicon photonics is an important platform with high complementary metal-oxide semiconductor (CMOS) compatibility, which brings the feasibility of low-cost and large-scale production at the wafer level[4-6]. However, as core electro-optical (EO) conversion devices, pure silicon modulators have a limited plasma dispersion effect[7], thereby typically possessing low baud rates[8,9]. Nevertheless, the utilization of high-order formats can

maximize the limited bandwidth resources of pure silicon modulators[10]. For optical interconnection of 1.6 TbE, intensity modulation/direct detection (IM/DD) represented by four-level pulse amplitude modulation (PAM-4) is standardized in IEEE Standard 802.3dj[11]. Compared to the complex coherent scheme, the IM/DD system, where the signal is modulated over the intensity of light, has been chosen as the definite technology route for short-reach optical connections in computing centers because of its ease of implementation[12]. But so far, due to the bandwidth-efficiency trade-off and nonlinear distortions, the growth trend of transmission rates

[1]State Key Laboratory of Photonics and Communications, School of Electronics, Peking University, Beijing, China. [2]Department of Electrical and Computer Engineering, University of California, Santa Barbara, CA, USA. [3]Key Laboratory of Information and Communication Systems, Ministry of Information Industry, Beijing Information Science and Technology University, Beijing, China. [4]Peking University Yangtze Delta Institute of Optoelectronics, Nantong, China. [5]Peng Cheng Laboratory, Shenzhen, China. [6]Frontiers Science Center for Nano-optoelectronics, Peking University, Beijing, China. [7]These authors contributed equally: Changhao Han, Qipeng Yang, Jun Qin, Yan Zhou. ✉e-mail: haowenshu@pku.edu.cn; bowers@ece.ucsb.edu; xjwang@pku.edu.cn

based on silicon modulators has been slow over the years. Although the data rates have continued to increase with the proposal of various solutions[13,14], the reported rates still exist a gap from single-lane 400 Gbps. This transmission bottleneck has raised concerns about the viability of pure silicon modulators for ultrahigh-speed scenarios, especially when compared to heterogeneous integration routes represented by thin film lithium niobate[15–19], plasmonics materials[20,21] and organic polymers[22,23]. For pure silicon modulators, it is challenging to increase the transmission rate significantly, while achieving a high integration density and a wide optical passband simultaneously[13]. Although silicon Mach-Zehnder modulators (Si-MZMs) possess the merits of large operating wavelength range, their excessively long modulation arms reduce integration density[24–32]; while silicon microring modulators (Si-MRMs) achieve a compact footprint, the resulting narrow passbands and low thermal stability require additional feedback mechanisms[33–40]. These issues restrict the deployment of silicon modulators and raise concerns about the future development path of silicon photonics[41]. To overcome the above limitations, silicon slow-light modulators (Si-SLMs) are proposed as a promising approach on silicon-on-insulator (SOI) platforms[42–46]. Recently, the theoretical design and device-level research on Si-SLMs have been conducted, leading to a leap in bandwidth performance[47]. Therefore, the potential of high-bandwidth Si-SLM schemes inspires further exploration of the speed limit of silicon photonics by utilizing high-order formats. Also, developing compact Si-SLM systems to enhance the on-chip data-rate density of silicon platforms can reduce the wafer area budget for a certain data capacity. Accordingly, taking full advantages of Si-SLMs and adopting high-order formats to construct a high-capacity transmission system is needed for 1.6 TbE, and is expected to provide key solutions for next-generation 3.2 TbE optical interfaces.

In practical application scenarios, the adoption of equalization is necessary, especially for high-order modulation[48]. In most transmission works, linear equalizers such as feed-forward equalization (FFE) and decision-feedback equalization (DFE) are usually adopted and effectively mitigate linear distortions in the system[49,50]. However, in the process of compensating nonlinear distortions including nonlinear modulation and chirp characteristics, which is particularly important in pure silicon platforms, great challenges have occurred for above conventional equalization methods. On the other aspect, an increasing number of works utilize artificial intelligence (AI) technology to accelerate the discovery in sciences[51]. In the optoelectronics, silicon photonics has promoted the deployment of optical computing for AI applications[52,53] (SiPh for AI), while AI technology can also be applied to realize the potential of silicon photonics[54,55] (AI for SiPh). Specially, for signal processing applications, artificial neural network (ANN) equalizers have been proposed to construct a complex map with nonlinear boundaries between the input and output spaces[56,57]. Compared to conventional equalizers, ANN models the equalization as muti-level problems to mitigate the nonlinear distortions and can provide effective compensation solutions[58]. More importantly, ANN equalizers are particularly favorable for the optical transmission based on Si-SLMs. Although the bandwidth-efficiency trade-off of pure silicon modulation can be optimized effectively by introducing slow-light resonators, the physical principle of Si-SLMs is still the nonlinear plasma dispersion effect, and the nonlinear cosine transfer functions of the device architecture will also introduce signal distortions. Therefore, by using an ANN equalizer, the nonlinear distortions of Si-SLMs can be reduced. At the manufacturing level, achieving complete uniformity remains challenging in the doping process, while the depletion-mode PN junction is the modulation basis for Si-SLMs[9,13]. Simultaneously, although one-dimensional waveguide grating structure has improved the process stability compared to two-dimensional photonic crystals[42,45], the structural fluctuations of doped silicon gratings still exist due to the fabrication limitations. Therefore, Si-SLMs require an adaptive technical approach at the back end to eliminate the

variations caused by process deviations, thus making the deployment of large-scale high-density Si-SLM systems possible.

In this work, we propose an AI-accelerated silicon photonic slow-light technology to explore 400 Gbps per wavelength transmission. Using a standard silicon photonic process, we fabricate an 8-channel wavelength-division-multiplexed (WDM) Si-SLM chip on a SOI platform. Benefiting from the innovative slow-light design, the intrinsic bandwidth-efficiency trade-off for pure silicon modulators is effectively mitigated. Under the precise optimization, our compact Si-SLMs exhibit an ultrahigh EO bandwidth of 90 GHz, a remarkable modulation efficiency of 0.82 V·cm while possessing a wide optical passband of 7 nm around 1550 nm simultaneously. By utilizing the ANN equalizer, we demonstrate a total data capacity of 3.2 Tbps based on our thermal-insensitive Si-SLM chip, with all bit error rates (BERs) below hard-decision forward error correction (HD-FEC) threshold, leading to an on-chip data-rate density of 1.6 Tb/s/mm². Meanwhile, the transmission link does not require individual resonant wavelength adjustment and additional thermo-electric cooler (TEC) platforms, thus reducing the system budget. Notably, adopting the fundamental industry-standard IM/DD format PAM-4, we realize an unprecedented 400 Gbps optical transmission per wavelength successfully, which is, to the best of our knowledge, the highest single-lane transmission rate achieved in standard silicon photonic platforms. Our work reveals the great potential of standard silicon photonic platforms in 3.2 TbE optical interconnections, validating the significant value of AI technology for silicon photonics.

## Results
### Device and chip design
The overall application architecture of the AI-accelerated slow-light technology is conceptually demonstrated in Fig. 1. Here, the AI-accelerated slow-light transceiver module provides an ultrahigh-speed solution for interconnections in computing centers (applications in photonics include neural networks, inverse design, dynamic simulation, etc.). Through collaborative integration with ANN, the Si-SLM system can achieve a leap in throughput per lane and elevates the interface rate of computing centers, thereby leading to faster training iterations for weight programming of ANN and enabling more immediate and efficient signal processing for the Si-SLM system. In turn, the enhanced transceiver module can further increase the interconnection rate for computing centers to achieve more powerful computility. This bi-directional promotion mechanism forms a self-optimizing positive feedback loop between AI-accelerated Si-SLM systems and computing centers, continuously improving the overall architecture performance.

Based on the above design concept, we designed an ultrahigh-speed WDM Si-SLM chip for computing centers. The slow-light effect is an effective approach to enhance the interaction between light and matter, which has been confirmed in both theory and experiment[59–64]. Especially, thanks to the CMOS compatibility of the silicon slow-light structure, Si-SLMs for ultrahigh-speed transmission can be fabricated under a standard silicon photonic process, without introducing complex manufacturing process and heterogeneous materials, and an 8-channel Si-SLM chip was realized on a 200-mm SOI wafer (Fig. 2a). Benefiting from the compact footprint and dense distribution of Si-SLMs, the actual modulation area (bottom) occupies only 4 mm × 0.5 mm, with the pitch of 500 μm between neighboring devices. Also, corresponding to the transmitter area, the 8-channel GeSi photodetectors (PDs) are also designed and arranged (top) for the complete transceiver. It can be seen that the adoption of Si-SLMs makes the transmitter area already close to the receiver area. For the optical ports, the edge coupler array (left) is used on the same side, and the three groups from bottom to top are modulator inputs, modulator outputs and PD inputs, respectively. Meanwhile, the direct-current (DC) pad array (right) is applied to control the modulator operating

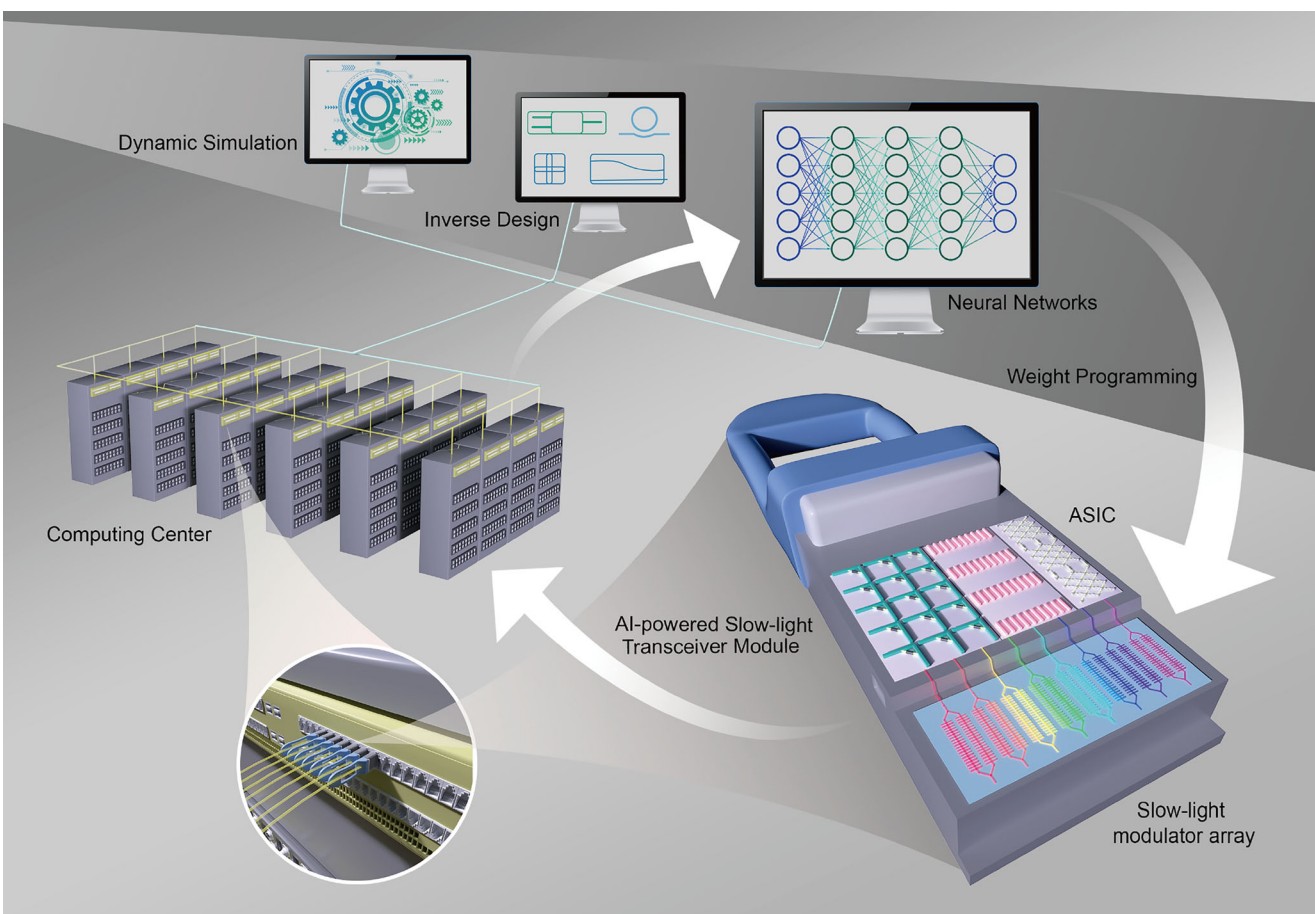

**Fig. 1 | Artificial intelligence (AI)-accelerated silicon photonic slow-light technology.** Conceptual drawing of the application architecture of the AI-accelerated slow-light technology. The AI-accelerated slow-light transceiver module provides an ultrahigh-speed solution to increase interconnection rates in computing centers (applications in photonics include neural networks, inverse design, dynamic simulation, etc.). The enhanced computility can lead to faster training iterations for weight programming of artificial neural network (ANN), thus promoting the signal processing for silicon slow-light modulator (Si-SLM) systems to be more immediate. In turn, the enhanced transceiver module can further increase the interface rates of computing centers. Therefore, based on this bi-directional promotion mechanism, a self-optimizing positive feedback loop can be constructed to continuously improve the overall architecture performance.

point effectively through the TiN heater of each device. For the single device, the architecture and morphology of one Si-SLM can be seen in Fig. 2b. Under a compact Mach-Zehnder interferometer (MZI) structure, the modulation arms can be shrunk to only 249 μm due to the slow-light effect, which is an order of magnitude shorter than conventional Si-MZMs, thus enhancing system integration density. Simultaneously, to enlarge the phase accumulation, also for the better photoelectric integration and low chirp, a dual-drive architecture of GSGSG-type radio-frequency (RF) electrodes is adopted here, which enables the modulator to operate under the push-pull driving mode. At the remote end of the RF electrodes, on-chip termination resistors are integrated to reduce microwave reflection. Near the waveguide output port, TiN heaters are adopted on the waveguides to control the operating point of the modulator (See Supplementary Section 1 for more details). In the modulation arms, the slow-light effect is generated by the coupled-resonator optical waveguide (CROW), which is a one-dimensional waveguide grating structure. The scanning electron microscope (SEM) image of the fabricated slow-light waveguide is also illustrated below the device photograph in Fig. 2b. Here, a complete resonator is constructed by a $\lambda/4$ phase shifter region with broader width in the middle position and an equal number of Bragg gratings on both sides with a period of around 300 nm, along the direction of light propagation in the waveguide. Each certain number of gratings construct one side beam of a resonator, and the two beams are separated by the phase shifter from their adjacent one. Therefore, the supercell is

created through the phase shifter region, which leads to a mid-gap mode embedded in the bandgap. In our reconfigurable slow-light model, a finite number of resonators are cascaded to construct the modulation arm, and the structure parameters can be selected flexibly, according to specific application scenarios. Based on the grating waveguide configuration, the PN junction with a periodic structure is formed, which will work in the depletion mode under reverse bias voltage. Actually, the slow-light approach is a pure silicon solution with considerable high design freedom. For example, if the design goal is to push the bandwidth to the extreme limit, so as to achieve ultra-low-cost OOK transmission, then fewer resonators are required[47]. Here, for achieving a deeper modulation depth and better separation of ultrahigh-speed multi-level signals, we redesigned structures fully, cascading more resonators to enlarge the phase accumulation, and focus on the design of more resonators to achieve a balance between bandwidth and efficiency (Supplementary Section 2).

## Performance characterization

To prove the feasibility of the proposed Si-SLMs for high-order signal transmission, we comprehensively characterized the device performance in detail, including both dynamic and static parameters. First, we evaluated the high-frequency EO response of the designed device, which is a prerequisite for achieving ultrahigh-speed transmission. Also, for Si-SLMs, an ultrahigh EO bandwidth is the most prominent advantage, dictating the achievable data rate. Especially, considering

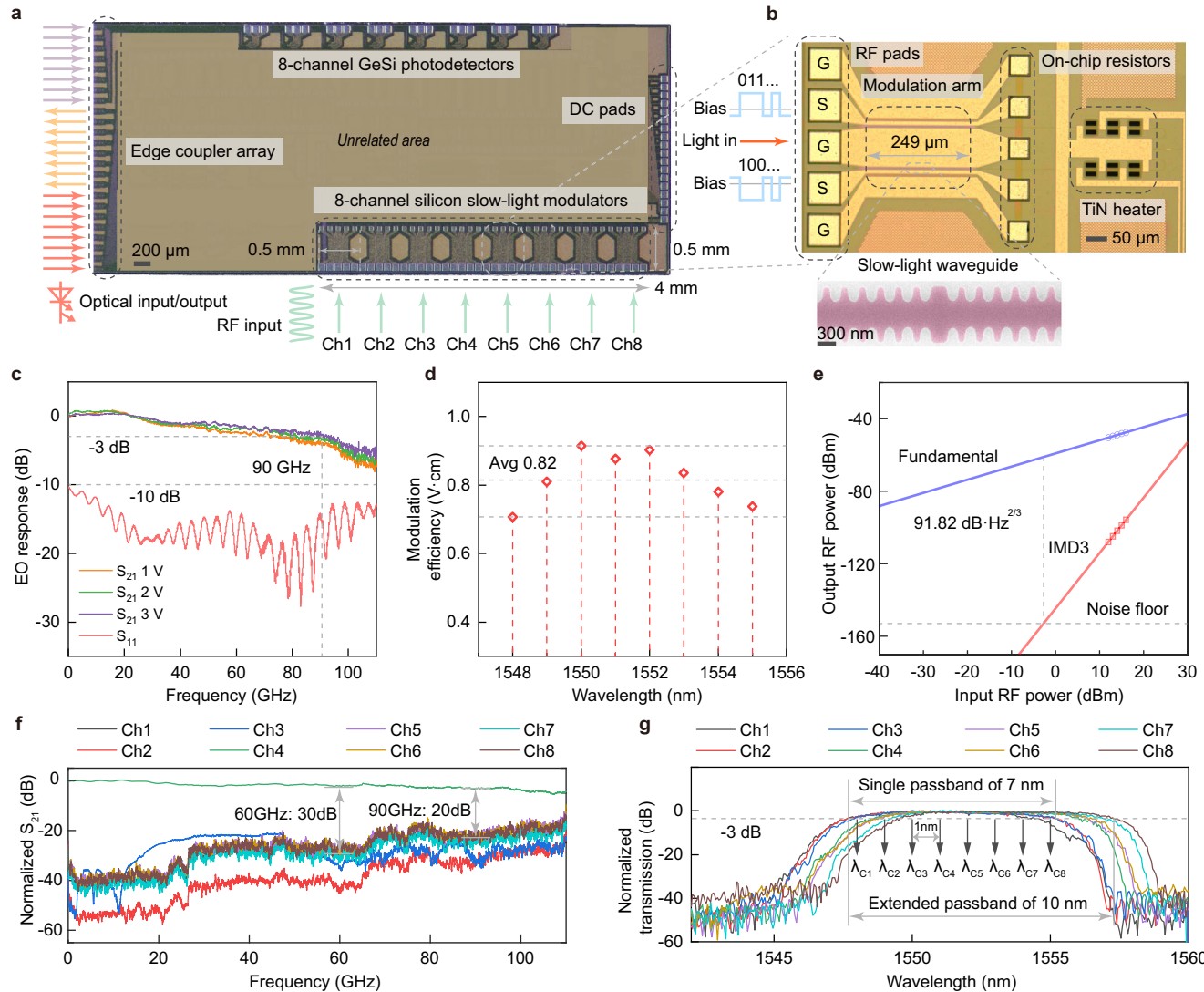

**Fig. 2 | Design and characterization of the Si-SLM chip. a** Optical micrograph of the Si-SLM chip. An 8-channel Si-SLM array was fabricated on a 200-mm silicon-on-insulator (SOI) wafer, and the actual modulation area occupies only 4 mm × 0.5 mm, with the pitch of 500 μm between neighboring devices. **b** Device morphology of the fabricated Si-SLM. The modulator is under a Mach-Zehnder interferometer (MZI) structure, with the compact modulation arm of only 249 μm. A dual-drive architecture of GSGSG-type radio frequency (RF) electrodes is adopted, which enables the modulator to operate under the push-pull driving mode to enlarge the phase change. The scanning electron microscope (SEM) image of the fabricated slow-light waveguide is shown as the inset below. The waveguides are fabricated under a standard silicon photonic process and all feature sizes are suitable for the commercial silicon photonic foundry. **c** Measured electro-optic (EO) response under different bias voltages. The $S_{21}$ response demonstrates that the modulator has an EO bandwidth of 90 GHz under the bias voltage of 3 V. The $S_{11}$

response is maintained below −10 dB, indicating that the reflection of the modulator is weak. **d** Modulation efficiency under different wavelengths. An averaged enhanced modulation efficiency of 0.82 V·cm is obtained due to the introduction of slow-light structure. **e** Linearity of the Si-SLM. The measured third-order intermodulation spurious free dynamic range (IMD3 SFDR) is 91.82 dB · Hz^{2/3}, mainly due to the nonlinear limitation of the plasma dispersion effect in silicon materials. **f** Crosstalk of parallel Si-SLMs. The measured crosstalk is around 30 dB at 60 GHz and 20 dB at 90 GHz. Due to the ultra-short RF links, the crosstalk of the Si-SLM array is at a low level and the impact on parallel signal transmission is small. **g** Passbands of the 8-channel Si-SLM array. The modulator possesses a flat passband of 7 nm around 1550 nm, and the passbands are basically consistent between different devices. For wavelength-division-multiplexed (WDM) system, the specific wavelengths of 8-channels are from 1548 nm to 1555 nm, with the spacing of 1 nm.

that the ANN equalizer adopted later will improve signal quality, the first thing to ensure is that the signal can complete the entire EO conversion and transmission process with minimal RF loss. Here, to meet the demand of high-order modulation, we control the structure parameters to maximize the EO bandwidth while ensuring a balance between bandwidth and efficiency. We experimentally characterized the small-signal properties of the Si-SLMs through S-parameter measurements to obtain the EO bandwidth, including $S_{21}$ transmission and $S_{11}$ reflection responses. Through the careful regulation strategy, the modulator has an EO bandwidth of 90 GHz under the bias voltage of 3 V, as shown in Fig. 2c, which is a fairly high bandwidth value for pure

silicon modulators. While the device bandwidth increases with the bias voltage, it already exceeds 70 GHz at only 1 V, and the low bias voltage requirement facilitates the co-packing with integrated driver chips. In terms of reflection performance, the $S_{11}$ response is maintained below −10 dB, indicating that the reflection of the modulator is weak. Simultaneously, with the adoption of slow-light effect, the modulation efficiency can be improved by enhancing the interaction between light and matter. Here, we apply a low-frequency small signal to the device around the quadrature bias point and obtain the modulation efficiency by fitting the phase variation at multiple wavelengths. Figure 2d demonstrates the modulation efficiency measured under different

wavelengths. The modulation efficiency increases slightly closer to the edge of the passband, mainly due to the relatively higher group index at the band edge region. In the passband around 1550 nm, an averaged modulation efficiency of 0.82 V·cm is obtained, which is also a remarkable value for pure silicon modulation based on plasma dispersion effect. Therefore, by adopting the slow-light approach, we achieved improvements in both bandwidth and modulation efficiency for pure silicon modulators simultaneously. As an approach that can be utilized on SOI platforms, the slow-light structure can mitigate the intrinsic bandwidth-efficiency trade-off in silicon material, achieving the promotions in both EO bandwidth and modulation efficiency. For the optical loss of the modulator, the unoptimized coupling loss between a pair of edge couplers and the fibers is 10 dB (5-dB loss for each one), and the insertion loss of the Si-SLMs is measured to be 10.5 dB (9.1 dB from the modulation arms, remaining from directional couplers and routing waveguides). The optical loss can be further reduced by improving the manufacturing process or introducing a transition structure between the conventional waveguide and the slow-light waveguide. Moreover, by embedding CROW into the modulation arms, the advantages of MZI can be leveraged to maintain a sufficient static extinction ratio (ER) while reducing the device footprint. Through the standard thermal tuning mechanism by TiN heaters, the static ER of the compact modulator is measured to be 36 dB, provided by the resonator-assisted MZI architecture. Next, to characterize the linearity for measuring the multi-level separation capabilities in high-order transmission, we further examined the third order intermodulation (IMD3) spurious free dynamic range (SFDR). For our Si-SLMs, the measured IMD3 SFDR is 91.82 dB·Hz$^{2/3}$, shown in Fig. 2e, which is similar to conventional silicon modulators, mainly due to the nonlinear limitation of the plasma dispersion effect of silicon materials, together with the nonlinear cosine transfer functions of the MZI architecture. As a pure silicon device, the physical essence of Si-SLMs is still based on the nonlinear silicon modulation, and the mitigation for subsequent nonlinear distortions is a challenge especially for ultrahigh-speed transmission, which is exactly what the ANN model excels at.

Afterwards, we experimentally characterized the performance of the modulator array on the Si-SLM chip. For WDM modulators, the channel crosstalk is a significant performance indicator to evaluate the overall transmission quality. Also, in terms of the designed layout of the modulator chip, to increase the rate density, a dense arrangement of devices is necessary. Here, the pitch of the modulator array is 500 μm (Fig. 2a), resulting in a minimum distance of only 20 μm between modulators. Nevertheless, Si-SLMs are favorable for dense array designs, because the ultra-compact footprint of Si-SLMs can shorten RF links (approximately ten times that of Si-MZMs), which will reduce the crosstalk between parallel signals at an ultrahigh-frequency region. To quantify the crosstalk, we tested the EO response of parallel modulators by applying a small RF signal on one channel (Ch4) and simultaneously receiving the transmitted signal on the other channels (Ch1, Ch2, Ch3, Ch5, Ch6, Ch7 and Ch8). Figure 2f demonstrates the experimental result of crosstalk, in which the curve of Ch4 is its own anticipated EO bandwidth curve, and the response curves of other channels illustrates the measured crosstalk is around −30 dB at 60 GHz and −20 dB at 90 GHz. In the ultra-compact and ultrahigh-density chip layout, the crosstalk of the WDM Si-SLMs remains at a low level and the impact on parallel signal transmission is small due to the ultra-short RF links. Furthermore, we tested the spectral performance for all channels on the same WDM Si-SLM chip. In practice, the optical passband of the modulator is the reflection of the optical bandwidth, and a wide passband is essential for multi-wavelength applications with high thermal robustness. Theoretically, for the photonic bandgap of the designed structure, the topologically mid-gap mode is generated in the bandgap between the antisymmetric and symmetric transverse electric bands[65–68], as the mode adopted for practical communication

application. Thus, the supercell band opens multiple windows including one large passband around 1550 nm. By cascading more resonant cavities, the passband can be kept wide enough while ensuring sufficient efficiency. Here, under careful regulation, the fabricated Si-SLM possesses a flat passband (mid-gap mode) of 7 nm around 1550 nm, with an out-of-band rejection ratio of 50 dB, shown in Fig. 2g. Meanwhile, the high passband conformity between devices can demonstrate that the fabricated Si-SLMs have a relatively favorable process consistency in general. However, from the spectral results, it can be seen that a wavelength shift still occurs, due to the unavoidable fabrication deviations. Although the manufacturing uniformity of Si-SLMs has been improved by introducing one-dimensional CROW compared to two-dimensional photonic crystals, the performance fluctuations between doped silicon grating structures still exist. Therefore, for large-scale Si-SLM systems, the AI equalization at the back end is necessary to minimize device performance fluctuations caused by process errors. Despite this, on the other aspect, the appropriate slight wavelength shift can still extend the available passband to 10 nm on one chip and the passband resources have not been fully utilized yet. Here, for our designed WDM Si-SLM system, the specific wavelengths of 8-channels are from 1548 nm to 1555 nm, with the spacing of 1 nm (Fig. 2g). By increasing the wavelength operating range (e.g. 10 nm), a larger wavelength interval (e.g. 1.25 nm) can be adopted between channels. It is worth noting that, thanks to the wide passband and high uniformity of Si-SLMs, compared with Si-MRMs, there is no need to design separately to obtain individual operating wavelengths for different channels, nor to find the precise wavelength operating point between different resonance periods. Moreover, the compact Si-SLMs possess high thermal robustness, which can reduce the requirements for additional TEC operating platforms, thus saving the system budget.

In short, optimizing the bandwidth-efficiency trade-off while maintaining a sufficiently wide passband through Si-SLMs is the device-level guarantee for achieving ultrahigh-speed transmission per lane. Based on the designed Si-SLMs, the compact WDM chip with low crosstalk and flat passbands can realize ultra-high integration density and lead to a remarkable total capacity.

### DNN transmission

For the system level, based on our carefully designed high-bandwidth Si-SLMs, we build an 8-channel WDM Si-SLM transmission system, which is demonstrated in Fig. 3a. On each channel, the Si-SLM encodes the carrier into PAM-4 signal format at different symbol rates. At the receiving side, the signal can be partly coupled to an on-chip GeSi PD on the same chip, whereas the remaining part is sent into a commercial PD. The transmitted ultrahigh-speed signals are recorded by a real-time oscilloscope and then processed with AI equalizers. For the algorithm level, leveraging AI technology to improve the transmission capacity, we adopt the ANN equalizers to improve the nonlinear limitation of conventional equalizers such as FFE and DFE. Although the bandwidth-efficiency trade-off has been mitigated, the Si-SLMs will still introduce nonlinear distortions during signal transmission due to the nonlinear modulation of the pure silicon material and the nonlinear cosine transfer functions of MZI architecture. At the system level, the primary electrical nonlinear sources originate from RF components due to the inherent nonlinear behavior of charge transport at the transistor junction, which is predominantly observed in the transistor driver (DRV) and transimpedance amplifier (TIA). Also, the utilization of erbium-doped fiber amplifiers (EDFA) could introduce nonlinear Kerr impairments to the optical link. Therefore, the resulting nonlinear signal distortions become the bottleneck for improving the signal quality, especially for the multi-level formats at ultra-high speeds. The ANN equalizers, which specialize in compensating nonlinear distortions, are exactly suitable for application in the Si-SLM system.

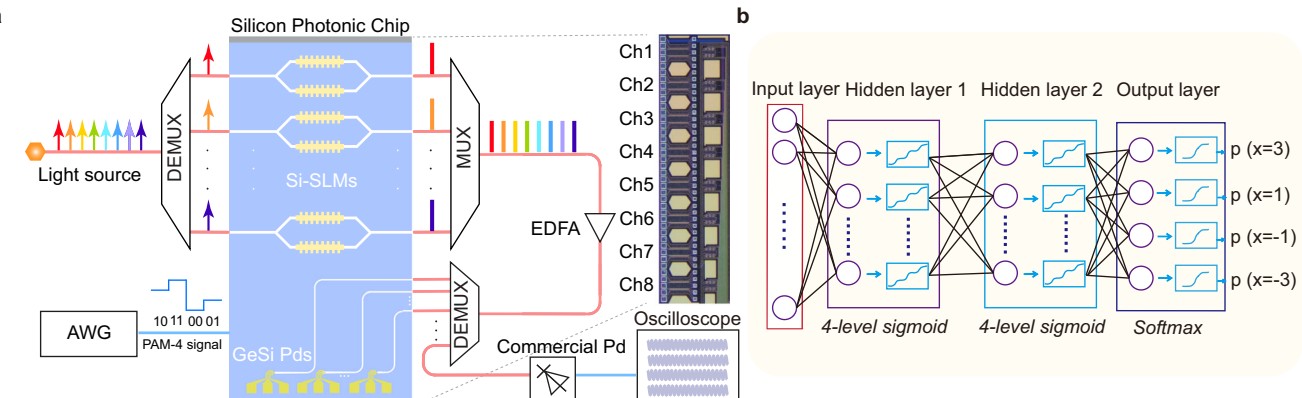

**Fig. 3 | Deep-learning neural network(DNN) transmission system. a** Schematic of the WDM Si-SLM transmission system. On each channel, the Si-SLM encodes the carrier into four-level pulse amplitude modulation (PAM-4) signal format at different data rates. The transmitted signals are first sampled by a real-time oscilloscope and afterward processed with AI equalizers offline. **b** The principal structure of DNN equalizer. In the experiment, the input layer is set with 25 neurons and two hidden layers with 30 neurons are adopted.

To minimize computing resources, we develop a relatively simple ANN model first, extend the network to a deep-learning neural network (DNN) equalizer with only two hidden layers to enhance the system performance. In the DNN, to classify the signal into four categories for PAM-4 signal[58], an activation function with four saturation level regions is implemented through equation $f(x) = 2\eta_2/(1+e^{-\eta_1(x-2\alpha)}) - \eta_2 + 2\alpha$, where $\alpha$ equals to −1, 0, 1 when x ≤ −1, −1 < x ≤1 and x > 1, respectively, and $\eta_2 = (1+e^{-\eta_1})/(1-e^{-\eta_1})$. The constructed $f(x)$ has four saturation regions which are close to the amplitude of PAM-4 (−3, −1, 1, 3) which makes it suitable for PAM-4 equalization. Similarly, it can be extended to 8-level *sigmoid* function which can be used to equalize the modulated signals such as PAM-8. Details can be found in Supplementary Section 3. Two hidden layers are incorporated in the DNN. Figure 3b shows the tap-delay two hidden layer DNN equalizer structure. The activation function in each neurons determines the nonlinear mapping characteristic across the network. The weight optimization process involves a gradient descent scheme coupled with the error back-propagation algorithm to iteratively update the network parameters.

Since 224 Gbps PAM-4 is the standard rate for 1.6 T interfaces, we focus on the 224 Gbps PAM-4 signal transmission here. Thanks to the ultrahigh-bandwidth and wide flat passband of Si-SLMs, we obtained clear eye diagrams of 224 Gbps PAM-4 together with 200 Gbps PAM-4 at all channels of different wavelengths (1548–1555 nm, with 1 nm spacing, Ch2, Ch4, Ch6 and Ch8 with corresponded wavelengths are demonstrated here), shown in Fig. 4a, with all BERs lower than $2 \times 10^{-2}$ (transmission results for all channels are shown in Supplementary Section 3), leading to a total capacity of 1.6 Tbps. Specifically, Fig. 4b demonstrates the BERs at different wavelengths in the passband for 8 channels, with the PAM-4 signal speed set as 130 Gbps, 170 Gbps, 200 Gbps and 224 Gbps, and the consistency between all WDM channels is favorable. Moreover, the BER curves with increasing data rates of different channels (Ch2, Ch4, Ch6 and Ch8 are shown for example here) are illustrated in Fig. 4c, while the almost overlapped lines confirm the uniformity between different channels.

Furthermore, we verify that the implemented DNN equalizer captures channel response instead of the generation pattern of the pseudo random bit sequence (PRBS). Different patterns of PRBS are generated, Pattern 0 is used to train the DNN, the others (Pattern 1, 2 and 3) are used for BER test. The results of 170 Gbps, 200 Gbps and 224 Gbps are shown in Fig. 4d, there is a similar performance under different signal patterns, which indicates that the DNN equalizer learns the information of channel rather than the signal pattern and has a robust equalization ability for different data patterns. For the neuron numbers, the BER performance of the DNN initially for all data rates improves when the number of input neurons increases, as shown in

Fig. 4e. However, after reaching a certain neuron count, such as 25 neurons, the BER performance ceases to improve further. Meanwhile, as the number of neurons in the hidden layer increases, there is no significant (exponential order) variation in the BER performance, as depicted in Fig. 4f. In the experiment, the input layer is set with 25 neurons and two hidden layers are set with 30 neurons.

As a relatively simple AI equalizer, our designed DNN equalizer with only two hidden layers takes up less computing resources, but still achieves high-quality transmission for 1.6 TbE interface with 224 Gbps per lane. As the simplest high-order modulation format, PAM-4 is highly feasible for data center scenarios. Therefore, it is crucial to realize optical transmission of 224 Gbps PAM-4 per channel based on a Si-SLM chip. Furthermore, based on 224 Gbps PAM-4 transmission per lane, a 3.2 TbE interface can be realized by scaling out the channels. More importantly, the high-quality PAM-4 results of DNN equalizer which takes less resources demonstrate the potential of ANN solutions and its adaptability to slow-light systems, which encourages us to continue developing more efficient ANN equalizers.

### GRU transmission

By adopting the designed simple DNN equalizer with less computing resources, we achieve a total capacity of 1.6 Tbps transmission with 224 Gbps PAM-4 per lane based on our WDM Si-SLM chip. This remarkable result demonstrates the high compatibility between the AI-accelerated technology and Si-SLM systems. Therefore, for the purpose of exploring the potential of AI-accelerated slow-light systems, we develop a more efficient ANN equalizer to push the silicon photonics transmission to the next stage.

Recurrent neural networks (RNN) are widely utilized for their proficiency in modeling temporal dynamics and processing sequential data. In principle, bi-directional RNNs (bi-RNNs) can efficiently handle not only inter-symbol interference among preceding and succeeding symbols caused by chromatic dispersion, but also the nonlinear impairments caused by devices and fiber transmission links[69-71]. Additionally, compared to unidirectional RNN, bi-RNNs model the dependence on past and future states. A gated recurrent unit (GRU) contains two gates including reset gate and update gate[70,72]. GRU is a less complex variant compared to the long short-term memory (LSTM)[70], which is an advanced type of RNN that demonstrates robust capabilities for capturing and modeling long-term dependencies[57]. The detailed structure of a GRU unit is demonstrated in Fig. 5a. The bidirectional GRU (bi-GRU) model comprises two unidirectional GRU layer operating in opposite directions. By combining forward and backward GRU processing, the model incorporates information from both the future and the past to influence its current states. Figure 5b shows the

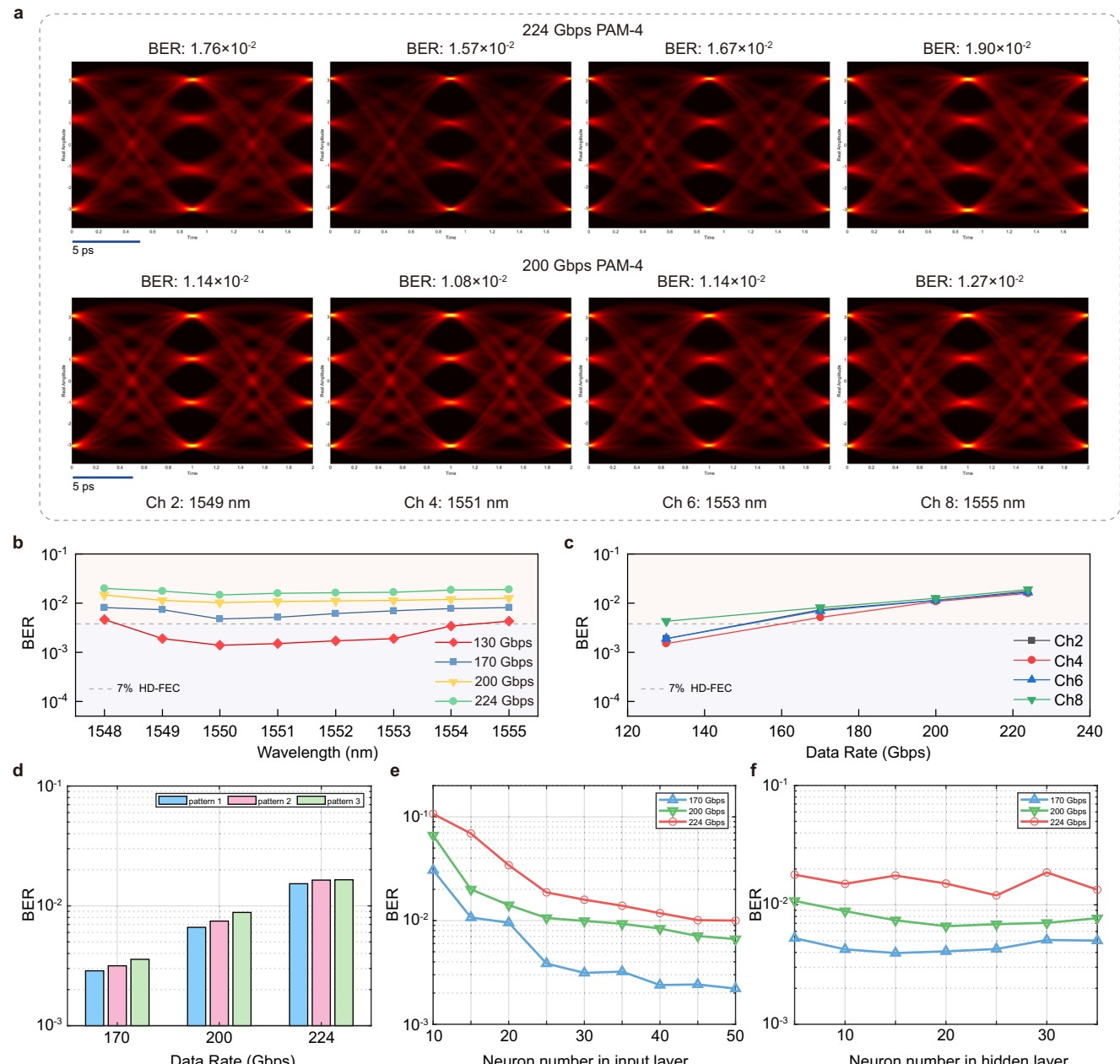

**Fig. 4 | DNN transmission results. a** 224 Gbps and 200 Gbps PAM-4 eye diagrams of different channels. Ch2, Ch4, Ch6 and Ch8 are shown here. **b** Bit error rate (BER) curves of different data rates under different wavelengths for 8 channels. The PAM-4 signal speeds are set as 130 Gbps, 170 Gbps, 200 Gbps and 224 Gbps. The consistency of different wavelengths between channels is favorable. **c** BER curves with the increasing data rates. Ch2, Ch4, Ch6 and Ch8 are shown here. The almost overlapped lines confirm the uniformity between different channels. **d** BERs of different patterns under different data rates. There is a similar performance under different signal pattern, which indicates that the implemented DNN equalizer has a robust equalization ability for different data patterns and captures channel response instead of the generation pattern of the pseudo random bit sequence (PRBS). **e** BER curves of different neuron numbers in the input layer. The BER performance improves when the number of input neurons increases. After reaching a certain neuron count, the BER performance ceases to improve further. **f** BER curves of different neuron numbers in the hidden layer. No significant variation of BERs is observed as the number of neurons in the hidden layer increases.

architecture of the implemented bi-GRU network for nonlinear equalization in our Si-SLM transmission system. The first layer is the input layer, where the current symbol $x_i$ is enclosed with its $k$ preceding and $k$ succeeding symbols. This sequence serves as the input to the bi-GRU network. The subsequent layer is the GRU layer, which consists of two GRU links. The input sequence is first passed through the initial GRU layer. Subsequently, the sequence is reversed and processed through the second GRU layer. This approach allows us to process both preceding and subsequent data simultaneously, enabling information from both the past and future to influence the current states. The outputs of the bi-GRU model layer are fully connected to a

linear layer. As a result, the output layer produces the predicted class for the current symbol. Consequently, the predicted class $y_i$ for the i-th symbol $x_i$ is determined. Some reports employ bi-GRU to compensate the fiber nonlinear impairment during the long-distance transmission in coherent system[70,73], but there are still less bi-GRU performance evaluation reports in short-reach IM/DD systems, especially for systems that employ silicon devices with data rates around 400 Gbps in a high-density silicon chip. In this work, we employ the bi-GRU network to promote the transmission performance. Meanwhile, based on bi-GRU, a new three-layer GRU equalizer (T-biGRU) is further proposed and implemented. The T-biGRU model is determined based on the

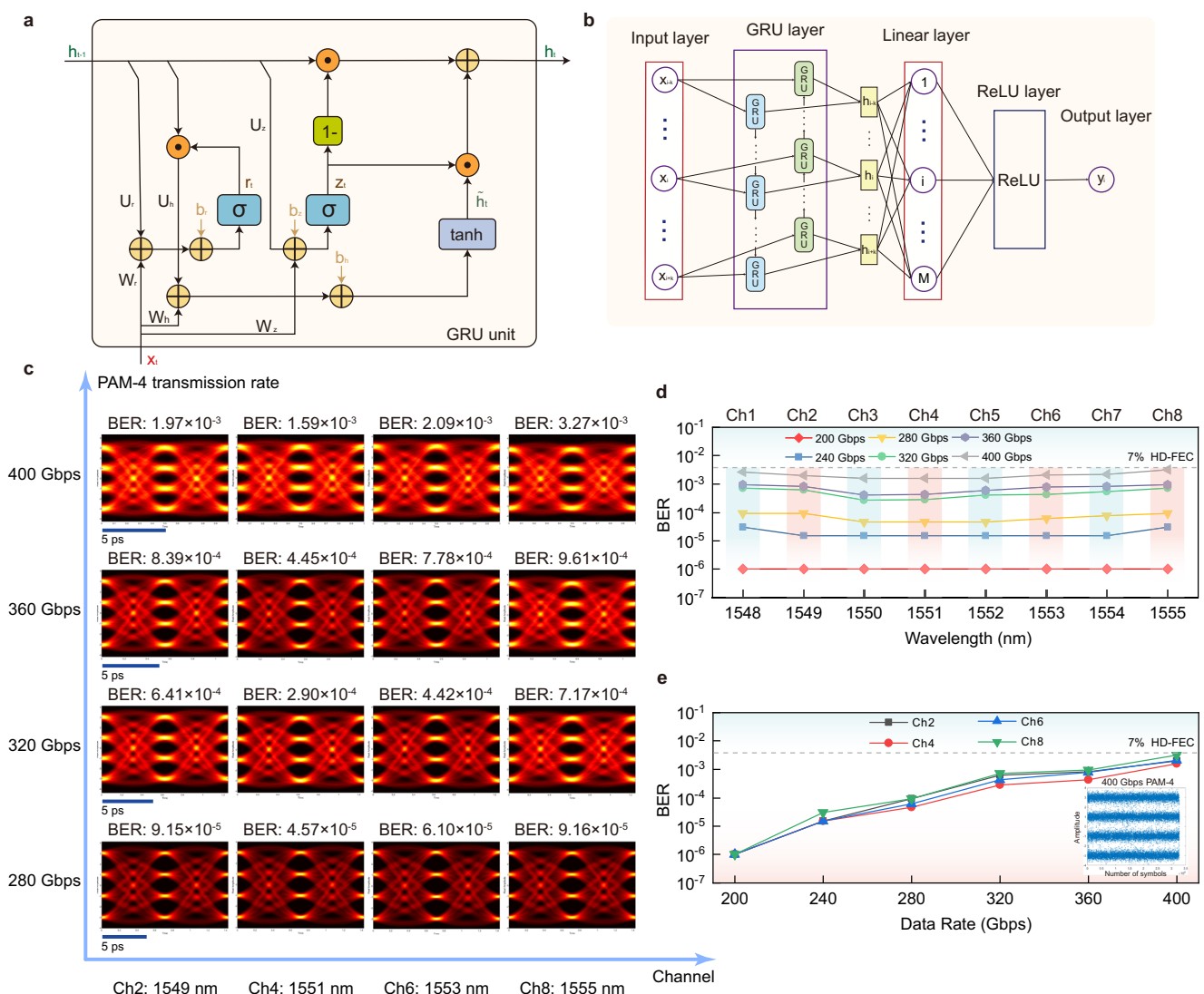

**Fig. 5 | Bidirectional gated recurrent unit (bi-GRU) transmission results of PAM-4 signal. a** Scheme of one gated recurrent unit (GRU) unit. The detailed structure of a GRU unit includes a reset gate and an update gate. **b** The architecture of bi-GRU equalizer. The input layer, the GRU layer consisting of two GRU links, the linear layer, the ReLU layer and the output layer are included in the implemented bi-GRU network. **c** bi-GRU PAM-4 eye diagrams of different data rates. 200 Gbps, 240 Gbps, 280 Gbps, 320 Gbps, 360 Gbps and 400 Gbps for Ch2, Ch4, Ch6 and Ch8 are shown here, with all BERs under hard-decision forward error correction (HD-FEC)

threshold. **d** BER curves of all 8-channel PAM-4 signals with respect to different data rates. The passband consistency remains favorable and a total data capacity of 3.2 Tbps is achieved based on the Si-SLM chip. **e** BER curves of PAM-4 signal with increasing data rates of different channels. Ch2, Ch4, Ch6 and Ch8 are shown here. The variation trend of BER curves is relatively smooth, and no point where the BER abruptly raises is observed. The inset is the constellation of 400 Gbps PAM-4 transmission.

state of three GRU layers. The first GRU layer processes the data in a forward direction, starting from the beginning of the sequence. The second GRU layer processes data in a backward direction to capture reverse temporal dependencies. The third GRU layer, like the first, processes data in a forward direction from the beginning of the sequence (see Methods). The hidden state encapsulates the flow of symbolic information across recurrent time steps, ensuring continuity and context throughout the sequence. The bi-GRU model relies on the states of two GRU layers, whereas the T-biGRU model utilizes the states of three GRU layers. By integrating forward, backward and repeated forward GRU processing, the T-biGRU model more comprehensively extracts both global and local features of the sequence, thereby further enhancing equalization performance.

Next, we evaluated the practical PAM-4 transmission for the Si-SLM chip by employing bi-GRU and T-biGRU, as depicted in Figs. 5–9. Here, our goal is to increase the single-lane data rate of Si-SLMs significantly by AI approach. However, the frequency broadening brought

by IM/DD signals cannot be ignored at ultrahigh-speed data rates. Therefore, in the practical GRU WDM transmission experiment, we select on-chip channels Ch1, Ch3, Ch5, Ch7 (odd channels) as one WDM transmission path and Ch2, Ch4, Ch6, Ch8 (even channels) as the other WDM transmission path to reduce modulation crosstalk by increasing the spectral separation between channels. After assembling each group of on-chip signals through WDM, the two paths are then transmitted in parallel to reach a total capacity of 3.2 Tbps, similar to the solution adopted by commercial optical module manufacturers[74]. By adopting bi-GRU equalization, we can realize high-quality eye diagrams of PAM-4 signal up to 400 Gbps, indicating 3.2 Tbps aggregation data rate of 8 parallel channels. Figure 5c summarizes the optical eye diagrams of PAM-4 signals at gradually increasing rates for different channels, in which the horizontal axis shows the different channels (Ch2, Ch4, Ch6 and Ch8 are shown for examples here) in the WDM system together with their corresponding wavelengths, and the vertical axis shows the different PAM-4 transmission rates (280 Gbps,

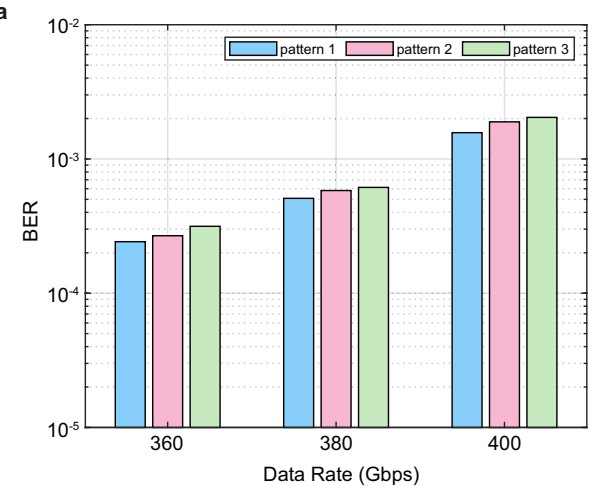
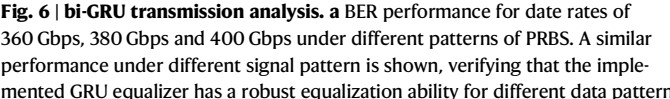
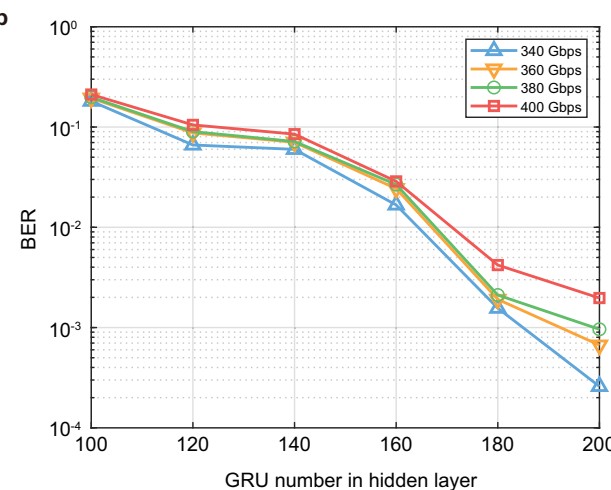

**Fig. 6 | bi-GRU transmission analysis. a** BER performance for date rates of 360 Gbps, 380 Gbps and 400 Gbps under different patterns of PRBS. A similar performance under different signal pattern is shown, verifying that the implemented GRU equalizer has a robust equalization ability for different data patterns and captures the channel response rather than PRBS pattern. **b** BER curves of different GRU numbers in the hidden layer from 340 Gbps to 400 Gbps. With more GRU units in the hidden layer, the BERs will be better. In the experiment, 200 GRU units is chosen for the equalizer.

320 Gbps, 360 Gbps and 400 Gbps are demonstrated). All the eye diagrams are quite clear and the BERs are all below HD-FEC threshold, even up to 400 Gbps. Actually, taking 280 Gbps PAM-4 as an instance, which is already a fairly high rate for pure silicon modulators, our solution can reduce the BERs to only around $10^{-5}$ order at this rate. More importantly, even when the rate increases to 400 Gbps, the BERs still maintain below HD-FEC threshold, and the performance consistency between channels does not show downward trend. To the best of our knowledge, this is the highest single-lane IM/DD transmission rate for silicon modulators. In particular, we achieve this goal only using the industry-standard format PAM-4 for computing centers. Simultaneously, the wide optical passbands of Si-SLMs provide the prerequisite for multi-wavelength communication. The BERs at different wavelengths of all channels are well below the HD-FEC threshold for the whole passband, as demonstrated in Fig. 5d, and the passband consistency is favorable, which is critical for multi-wavelength applications. By leveraging single-lane 400 Gbps PAM-4 transmission, a total data capacity of 3.2 Tbps is achieved based on the 8-channel Si-SLM chip with a compact modulation area of only 4 mm × 0.5 mm, leading to a remarkable on-chip data-rate density of 1.6 Tb/s/mm². Based on the improved data-rate density, more modulator arrays can be integrated on a single wafer with the same tape-out budget, thereby significantly increasing the total transmission capacity for one chip, or reducing the average cost for a certain targeted capacity. For the trend of BERs with increasing rates, the BER curves for different channels are demonstrated in Fig. 5e, and the nearly overlapped curves illustrate the performance uniformity between different channels. Especially, the variation trend of BER curves is relatively smooth, and no point where the BER abruptly raises is observed with the increasing rates. Also, the PAM-4 constellation at 400 Gbps is illustrated as the inset, high-quality differentiation between the four levels can be observed. For complete results, all the PAM-4 eye diagrams, constellations and corresponded BERs for all channels and data rates are summarized in Supplementary Section 4.

Moreover, similarly like the case of DNN, the system performance of bi-GRU for data rates of 360 Gbps, 380 Gbps and 400 Gbps under different patterns of PRBS are analyzed, as shown in Fig. 6a. There is a similar performance under different signal patterns, which proves that the implemented bi-GRU equalizer captures the channel response rather than PRBS pattern. Normally, as shown in Fig. 6b, with more GRU units in the hidden layer, the BER will be better. From 340 Gbps to 400 Gbps, it is effective to reduce BERs by changing GRU numbers,

and there is no obvious downward trend found in this effectiveness, which reflects the applicability of the GRU solution to higher rates. In the specific experiment, 200 GRU units is chosen for the equalizer here. In more detail, the comprehensive discussions on AI equalizer network configurations are provided in Supplementary Section 5. Simultaneously, the robustness of the AI network equalizers is also analyzed and the improvement methods are given in Supplementary Section 7.

The PAM-4 results based on the AI-accelerated slow-light technology demonstrate the great potential of silicon modulators for ultrahigh-speed applications. The ability of pure silicon modulators to achieve such high rates and maintain low BERs in PAM-4 signals, the simplest high-order format that complies with IEEE standards, is an encouragement for the deployment of silicon photonics. Moreover, in order to verify the scalability of our AI-accelerated slow-light system for higher-order signals, we utilized high-speed PAM-8 signal onto the Si-SLM chip. Even for PAM-8 signals with more levels compared to PAM-4, the bi-GRU equalizer still possesses a powerful ability for mitigating the nonlinear distortions in muti-level signals. Here, the specific PAM-8 eye diagrams at 1550 nm (Ch3) from 240 Gbps to 390 Gbps are demonstrated in Fig. 7a, with all BERs below HD-FEC threshold. Also, the constellations corresponded to each data rates are illustrated, with favorable separation between the 8-levels. For WDM systems, similar to the PAM-4 results, the BERs for all wavelengths in 8 channels are still below HD-FEC threshold in Fig. 7b (all the PAM-8 results are demonstrated in Supplementary Section 4). Figure 7c illustrates the BER curves changing with data rates, and the consistency between channels are quite well. Compared with the PAM-4 format, when the data rates is around 240 Gbps, the BERs of PAM-8 is slightly lower than BERs of PAM-4. This is because the baud rate of PAM-8 (80 Gbaud) is at a fairly low level, while the PAM-4 baud rate already reaches 120 Gbaud. For a relatively low baud rate especially under 100 Gbaud, the PAM-8 signal is more advantageous. However, as the data rate continues to rise, the BERs of PAM-8 increase indeed faster than that of PAM-4, and the two are basically the same when the data rate reaches around 400 Gbps. The reason is that, as the data rate rises, the nonlinear compensation required for PAM-8 with more signal levels at ultrahigh-speed region gradually increases compared with PAM-4 under the same linearity. Despite this, it is effective to increase the total rate under the limited baud rate by adopting advanced formats with more levels, and our approach can also adapt to the higher-order formats.

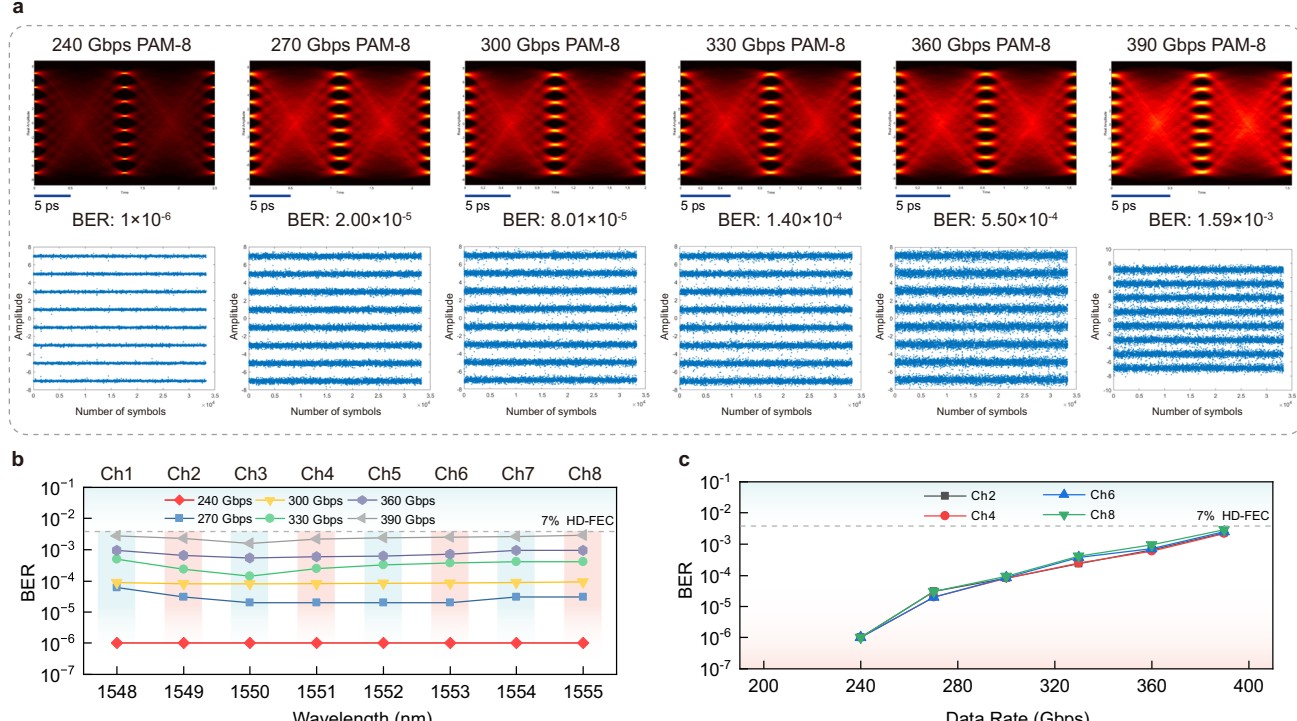

**Fig. 7 | bi-GRU transmission results of PAM-8 signal. a** bi-GRU PAM-8 eye diagrams and constellations of different data rates at 1550 nm. 240 Gbps, 270 Gbps, 300 Gbps, 330 Gbps, 360 Gbps and 390 Gbps are shown here, with all BERs under HD-FEC threshold. Favorable separations are achieved between the 8-levels. **b** BER curves of all 8-channel PAM-8 signals with respect to different data rates. Similarly to PAM-4 signal, all the BERs for different wavelengths in 8 channels are below HD-FEC threshold with high passband consistency. **c** BER curves of PAM-8 signal with increasing data rates. Ch2, Ch4, Ch6 and Ch8 are shown here. The nearly overlapped curves illustrate the performance uniformity between different channels.

The above experimental results are based on back-to-back (B2B) scenario, and the transmission penalty for 100 m, 200 m and 300 m standard single-mode fiber (SSMF) is then experimentally assessed for both PAM-4 and PAM-8 signals with the bi-GRU equalizer. Considering the consistency of all channels has been proven to be favorable, the BER penalty is illustrated for one channel (Ch3) at 1550 nm. The transmission results for different SSMF distances are demonstrated in Fig. 8, in which Fig. 8a shows the BERs of PAM-4 (280 Gbps, 320 Gbps, 360 Gbps, 400 Gbps) and Fig. 8b shows the BERs of PAM-8 (300 Gbps, 330 Gbps, 360 Gbps, 390 Gbps). Due to the impact of the power fading effect in optical fiber, the BER performance gradually degrades with increasing transmission distance compared to B2B scenario for both cases of PAM-4 and PAM-8. For the transmission distance of 300 m, the BERs still remain below HD-FEC threshold at 360 Gbps, but deteriorate beyond HD-FEC threshold at 400 Gbps.

Furthermore, the transmission results of employing T-biGRU equalizer (Fig. 9a) are evaluated for both PAM-4 (400 Gbps in Fig. 9b) and PAM-8 (390 Gbps in Fig. 9c). For BER curves of different channels (Ch2, Ch4, Ch6, Ch8), PAM-4 results (280 Gbps, 320 Gbps, 360 Gbps, 400 Gbps) are shown in Fig. 9d, while PAM-8 results (300 Gbps, 330 Gbps, 360 Gbps, 390 Gbps) are shown in Fig. 9e. The results indicate that by employing T-biGRU equalizer, the BER performance for around 400 Gbps transmission of all channels can be improved to smaller than $10^{-3}$, which is slightly better than the bi-GRU case. The T-biGRU results for all eye diagrams, constellations and BERs corresponding to Fig. 9 are demonstrated in Supplementary Section 4. Meanwhile, the performance comparison of three AI network equalizers (DNN, bi-GRU, T-biGRU) together with traditional algorithms (DFE, VNLE) is shown in Supplementary Section 6. However, it is worth noting that the computational complexity of the T-biGRU is 1.5 times that of the bi-GRU (see Methods), leading to increased multiplications and higher power consumption

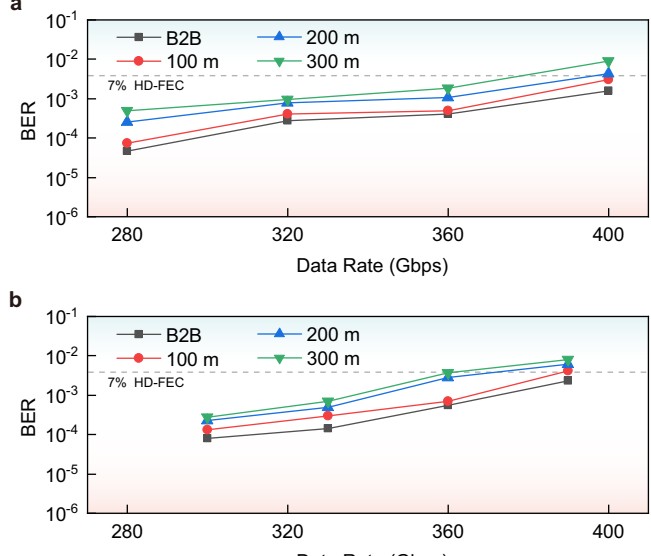

**Fig. 8 | BER performance for different transmission distances. a** PAM-4 signal of 280 Gbps, 320 Gbps, 360 Gbps, 400 Gbps for B2B, 100 m, 200 m and 300 m SSMF. **b** PAM-8 signal of 300 Gbps, 330 Gbps, 360 Gbps, 390 Gbps for B2B, 100 m, 200 m and 300 m SSMF. The bi-GRU equalizer is utilized here. For the transmission distance of 300 m, the BERs remain below HD-FEC threshold at 360 Gbps, but deteriorate beyond the HD-FEC threshold at 400 Gbps.

(Supplementary Section 8). In the equalizer selection, it is important to strike a balance among performance, computational complexity, and power consumption to achieve the desired system performance effectively.

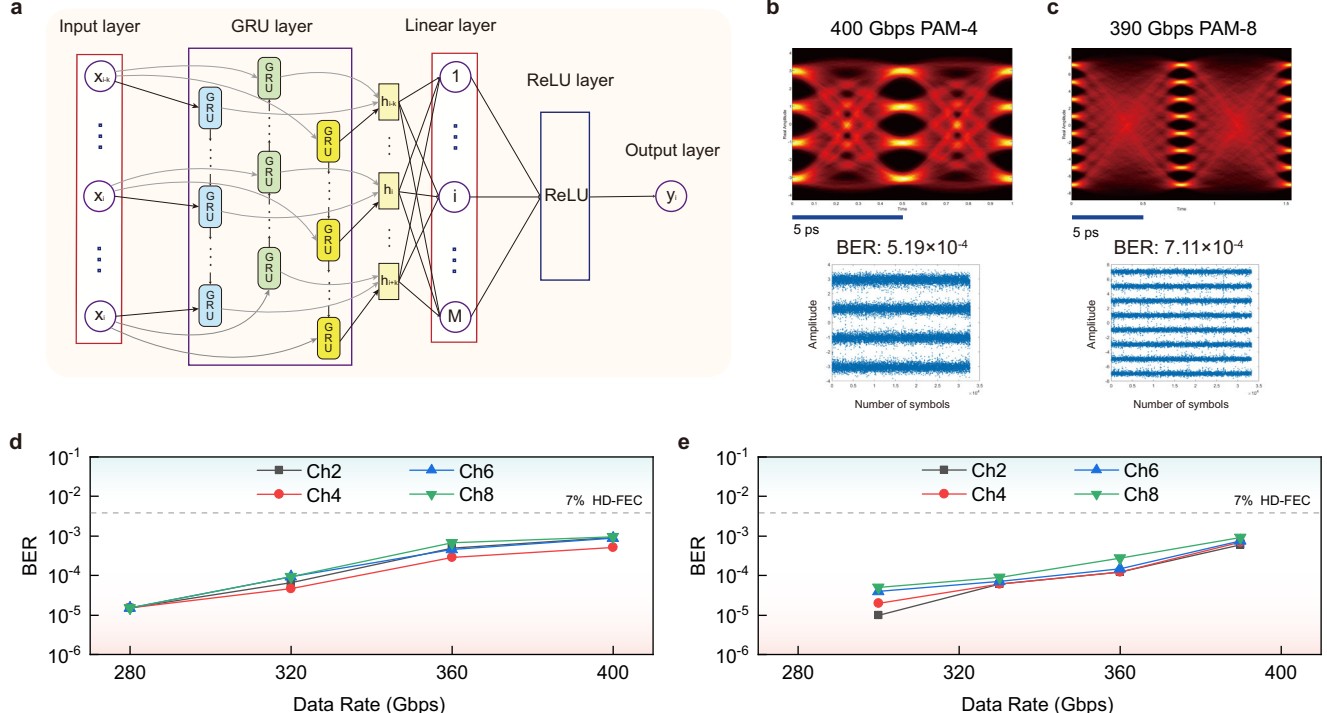

**Fig. 9 | Three-layer bidirectional gated recurrent unit (T-biGRU) transmission results. a** The architecture of T-biGRU equalizer. By integrating forward, backward and repeated forward GRU processing, the T-biGRU model extracts global and local features of the sequence more comprehensively, thereby further enhancing equalization performance. **b** T-biGRU PAM-4 eye diagram and constellation at 400 Gbps. **c** T-biGRU PAM-8 eye diagram and constellation at 390 Gbps. **d** BER curves of PAM-4 signal with increasing data rates (280 Gbps, 320 Gbps, 360 Gbps and 400 Gbps) of different channels. **e** BER curves of PAM-8 signal with increasing data rates (300 Gbps, 330 Gbps, 360 Gbps and 390 Gbps) of different channels. Ch2, Ch4, Ch6 and Ch8 are shown here, with all BERs lower than $10^{-3}$.

## Discussion

To explore 400 Gbps per wavelength of silicon photonics, we comprehensively optimize the device metrics with careful design. By adopting the slow-light approach, the bandwidth-efficiency trade-off for pure silicon modulators is greatly optimized, paving the way for 400 Gbps transmission per wavelength. Our compact Si-SLM presents an ultrahigh EO bandwidth of 90 GHz and a remarkable modulation efficiency of 0.82 V·cm, while possessing a wide optical passband of 7 nm around 1550 nm. Thanks to the high CMOS compatibility of Si-SLMs, we fabricate an 8-channel Si-SLM chip on a SOI platform using a standard silicon photonic process. In the fabrication flow, no additional heterogeneous materials and complex process are introduced, and this superiority provides a prerequisite for large-scale and low-cost production at the wafer level. Our Si-SLM chip, with its favorable basic performance, provides a foundation for transmission capacity enhancement, and the advantage of wafer-level production utilizing the standard silicon photonic process ensures the feasibility of extensive deployment of this solution.

More specifically, Table 1 summaries the properties of the representative results of Si-MZMs, Si-MRMs and Si-SLMs on SOI platforms in recent years. From the comparison, we can see that the metrics including bandwidth, modulation efficiency, and passband of our designed Si-SLMs are all relatively prominent for pure silicon modulators. While the bandwidth of our device exceeds that of previous modulators, the modulation efficiency has been improved compared to Si-MZMs, and the optical passband is greater than that of Si-MRMs (a full width at half maximum in one resonance period). Also, the transmission performance of our device is ahead of other Si-SLMs. It can be seen that, in recent years, although the transmission rate has continued to raise with the proposal and optimization of various solutions, the overall upward trend is still slow. Notably, our modulator has achieved single-lane 400 Gbps transmission using the industry-

standard format PAM-4, with all BERs below HD-FEC threshold in a wide optical passband. As the highest single-lane rate demonstrated in standard silicon photonic platforms so far, this record-high result reveals the great high-speed transmission potential of silicon photonics.

The precise adoption of AI technology is an important guarantee for realizing the potential of our Si-SLM chips, and the slow-light systems are well suitable for accelerated driving with ANN equalizer. While the slow-light approach has maximized the overall performance of pure silicon modulators, its physical essence is still based on non-linear modulation of doped silicon material (the prerequisite for large-scale industrial application), and the ANN model specializes in compensating for nonlinear distortions exactly. By embedding ANN into the signal processing workflow of Si-SLMs, the intrinsic nonlinear distortions in pure silicon devices can be effectively mitigated. At the manufacturing level, since silicon material does not possess a high and consistent EO coefficient, forming PN junction is the basis for realizing pure silicon modulation, while achieving complete uniformity in doping process is still challenging. Simultaneously, although the CMOS-compatible waveguide grating structure has already improved process stability compared to photonic crystals, the structural fluctuations of doped silicon gratings still exist due to the unavoidable fabrication deviations. Therefore, through AI equalization at the back end, the variations between Si-SLMs caused by process errors can be eliminated, thereby making the deployment of large-scale transmission systems possible. Here, three ANN models have been developed here, including the DNN, bi-GRU and T-biGRU equalizers. First, we adopt the simple DNN equalizer with only two hidden layers, which takes up less computing resources, and achieve a total capacity of 1.6 Tbps with 224 Gbps PAM-4 per lane under the IEEE rate standard. More importantly, by adopting the bi-GRU equalizer, we achieve an ultrahigh-speed transmission of 400 Gbps per lane and a total capacity

**Table 1 | Comparison with representative silicon modulators**

| Ref. | Type | Operating wavelength (nm) | EO bandwidth (GHz) | Modulation efficiency (V · cm) | Optical passband (nm) | Data rate (Gbps) |
|---|---|---|---|---|---|---|
| 27 | Si-MZM | C-band | 60 | 1.40 | NA | 1 × 128PAM-4 (SD-FEC) |
| 29 | Si-MZM | C-band | 60 | NA | NA | 4 × 200PAM-4 (SD-FEC) |
| 30 | Si-MZM | O-band | 47 | 1.35 | NA | 1 × 225PAM-8 (SD-FEC) |
| 31 | Si-MZM | C-band | 67 | 3.00 | NA | 1 × 240ASK-4 (HD-FEC) 1 × 360ASK-8 (SD-FEC) |
| 32 | Si-MZM | C-band | 65 | 1.90 | NA | 1 × 224PAM-4 (SD-FEC) |
| 35 | Si-MRM | O-band | 50 | 0.52 | ~ 0.3 | 1 × 128PAM-4 (SD-FEC) |
| 37 | Si-MRM | C-band | 67 | NA | ~ 0.5 | 1 × 200PAM-4 (HD-FEC) |
| 38 | Si-MRM | O-band | 60 | 0.80 | 0.28 | 1 × 240PAM-4 (SD-FEC) |
| 39 | Si-MRM | C-band | 50 | 0.63 | ~ 0.3 | 1 × 260PAM-4 (SD-FEC) 1 × 330PAM-8 (SD-FEC) |
| 40 | Si-MRM | O-band | 48 | 0.60 | 0.35 | 5 × 200PAM-4 (SD-FEC) |
| 43 | Si-SLM | C-band | 38 | 0.44 | 15.0 | 1 × 64PAM-4 (SD-FEC) |
| 46 | Si-SLM | C-band | 40 | 0.51 | 2.0 | 1 × 90PAM-4 (SD-FEC) |
| This work | Si-SLM | C-band | 90 | 0.82 | 7.0 | 8 × 400PAM-4 (HD-FEC) 8 × 390PAM-8 (HD-FEC) |

of 3.2 Tbps based on the Si-SLM chip, utilizing the standard high-order format PAM-4, with all BERs under HD-FEC threshold. Considering the ultra-compact occupied area (4 mm × 0.5 mm), we achieved an on-chip data-rate density of 1.6 Tb/s/mm². Meanwhile, benefiting from the wide flat passband around 1550 nm, the whole links are without individual resonant wavelength adjustment and TEC operating platforms, thus reducing the system budget. Next, we have verified the feasibility of the PAM-8 signal, demonstrating the applicability of our AI-accelerated slow-light solution for higher-order data transmission with more levels. Furthermore, the transmission performances of employing the T-biGRU equalizer are evaluated, and the BERs for around 400 Gbps transmission can be slightly improved for both PAM-4 and PAM-8 signals.

In addition, this work illustrates the potential of AI technology in photonics filed, and the proposed AI-accelerated Si-SLM system is a good exemplification of "AI for SiPh" together with "SiPh for AI". For the optical interconnections in computing centers, the AI-accelerated Si-SLM technology provides an ultrahigh-speed deployment solution that enables a leap in single-lane transmission rate of the interfaces. Therefore, the computility of the computing centers can be improved, thus leading to faster training iterations of the ANN. This enhanced ANN will promote the signal processing for Si-SLM systems to be more immediate, thereby further improving the interconnection rates for computing centers. In short, a self-optimizing positive feedback loop between computing centers and Si-SLM systems through ANN can be constructed based on this bi-directional promotion mechanism, leading to a continuous improvement in the whole architecture. Especially, through the efficient photonic neural networks, the deep fusion of optical interconnection and optical computing systems based on this synergy can capitalize on the inherent advantages of photonics to facilitate real-time system optimization.

In summary, we have demonstrated the first pure silicon photonic system with a total capacity of 3.2 Tbps based on AI-accelerated slow-light technology on a SOI platform. Integrating AI technology with Si-SLM systems offers a promising pathway to overcome the limitations of silicon photonics in ultrahigh-speed transmission. By utilizing the industry-standard IM/DD format PAM-4, we achieve 400 Gbps/λ optical transmission in all wavelength channels, with BERs below HD-FEC threshold. To our best knowledge, this is the highest single-lane transmission rate demonstrated in standard silicon photonic platforms. Under precise regulation, our compact thermal-insensitive Si-SLM chip possesses an ultrahigh bandwidth, a remarkable modulation efficiency, a wide optical passband simultaneously, leading to an on-

chip data-rate density of 1.6 Tb/s/mm², without individual resonant wavelength adjustment requirements and additional TEC operating platforms. This proposed AI-accelerated slow-light technology increases the transmission rate of silicon photonic significantly, paving the way for 400 Gbps optical transmission per lane. Our work demonstrates the great value of AI for silicon photonics and highlights the potential of standard silicon photonic platforms for next-generation optical interconnections of 3.2 TbE and beyond.

## Methods

### Fabrication of the devices

The silicon slow-light structure is CMOS-compatible, thus the Si-SLMs can be fabricated under a standard silicon photonic process, leveraging the industrial advantages of silicon photonics. The Si-SLM chips adopt a standard 90 nm photolithography process on a 200-mm SOI wafer with a silicon thickness of 220 nm at CompoundTek Pte. The 90-nm-thick rib area is etched partially for carrier doping and metal contact in 220-nm-thick silicon. The concentrations of P-type and N-type doping in the depletion-mode PN junction are both $5.0 \times 10^{17}$/cm³, to ensure a sufficiently high electrical bandwidth. All feature parameters of the slow-light waveguide satisfy the requirements of the commercial silicon photonic foundry, providing a basis for large-scale and low-cost production at the wafer level.

### Experimental details

In the electro-optical response test, a vector network analyzer (Keysight PNA-X Network Analyzer N5247B) with its lightwave component analyzer (Keysight LCA Optical Receiver N4372E) is adopted to measure the bandwidth. For the optical test, the spectra are obtained by a high-resolution optical spectrum analyzer (Yokogawa AQ6370C). For the transmission experiments, the high-speed signal is generated by an arbitrary wave generator (Keysight M8199B), and then the differential signals are amplified by a commercial SHF single-ended driver to obtain a 5-V Vpp and then injected into the modulator working under the dual-drive push-pull configuration. After the electro-optical conversion, the modulated optical signal is amplified by an Amonics pre-amp erbium-doped fiber amplifier to compensate for the loss. At the receiving end, the signals are fast sampled by an oscilloscope (Keysight UXR-Series 256-GSa/s real-time) and afterward offline processed.

### Algorithm details

**DNN.** The implemented lite DNN equalizer consists of an input layer with $M$ neurons, followed by two hidden layers with $N_1$ and $N_2$ neurons,

respectively. The architecture culminates in an output layer with a single neuron. The DNN is trained using the mean-square-error (MSE) criterion, employing the back-propagation (BP) algorithm. The training process is executed in two distinct steps. Initially, the forward propagation phase calculates the output based on the input feed. This process is detailed as

$$
\begin{cases}
Y_1(n) = f(W_1(n)^T X(n)) \\
Y_2(n) = f(W_2(n)^T Y_1(n)) \\
y_{out}(n) = f(W_3(n)^T Y_2(n))
\end{cases}
\tag{1}
$$

where

$$
X(n) = \begin{bmatrix} x_1(n) & x_2(n) & \dots & x_M(n) \end{bmatrix}^T
\tag{2}
$$

$$
W_1(n) = \begin{bmatrix}
w_{1,11}(n) & w_{1,12}(n) & \dots & w_{1,1M}(n) \\
w_{1,21}(n) & w_{1,22}(n) & \dots & w_{1,2M}(n) \\
\dots & \dots & \dots & \dots \\
w_{1,N_11}(n) & w_{1,N_12}(n) & \dots & w_{1,N_1M}(n)
\end{bmatrix}^T
\tag{3}
$$

$$
W_2(n) = \begin{bmatrix}
w_{2,11}(n) & w_{2,12}(n) & \dots & w_{2,1N_1}(n) \\
w_{2,21}(n) & w_{2,22}(n) & \dots & w_{2,2N_1}(n) \\
\dots & \dots & \dots & \dots \\
w_{2,N_21}(n) & w_{2,N_22}(n) & \dots & w_{2,N_2N_1}(n)
\end{bmatrix}^T
\tag{4}
$$

$$
W_3(n) = \begin{bmatrix} w_{3,1}(n) & w_{3,2}(n) & \dots & w_{3,N_2}(n) \end{bmatrix}^T
\tag{5}
$$

$$
Y_1(n) = \begin{bmatrix} y_{1,1}(n) & y_{1,2}(n) & \dots & y_{1,N_1}(n) \end{bmatrix}^T
\tag{6}
$$

$$
Y_2(n) = \begin{bmatrix} y_{2,1}(n) & y_{2,2}(n) & \dots & y_{2,N_2}(n) \end{bmatrix}^T
\tag{7}
$$

$f(\cdot)$ is the activation function (Supplementary Section 3), as shown in the DNN transmission parts, $X(n)$ is the input vector, $Y_1(n)$ and $Y_2(n)$ are the outputs of the first and second hidden layers respectively, $y_{out}(n)$ is the output signal, $W_1$, $W_2$ and $W_3$ are the weight matrices between different layers of the DNN equalizer.

The second step in the training process is back-propagation, where the calculated error term is propagated from the output layer back to the input layer. This mechanism allows for the adjustment of the network's weights to minimize the error. The square error for a single training example is defined as

$$
E(X(n), info(n)) = |y_{out}(n) - info(n)|^2
\tag{8}
$$

where $info(n)$ is the desired signal.

The stochastic gradient descent is utilized to optimize the cost function with respect to the weight matrices, including $W_1$, $W_2$ and $W_3$. The updated weight matrix for the subsequent iteration can be derived by

$$
W_i(n+1) = W_i(n) - \eta \frac{\partial E(X(n), info(n))}{\partial W_i(n)} \qquad i = 1, 2, 3
\tag{9}
$$

where $\eta$ indicates the learning rate of the network.

Consider the four-layer DNN equalizer with $n_{in}$, $n_H$, $n_{out}$ nodes in the input, hidden and output layer, respectively. Typically, the output layer has a fixed single node. Here, $n_{in}$ can be viewed as the tap number in a digital filter. The primary computational burden in the equalizer arises from the error propagation needed to calculate the squared error derivative of each node in all the hidden layers. Compared to the DNN, which employs a *sigmoid* function with two saturation regions, the DNN that employs multi-level *sigmoid* does not incur additional computational requirements. The computational complexity of the DNN equalizer per iteration can be characterized from the program as:

$$
C_{DNN} = 8n_H^2 + 7n_{in}n_H + 12n_H + 4n_{in} + 3n_{out}
\tag{10}
$$

**GRU**. A GRU unit is composed of a reset gate $r_t$ and an update gate $z_t$. The output $h_t$ is determined by both current input $x_t$ and previous state $h_{t-1}$ under the control of these two gates. The outputs of the gates and the GRU unit are calculated as follows:

$$
\begin{aligned}
r_t &= \sigma(W_r x_t + U_r h_{t-1} + b_r) \\
z_t &= \sigma(W_z x_t + U_z h_{t-1} + b_z) \\
\tilde{h}_t &= tanh\left[W_h x_t + U_h(r_t \odot h_{t-1}) + b_h\right] \\
h_t &= (1 - z_t) \odot h_{t-1} + z_t \odot \tilde{h}_t
\end{aligned}
\tag{11}
$$

where $W_r$, $U_r$, $W_z$, $U_z$, $W_h$ and $U_h$ are the weight matrices. $b_r$, $b_z$, $b_h$ are the synthesis of bias vectors for input $x_t$ and previous state $h_{t-1}$, $\sigma$ is the logistic sigmoid function, *tanh* is the hyperbolic tangent activation function, $\odot$ denotes the Hadamard product.

Models with bidirectional structure are capable of learning information from both preceding and following data when processing the current data. The bi-GRU model employed in this work comprises two unidirectional GRU layer operating in opposite directions. By combining forward and backward GRU processing, the model incorporates information from both the future and the past to influence its current states. The bi-GRU model relies on the states of two GRU layers, whereas the T-biGRU model utilizes the states of three GRU layers. By integrating forward, backward and repeated forward GRU processing, the T-biGRU model more comprehensively extracts both global and local features of the sequence, thereby further enhancing equalization performance. The bi-GRU model can thus be mathematically described as follows:

$$
\begin{aligned}
\overrightarrow{h}_{t\,bi-GRU} &= GRU_{fwd}(x_t, \overrightarrow{h}_{t-1}) \\
\overleftarrow{h}_{t\,bi-GRU} &= GRU_{bwd}(x_t, \overleftarrow{h}_{t+1}) \\
h_{t\,bi-GRU} &= \overrightarrow{h}_{t\,bi-GRU} \oplus \overleftarrow{h}_{t\,bi-GRU}
\end{aligned}
\tag{12}
$$

where $\overrightarrow{h}_{t\,bi-GRU}$ and $\overleftarrow{h}_{t\,bi-GRU}$ is the state of the forward and backward GRU, respectively. $h_{t\,bi-GRU}$ is the output of bi-GRU. $\oplus$ indicates the operation of concatenating two vectors.

For the case of T-biGRU:

$$
\begin{aligned}
\overrightarrow{h}_{t1\,T-biGRU} &= GRU_{fwd}(x_t, \overrightarrow{h}_{t-1}) \\
\overleftarrow{h}_{t\,T-biGRU} &= GRU_{bwd}(x_t, \overleftarrow{h}_{t+1}) \\
\overrightarrow{h}_{t2\,T-biGRU} &= GRU_{fwd}(x_t, \overrightarrow{h}_{t-1}) \\
h_{t\,T-biGRU} &= \overrightarrow{h}_{t1\,T-biGRU} \oplus \overleftarrow{h}_{t\,T-biGRU} \oplus \overrightarrow{h}_{t2\,T-biGRU}
\end{aligned}
\tag{13}
$$

where $\overrightarrow{h}_{t1\,T-biGRU}$, $\overleftarrow{h}_{t\,T-biGRU}$ and $\overrightarrow{h}_{t2\,T-biGRU}$ are the states of the first forward GRU, the backward GRU and the second forward GRU, respectively. $h_{t\,T-biGRU}$ represents the output of T-biGRU.

The main computation complexity of one GRU layer can be decribed as:

$$
C_{GRU} = 3n_H(n_E + n_H)
\tag{14}
$$

where $n_E$ refers to the input size of the GRU layer, and $n_H$ represents the number of GRU units that used in the layer. The complexity of bi-GRU and T-biGRU layer can be calculated as:

$$
\begin{aligned}
C_{bi-GRU} &= 2 \times 3n_H(n_E + n_H) \\
C_{T-biGRU} &= 3 \times 3n_H(n_E + n_H)
\end{aligned}
\tag{15}
$$

## Reporting summary

Further information on research design is available in the Nature Portfolio Reporting Summary linked to this article.

## Data availability

The data that support the plots within this paper and other findings of this study are available on Zenodo database [https://doi.org/10.5281/zenodo.15631151]. All other data used in this study are available from the corresponding authors upon request.

## Code availability

The codes that support the findings of this study are available from the corresponding authors upon request.

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

## Acknowledgements

The authors thank Xi Xiao, Peiqi Zhou, Qiansheng Wang in National Information Optoelectronics Innovation Center for testing support. This work was supported by National Key Research and Development Program of China (2022YFB2803700 to H.S.), National Natural Science Foundation of China (62235002 to X.W., 62327811 to X.W., 12204021 to H.S., 62322501 to H.S., 12374340 to J.Q.), Beijing Municipal Science and Technology Commission (Z221100006722003 to X.W.), Beijing Municipal Natural Science Foundation (Z210004 to X.W.), IPOC (2021A03 to J.Q.) and Major Key Project of PCL to X.W.

## Author contributions

The experiments were conceived by C.H. The devices were designed by C.H. The equalizers were developed by J.Q. The experiments were performed by C.H., Q.Y. and Y.Zhou, with the assistance from Y.Zhang, H.W., L.W., Z.H., S.Y. and W.H. The slow light theory was developed by Z.Z and C.P. The data process was conducted by C.H., J.Q., Y.S., J.L. and Y.Wu. The device characterization was conducted by C.H., Q.Y., Y.Zhou. and Z.G. The results were analyzed by C.H., Q.Y., J.Q. and Y.Zhou. The figure optimization was conducted by C.H., Q.Y., Y.S, J.L. and Y.Wang. All authors participated the discussion of the research. The project was supervised by H.S., J.E.B. and X.W.

## Competing interests

The authors declare no competing interests.
