## [Transparent Peer Review file · Nature Communications]

Exploring 400 Gbps/ λ and beyond with AI-accelerated silicon photonic slow-light technology

Corresponding Author: Professor Xingjun Wang

Version 0:

Reviewer comments:

Reviewer #1

(Remarks to the Author)

This paper reports a record high-speed, pure silicon Mach-Zehnder modulator with slow light phase shifters of a small footprint. Compensating for the nonlinear response using an equalizer optimized by AI technologies, it worked beyond 200 Gbaud in PAM-4 format, resulting in >400 Gbps data rate at single wavelength and single lane, showing very clean eye diagrams which satisfy BER level below HD FEC. Due to a wide working spectrum of slow light and high uniformity, it potentially achieve a Tbps class transmission using WDM. It also allows the packaging of all channels in a compact pluggable form factor, thanks to the small footprint. These results are impressive, and it is worth publishing in Nature Communications after some major revisions. My comments are listed below.

1. A similar device has already been reported in Ref. 43, showing the EO bandwidth beyond 100 GHz. In this paper, it is also shown to be as wide as 90 GHz. I wonder if these wide bandwidths are attributed to a small RC constant. In the short slow-light phase shifter, the capacitance C is reduced, while parasitic resistance R is increased. This increase in R is reduced by reducing the distance between the PN junction and the highly doped regions, sacrificing the low optical loss. Please discuss more clearly the relation between the doping profiles, optical loss, group index of slow light and EO bandwidth.

2. The driving voltage V_{pp} is not shown in this paper. Please write it for each result. The modulation efficiency is written to be 0.82 Vcm, and the phase shifter length is 249 microns, the $V_{pi} = 33$ V. If I assume $V_{pp} = 5$ V written in Ref. 43, the phase shift is only 0.15pi. If the operating is centered at the quadrature point, the nonlinearity is minimized. Actually, the IMD3 is more than 40 dB lower than the fundamental wave, as shown in Fig. 2e. In such a situation, why was the optimizing of the equalizer so effective, particularly for PAM-4?

3. The authors mention that the nonlinearity of the slow light modulator is similar to that of standard Si modulators operated by carrier plasma dispersion. Does it mean that AI-assisted equalizer can improve the signal quality not only of the slow light modulator but also of other modulators? If so, this study includes two separate topics on slow light modulator and AI-assisted equalizer, and the sentence "AI-powered silicon photonic slow light technology" in the title is misleading.

4. In the acquisition of eye diagrams, the electrical signal was amplified using SHF S807C whose electrical bandwidth was limited to 55 GHz. I wonder if the optimized equalizer compensated for the nonlinearity of these electrical devices used.

5. Considering the small phase shift noted in Comment 2, the extinction ratio must be small, although it is not shown in this paper. Please show this and give some discussion.

6. The working spectral width is shown to be 7 nm at 1550 nm wavelengths, which corresponds to a frequency range of 0.87 THz. It is too narrow for 3.2 Tbps transmission using 8 wavelengths WDM, isn't it?

In addition, the paper seems to me long and very redundant. Please simplify well-known backgrounds, and remove repeated arguments.

Reviewer #2

(Remarks to the Author)

Exploring 400 Gbps/λ and beyond with AI-powered silicon photonic slow-light technology
by Han Chang-hao et al.

The submission represents an interesting improvement over existing similar technologies, and it would be well suited to be published in a technically oriented journal. Papers published by Nature Communications "aim to represent important advances of significance to specialists within each field", and its article format requires "a novel and important research study of high quality and of interest to that specific research community".

On the other hand, the Journal of Lightwave Technology, for example, invites "experimental results which advance the technological base of guided-wave technology". While the scopes of these journals have some overlap, the present submission advances well-known technologies incrementally, while it lacks groundbreaking results.

Therefore, I cannot recommend acceptance of the submission for publication, even after an urgently required major revision.

However, I want to list a few comments which could guide the authors toward an improved version of their report.

The submission consists of two separate parts which could stand for themselves. Certainly, the experiments combine both parts by employing a slow-light modulator (A) with AI-supported reception (B). The title tries to forge a connection between these basically unrelated parts and suggests that the slow-light technology is achieved by artificial intelligence - which is not the case. In the following, I shall comment on both parts (A and B) separately.

A1. The paper closest in topic to the present submission is (let aside the segmentation for avoiding a DAC):

[39] O. Jafari, S. Zhalehpour, W. Shi, S. LaRochelle: DAC-Less PAM-4 Slow-Light Silicon Photonic Modulator Providing High Efficiency and Stability. JLT 39 (2021) 5074-5082. doi:10.1109/JLT.2021.3083140

This paper is an excellent example of the way results should be presented: The basic physical mechanisms are clearly explained, measurement setups are elaborated, experimental results are well discussed, and the achievements are put in context with the state of the art.

A2. Reference [39] discusses exactly the same modulator structure as in the submission. Therefore I do not see the novelty. The improvements are incremental.

- As an aside: The design of a slow-light 100Gbit/s modulator with an optical bandwidth of 1THz was published as early as 2008: Brosi, J.-M.; Koos, Ch.; Andreani, L. C.; Waldow, M.; Leuthold, J.; Freude, W.: High-speed low-voltage electro-optic modulator with a polymer-infiltrated silicon photonic crystal waveguide, Opt. Express 16(2008) 4177–4191. doi:10.1364/OE.16.004177

- The claim "Our approach breaks the transmission limits of silicon photonics" cannot be interpreted as a "first" in view of [39]. However, this is suggested by the quoted sentence.

- In contrast to [39] the submission lacks an explanation of the structural design.

- Fig.1a is unclear. What is meant with "shadow areas"? Covered areas with an unrelated design? Then say so!

- The inset "Photonics chip photograph" in Fig.1 "(a) Optical micrograph of the Si-SLM transmitter chip, with the photograph of the overall chip as the inset." is irritating.

- The explanation Fig.1 "(b) Device morphology of the fabricated Si-SLM. A dual-drive architecture of GSGSG-type RF electrode under the Mach-Zehnder interferometer is adopted. The modulation arms are only 249 μm long and the SEM image of the fabricated slow-light waveguide is shown as the inset below." does not help in understanding.

- Neither does the main text provide a clear explanation (line 150-156): "Based on 151 the coupled-resonator optical waveguide (CROW), the depletion-mode PN junction has a periodic structure and the SEM image of the fabricated slow-light waveguide is illustrated below the device photograph, in which a complete resonator is constructed by a phase shifter and Bragg gratings with a period of around 300 nm on both sides. In our reconfigurable slow-light model, several resonators are cascaded to construct the modulation arm, and the parameters can be selected flexibly, according to specific application scenarios."

- Is the phase shifter only in the centre, or are not the Bragg gratings part of the junction structure, too?

- Is only the broader part between the mirrors regarded as a resonator? I would understand that a resonator consist of two mirrors and a light propagation region in-between.

- On chip resistors (which value?) can only be guessed, but not seen in Fig.1b.

- What is the meaning of the RHS of Fig.1b labeled "TiN heater"?

- The term "modulation efficiency" is frequently used, but is nowhere defined and can be guessed only from the units.

- In this context: What was the actual drive voltage swing during operation? With $V_{\pi} L = 8.2\text{Vmm}$ and $L = 250\mu\text{m}$, $V_{\pi} = 33\text{V}$ would result, an extraordinarily large value for a drive frequency of 90GHz and a symbol rate of 200GBd.

- No insertion loss of the modulator was provided, neither fibre-to-fibre, nor for the modulator alone.

- What was the modulator chirp and the interdependence of phase and amplitude modulation of the pn junctions?

- How would this "small" chirp influence the transmission rate through a 300m long fibre?

- What was the light power injected into the modulator?

- What was the energy requirement per bit?

- What was the slow-down factor in the slow-light waveguide?

- What was the wave impedance of the drive electrodes?

- Fig.1g and Table 1 name a "Single passband of 7nm" (corresponding to about 875GHz), but the (unexplained) Fig.S2a says that with "5 resonators" the optical bandwidth would be at best close to 250GHz.

B1. The AI equalizer uses deep neural networks (DNN) and gated recurrent units (GRU) as a gating mechanism in recurrent

neural networks. Both techniques were extensively treated in numerous publications, and even Wikipedia has articles with detailed descriptions.

B2. I do not see the novelty in providing "Algorithm details" instead of specifying the energy requirements, and how often the learning process must be repeated if any of the optical or electronic components has to be exchanged, or if the modulator ages and/or its bias drifts.

- I feel that the energy discussion (including the light source, the modulator, the electrical and the optical amplifiers as well as the AI equalizers and the individual energy for cooling where required) would be of paramount importance, if these devices, as is suggested by the submission, should become the future basis for optical interconnects inside a data center.
- In this case, the data of a transmission experiment with an optical fibre over a distance of at least 300m would be required.
- What would be the latency of such an optical interconnect?

I could go on for a long time in listing the numerous deficiencies of the submission, but I think it suffices for understanding why my recommendation is "reject, no revision".

Reviewer #3

(Remarks to the Author)

This paper presents an excellent idea of using AI to compensate for waveform distortion that arises when increasing the modulation frequency with slow-light technology, enabling the use of higher-order modulation formats. The authors demonstrate this concept effectively through a remarkable experimental achievement: achieving a data transmission rate of 400 Gbps per channel across 8-channel WDM on a Si-SLM chip, totaling 3.2 Tbps.

The technique of enhancing modulation frequency using slow light is an impressive approach previously described by the same authors (Han et al., Sci. Adv. 9, eadi5339 (2023) 20 October 2023). This work extends that concept by incorporating AI, yet the description of AI's specific impact seems limited, making it challenging to assess the level of innovation required by NCOM. If left as it is, Sci. Adv. might be a suitable publication venue. However, for potential acceptance in NCOM, the following points should be addressed:

1. Although the paper provides detailed information on Si-SLM, it would be beneficial to clarify how ANN can mitigate nonlinearities in Si-SLM. For example, showing how much the spurious-free dynamic range (SFDR) expands after applying a DNN equalizer to Si-SLM would strengthen the paper.
2. The output values following the GRU equalizer should be explicitly stated as either discrete or continuous. Comparing them to SD-FEC thresholds may be meaningless if they are discrete.
3. In real transmission scenarios, transmitter output may need adjustment based on line conditions. If output intensity varies, would DNN equalizers require retraining? If so, indicating the time required for retraining would be informative.

Version 1:

Reviewer comments:

Reviewer #1

(Remarks to the Author)

You mention an extinction ratio ER of 36 dB, which is a very high value for a Mach-Zehnder modulator and cannot be achieved without a phase shift close to π . I wonder if this value indicates the on/off ratio of the output light during modulation or the maximum on/off ratio of a Mach-Zehnder interferometer with an ideal phase shift given by the thermo-optic effect or similar. The former is impossible, considering the modulation efficiency and the length of the phase shifter. Even in the latter case, this value indicates a very well-fabricated symmetrical interferometer, which is not so easy to achieve. I think this value is just a typo or the authors misunderstood the ER. Please check again and correct it.

While I fundamentally believe this paper is worth publishing, such a simple misunderstanding reduces the credibility of this research and the statements in this paper. Please review all the content of this paper and its consistency.

Regarding the driving voltage, 5 Vpp is a large value and very power-hungry. Please comment on future prospects for reducing power consumption.

Reviewer #3

(Remarks to the Author)

The authors have addressed all of the reviewers' comments. Furthermore, they have proposed T-biGRU, which enhances the performance of the system. From the beginning, this paper demonstrated excellent results in terms of transmission characteristics, and I consider it to be an outstanding work. Therefore, it would be appropriate to submit it to a suitable

journal.

If the authors aim for publication in Nature Communications (NCOMMS), a more detailed analysis of the ANN equalizer would further enhance the paper. For example, it would be beneficial to specify the extent of variation in the Si-SLM characteristics that the ANN equalizer can compensate for, along with the range of its compensation capabilities. Additionally, the Bi-GRU NLE incorporates k preceding and k succeeding symbols in its input layer. It would be helpful to clarify how the value of k is determined as a hyperparameter and to provide some degree of physical interpretation for it. On the other hand, adding substantial new elements may cause the paper to lose focus, making its core contributions harder to convey. Thus, maintaining simplicity is advisable. Submitting to journals other than NCOMMS is also a valid option worth considering.

Reviewer #4

(Remarks to the Author)

This paper titled "Exploring 400 Gbps/λ and beyond with AI-driven silicon photonic slow-light technology" by Han et.al. presents a high-speed silicon modulator with slow-light effects together with a learned equalizer for signal correction. It demonstrates over 400 Gbps per channel transmission with PAM-4. By leveraging AI-based equalization, the system compensates for nonlinear distortions, achieving bit error rates below HD-FEC thresholds.

The approach presented can offer a scalable path for high-speed optical interconnects. However, despite the general presentation quality, the paper has a critical shortcoming. First of all, the modulator design is similar to prior work; and its performance does not surpass state-of-the-art silicon modulators. Secondly, the AI-optimized equalizer's novelty over those of existing equalization techniques is not clearly demonstrated. The paper also is missing sufficient detail on training, computational overhead, robustness, and stability of the equalizer in general. After the equalizer's novelty is clearly demonstrated through an extensive revision, the paper may then be suitable for publication in Nature Communications. Below are my detailed comments:

1. This paper really does have two separate parts presented together. A large part of the submission is dedicated to the characterization and performance of the modulator itself. However, the performance of the modulator array is fundamentally separate from the learned equalizer. From the perspective of the modulator alone, a push-pull type, MZI-based slow light modulator is quite common. Even other MZI-based or ring-based modulators can be operate faster (and also more energy efficient) than shown in this paper, with several pure-Si examples reaching above 100 Gbps (some of which are already referenced by the authors):

- Li, Miaofeng, et al. "Silicon intensity Mach–Zehnder modulator for single lane 100 Gb/s applications." *Photonics Research* 6.2 (2018): 109-116.
- Sun, Jie, et al. "A 128 Gb/s PAM4 silicon microring modulator with integrated thermo-optic resonance tuning." *Journal of Lightwave Technology* 37.1 (2018): 110-115.
- Li, Hao, et al. "A 112 Gb/s PAM4 silicon photonics transmitter with microring modulator and CMOS driver." *Journal of Lightwave Technology* 38.1 (2020): 131-138.

Given these existing results (and the identical structure in ref [39] of the paper), the presented modulator itself does not possess a great novelty in terms of its design/geometry or its performance metrics.

2. On the other hand, it can still be argued that coupling the modulator array with a learned receiver backend presents valuable advancements for a broad electronics/optics community. However, even though it is not explicitly mentioned by the authors, the learned equalizer's main goal here is to mitigate inter-symbol interference caused by various distortion mechanisms in the modulator (as well as through the other components in the transceiver pipeline). Other well-known equalizers (such as DTE or Volterra), or some form of equalization is already used quite frequently in modern high-speed communication networks. Given this fact, the authors should provide detailed analysis on how their learned equalizer compares with other equalization techniques in mitigating ISI for the slow-light modulator presented. Without this comparison, the submission does not clearly demonstrate novelty from the equalizer perspective either, at least in its current state.

3. Even more generally, the inner workings of the equalizer should be discussed in more detail. For instance, does the equalizer perform near-identical functions for wavelengths with similar insertion losses through the modulator. Do these adjacent wavelength channels experience similar levels or types of nonlinearities throughout the transceiver? If the network was only trained using a single channel, how well would it generalize to using multiple different channels through transfer learning? These questions would clarify the specific details and capabilities of the equalizer network, as its treatment as a black-box tool does not provide sufficient insight into its capabilities, limitations, or any channel-specific mechanisms.

4. I fully agree with reviewer #1's earlier comment that the title with "AI-powered silicon photonic slow light technology" or "AI-driven silicon photonic slow-light technology" (as it is currently written) does not convey the main novelty of the presented work. The slow-light modulator itself does not include any AI-driven or AI-designed elements. As reviewer #2 also noted, this updated title still creates a connection between the modulator and detection through a learned equalizer. This presentation leaves the reader with the impression that the modulator itself is comprised of AI-driven components. In my opinion, even the update to the title still does not resolve this evident potential misunderstanding.

5. The abstract states that the system operates "without additional wavelength adjustment and temperature feedback systems." However, a DC pad array is still used to control the modulator operating point via a TiN heater, which is a standard

thermal tuning mechanism in many silicon modulators. Are the authors referring to a different experimental configuration when making this claim? This requires clarification, as it could mislead readers regarding the tuning requirements of the system as well as its thermal stability.

6. Transmission is reported as a “per wavelength” basis. But the system’s capability to work without additional thermal tuning is also emphasized. Does this mean it would not be possible to use a tighter spectral spacing than 1nm used (from Fig 2g), corresponding to more than 8 wavelengths spectrally? Are there any potential routes that could enable 50GHz DWDM spacing in such a system? Does this pose any limitations on the spectral density of information that can be encoded using the presented system, or are there any other practical bottlenecks due to spectral broadening?

7. The equalizer’s workload should be reported as an overhead with all relevant time and energy metrics required for training and re-training the system, in case of configuration scenarios like those shown in Figure S10 by the authors. Particularly, it would be useful to see training curves for the learned model, how many iterations or batches were necessary for training, and the energy and time requirements for this process if it were to be performed as an initial calibration step before transmission and detection of real data.

8. Similarly, due to such strong emphasis on the capability of the learned equalizer, it would be good if the authors could provide a system performance snapshot both with and without the equalizer in place. This is also related to reviewer #3’s question on the improvement of SFDR (as well as other relevant performance metrics), even if an improved SFDR is not directly targeted by the system implemented. For instance, while clear eye diagrams are observed when the trained equalizer is used above 200 Gbps, how is the performance affected when no learned equalizer is used at all? At what speed the version of the system with no learned equalizer still achieves BERs compatible with HD-FEC? Can this potential reduction in speed (or eye clarity / BER metrics) justify the decrease in energy usage? This is particularly important as the reported power consumption metrics for the neural network components in Table S1 are larger in comparison to other parts of the presented transceiver.

9. The presented modulator still operates through the physical principle of this nonlinear plasma dispersion, which plays an important role in signal distortion. From the responses by the authors, it seems that the modulator’s 10V total swing is sufficient for about 30% of V_{π} , which is sufficient in case an equalizer is used. It would be important to quantify what changes to the equalization requirements are necessary if a smaller driving voltage is used, but at slightly lower speeds. Once again, a slight reduction in speed and driving voltage could yield a large decrease in signal distortion, rendering the equalizer unnecessary under certain circumstances. This should be tested for a more complete and comprehensive analysis of the transceiver system.

10. The explanation that “The retraining frequency depends on how fast the components degrade or how often they are replaced” does not fully convey the potential limitations of the system. From the authors’ description of the learned equalizer, it is evident that nonlinear distortions are a large part of what the equalizer compensates for. Therefore, any changes these distortions (including those induced by thermal fluctuations or physical displacements in the 100s of meters of fiber itself) would require potential recalibration. These types of changes are quite common in data center application settings. To this end, it would be crucial to quantify how robust the system is to these types of changes, and how much change in these parameters would require retraining of the equalizer. For example, does the model suffer from overfitting to any of these environmental conditions or the equipment used? In contrast, the presented hypothetical changes such as replacement of the driver, EDFA, or changing the DC bias are comparatively rare scenarios. This was also an important part of an earlier comment by reviewer #2, which was not fully addressed in this current revision. Another relevant parameter is this scenario is also the optical power, as indicated by reviewer #3 in their comment regarding output intensity. It would be good to see how much change in this power the system would be able to tolerate without needing recalibration. Currently, the only test performed by the authors is shown for a change of 2dB.

11. The reported insertion loss of the modulator is over 10 dB; and the majority of this is attributed to the modulation arms. In comparison to existing MZI-type modulators, this is relatively high and can be an important practical limitation. I suspect a large part of this is due to the free-carrier absorption loss, and some potentially due to mode conversion into and out of the Bragg waveguide. Can the authors quantify the different factors contributing to this loss? Are there any specific design modifications or fabrication improvements that could be used to reduce these losses?

12. In Fig 3e, there is a significant difference between the BERs measured for different channels, especially at lower data rates. Does this have anything to do with the wavelength dependence of the modulator response? Or are there any other reasons behind this difference?

Version 2:

Reviewer comments:

Reviewer #1

(Remarks to the Author)

In Supplementary Section 8, the bit energy consumption E_b of the modulator is discussed based on the equation in the first paragraph. However, this equation neglects the power consumption in the termination resistors. As discussed in Kawahara,

et al., Optica vol. 11, no. 9, pp. 1212-121, 2024, when the electrodes in the Mach-Zehnder modulator are terminated by impedance-matched resistors, the total power consumption of the modulator is determined by that at the resistors, regardless of the amount of charge and discharge at the junction capacitance of the phase shifters. This paper says that the electrodes were terminated by resistors to avoid the RF reflection, although its value is not written. In such a case, E_b should be estimated from that at the resistors, which might be much larger than that denoted in this supplement.

Reviewer #3

(Remarks to the Author)

I would like to express my appreciation to the authors for their thoughtful and thorough responses to the reviewer comments, including those I previously raised. It is clear that they have made significant efforts in revising the manuscript, providing additional experimental data, expanding technical explanations, and clarifying key points. These efforts are sincerely acknowledged.

The integration of an AI-based equalizer into a silicon photonic system to achieve 400 Gbps transmission is an impressive technical achievement. The direction of combining machine learning with silicon photonics is timely and has strong potential for impact in both fields.

At the same time, I still have some remaining questions, particularly regarding how deeply the manuscript explains the internal behavior of the proposed equalizer and to what extent the method could be applied more broadly across different photonic systems. While the results are clearly presented and well supported, I feel that additional discussion on the underlying mechanisms and general design principles could help further highlight the conceptual contribution of the work.

That said, I also acknowledge that some of these questions may stem from my own limited understanding of certain aspects of the system or model. If so, I appreciate the authors' patience and would welcome any additional clarification they might be able to provide.

I would be happy to see further elaboration on these points if feasible, and in the meantime, I defer to the editorial team regarding the manuscript's alignment with the scope and expectations of Nature Communications.

Reviewer #4

(Remarks to the Author)

I appreciate the effort that the authors have put in to revise their work. They have now addressed all of my comments in detail. The new supplementary details on how the implemented equalizer compares with existing methods are clear. It is also now much clearer why the implemented AI equalizer is particularly suitable for silicon slow-light modulators. The new title is more suitable for conveying the impact of the presented work. The provided quantitative details on the energy consumption, training of the equalizer, modulator loss, and driving voltage also provide better context and improve the reproducibility of the work. I now believe the work is suitable for publication at Nature Communications.

Version 3:

Reviewer comments:

Reviewer #1

(Remarks to the Author)

The revised paper still discusses the power consumption and bit energy consumption values at the p-n junction of the phase shifter, but this only misleads the reader.

The performance of the phase shifter should be evaluated only by $V_{\pi} L$, not by power consumption. An efficient phase shifter reduces V_{π} and/or shortens L . Some power is consumed by the parasitic junction resistance R_{pn} in the phase shifter, but it is negligible compared to the power consumption at the termination resistance R_L , which is given by $2 \times (V_{\pi}/2)^2/R_L$ in the case of push-pull driving. In an ideal situation, R_{pn} is almost zero, but this power consumption still remains.

Therefore, the power consumption and corresponding bit energy consumption values evaluated for the p-n junction are meaningless. Many previous papers claiming these consumption values are very misleading. This paper should stop doing that.

Reviewer #3

(Remarks to the Author)

I sincerely appreciate the authors' thoughtful and comprehensive responses to my previous comments. All of my concerns have been carefully addressed, with substantial improvements made to both the main text and supplementary materials. In particular, the enhanced explanations regarding the internal workings of the AI-based equalizer, the underlying design principles, and the potential applicability to broader photonic systems are clear and informative.

These clarifications significantly strengthen the conceptual and technical contributions of the manuscript. I believe the work is now well-suited for publication in Nature Communications.

Version 4:

Reviewer comments:

Reviewer #1

(Remarks to the Author)

Now I decided to accept the paper for publication.

Response letter

We appreciate the careful review by the reviewers and have modified the manuscript in accordance with their suggestions. Here, we present a point-by-point reply (**in black**) to the reviewers' comments (**in gray**), as well as the action taken (**in red**).

Response to the report from Referee #1

General comments: *"This paper reports a record high-speed, pure silicon Mach-Zehnder modulator with slow light phase shifters of a small footprint. Compensating for the nonlinear response using an equalizer optimized by AI technologies, it worked beyond 200 Gbaud in PAM-4 format, resulting in >400 Gbps data rate at single wavelength and single lane, showing very clean eye diagrams which satisfy BER level below HD FEC. Due to a wide working spectrum of slow light and high uniformity, it potentially achieve a Tbps class transmission using WDM. It also allows the packaging of all channels in a compact pluggable form factor, thanks to the small footprint. These results are impressive, and it is worth publishing in Nature Communications after some major revisions. My comments are listed below."*

Our reply: We appreciate the reviewer's recognition of our work. To further address the raised questions, we made the point-by-point response as follows.

Comment 1: *"A similar device has already been reported in Ref. 43, showing the EO bandwidth beyond 100 GHz. In this paper, it is also shown to be as wide as 90 GHz. I wonder if these wide bandwidths are attributed to a small RC constant. In the short slow-light phase shifter, the capacitance C is reduced, while parasitic resistance R is increased. This increase in R is reduced by reducing the distance between the PN junction and the highly doped regions, sacrificing the low optical loss. Please discuss more clearly the relation between the doping profiles, optical loss, group index of slow light and EO bandwidth."*

Our reply: Thanks for the valuable comments. Indeed, the high EO bandwidth is critical for ultrahigh-speed transmission applications of silicon modulators. For these two devices including Ref[43] above, the principles are both based on slow-light effect, but the structures and parameters are different for distinct performance orientations. Actually, the slow-light device model is a reconfigurable solution for designing modulators, which means that we can make targeted and effective changes to device performance by adjusting the design. The small RC constant is necessary for achieving high EO bandwidth. Besides, the EO bandwidth of silicon slow-light modulators (BW_{EO}) is determined together by the electrical bandwidth (BW_E) characterized by RC constant of PN junctions and metal contacts, and the optical bandwidth (BW_O) characterized by the photo lifetimes in resonance cavities, followed by:

$$\frac{1}{BW_{EO}^2} = \frac{1}{BW_E^2} + \frac{1}{BW_O^2}$$

Therefore, to improve BW_{EO} , it is essential to improve BW_E and BW_O simultaneously. While increasing bandwidth, parameters such as efficiency factor also need to be guaranteed to achieve a balance. Here, we conduct a discussion on the relationship between performance parameters such as bandwidth and efficiency factor with structural parameters including slow-light waveguide and PN junction doping profiles, to clearly demonstrate the optimization process.

For electrical design, the electrical bandwidth of the depletion-type PN junction is quantitatively calculated and simulated, which is determined by the junction capacitance C and the junction resistance

R of the PN junction. The built-in voltage V_{bi} of the PN junction is calculated as shown in equation below:

$$V_{bi} = \frac{k_B T}{q} \cdot \ln \left(\frac{N_A N_D}{n_i^2} \right)$$

in which, N_A is the acceptor doping concentration, N_D is the donor doping concentration, and n_i is the intrinsic carrier concentration. The depletion region W_D width is calculated based on the V_{bi} as:

$$W_D = \sqrt{\frac{2\varepsilon}{q} \cdot \frac{N_A + N_D}{N_A \cdot N_D} \cdot (V_{bi} + V)}$$

Therefore, the junction capacitance C_j of the PN junction is calculated as:

$$C_j = \frac{\varepsilon \cdot h_{rib}}{W_D}$$

where h_{rib} is the height of the rib waveguide of the PN junction, and ε is the dielectric constant.

For the resistance R of each region of the PN junction:

$$R_p = \frac{(W_{pn} - W_{rib})}{2qN_A\mu_h h_{slab}} + \frac{\frac{W_{rib}}{2} - \frac{N_D W_D}{N_D + N_A}}{qN_A\mu_h h_{rib}}$$

$$R_n = \frac{(W_{pn} - W_{rib})}{2qN_D\mu_e h_{slab}} + \frac{\frac{W_{rib}}{2} - \frac{N_A W_D}{N_D + N_A}}{qN_D\mu_e h_{rib}}$$

where W_{pn} is the PN junction width, W_{rib} is the rib width of the rib waveguide, and h_{slab} is the height of the slab layer. Furthermore, the resistance of PN junction is:

$$R_j = R_p + R_n$$

Therefore, the electrical bandwidth of the PN junction is:

$$BW_E = \frac{1}{2\pi R_j C_j}$$

Here, Figure R1-1 shows the variation trend of the designed PN junction characteristics including C_j and R_j with the bias voltage and doping concentration. The junction capacitance is the key factor limiting its high-frequency response. As the applied bias increases, the depletion region will gradually widen, the junction capacitance will be significantly reduced, and the junction resistance will also be reduced, which will increase the RC bandwidth. At the fabrication level, the PN junction doping concentration has an obvious influence on the modulator performance. As the PN junction doping concentration decreases, although the R_j increases to a certain extent, its C_j will be significantly reduced, and the total RC bandwidth will show an upward trend. It can be seen that low-concentration doping will be conducive to the realization of ultra-high bandwidth modulators. Excessive doping concentration will limit the RC bandwidth due to excessive junction capacitance. Therefore, in the actual modulator design, a relatively low doping level of $5.0 \times 10^{17} \text{ cm}^{-3}$ was chosen to support the realization of ultra-high bandwidth, which provides an ultra-high electrical bandwidth around 180 GHz at 3 V bias voltage.

Figure R1-1. Simulation of PN junction performance. (a) Resistance with bias and concentration. (b) Capacitance with bias and concentration. (c) Electrical bandwidth with bias and concentration.

For the efficiency factor, a higher doping concentration will lead to a larger efficiency factor, so a comprehensive consideration should be made to determine the appropriate doping concentration in a specific modulator. Although we chose a lower doping concentration in manufacturing to ensure high bandwidth, the efficiency factor can be effectively compensated and controlled by introducing a slow-light structure. At the same time, the low-loss advantage brought by low doping can improve the additional increased loss caused by the introduction of the slow-light waveguide, thereby completing the coordinated matching design of optical and electrical aspects.

Here, in the basic coupled-resonator optical waveguide (CROW) structure of the slow-light modulators, each number of period (N_p) of gratings with a lattice constant of a forms one arm of a resonator, and a finite number of cascaded resonators (N_r) are connected to construct a modulation arm (Figure S2a). In the slow-light modulator design, the phase accumulation within the length of $L = 2N_r N_p a$ is:

$$\phi = \Delta n_{eff} \cdot L_{eff} \cdot \frac{2\pi}{\lambda_0} = \eta \cdot n_g \cdot (2N_r N_p a) \cdot \frac{2\pi}{\lambda_0}$$

in which $\eta = \eta_0(\sqrt{V + V_{bi}} - \sqrt{V_{bi}})$, η_0 is related to the overlap integral of carriers and optical fields, V_{bi} is the built-in voltage of PN junction. Accordingly, we define the efficiency factor as $\phi/\pi V$ to characterize the phase change per voltage. The phase accumulation is enhanced by the slow-light effect in a factor of n_g .

For optical design, in the CROW structure, by evaluating the mid-gap mode around 1550 nm (Figure S2b), the coupling coefficient κ is calculated as:

$$\kappa = \kappa_0 e^{-2N_p a \Delta} / N_p$$

in which κ_0 is a constant determined by the unit cell structure and the gap depth Δ presented how strong the field is bound in the phase shifter region. Therefore, the group index of the passband around 1550 nm can be expressed as formula below, which shows that the group index increases exponentially with Δ and N_p , that is, the slow light effect is stronger.

$$n_g = \lambda_0 / 4\pi\kappa N_p a = (\lambda_0 / 4\pi\kappa_0 a) e^{2N_p a \Delta}$$

However, while introducing the slow-light effect to enhance the efficiency factor, it is necessary to set upper limits for n_g as well as N_p and N_r to ensure sufficient optical bandwidth. Specifically, the optical bandwidth can be expressed by the Q of the mid-gap mode (defect mode), which is,

$$BW_0 = \frac{c}{\lambda_0 Q}$$

and the Q factor is consisted by,

$$Q^{-1} = Q_{loading}^{-1} + Q_{propagating}^{-1}$$

in which $Q_{loading}$ represents the energy leakage at both ends of the slow light waveguide, while $Q_{propagating}$ represents the energy dissipated due to scattering loss and material absorption.

In the designed slow-light waveguide, the group index is chosen as 6.1. Actually, although the slow light effect is an effective way to promote efficiency factor, its side effect on limiting the optical bandwidth is usually ignored. For the ultrahigh-speed applications we focus here, if the slow-light effect is set to be over high, the insufficient optical bandwidth will reduce the EO bandwidth, which will become the primary bottleneck. Simultaneously, the group index cannot be too low because of the limitation in modulation efficiency. Here, the designed group index of $n_g=6.1$ is a carefully balanced value that balances the bandwidth and modulation efficiency for ultrahigh-speed applications.

Figure S2. Slow-light design model (a) Slow-light waveguide structure in the modulation arm. The SEM image is illustrated with actual fabricated parameters. (b) Photonic bandgap in the designed slow-light structure. The mid-gap mode is generated around 1550 nm. (c) The change of optical bandwidth and efficiency factor with number of resonators. (d) The change of optical bandwidth and efficiency factor with number of periods. (e) The influence of increasing resonator number for the phase accumulation under different voltages.

Furthermore, we calculated the optical bandwidth and efficiency factor as the function of N_p and N_r , as shown in Figure S2c and Figure S2d. It is worth noting that there is a trade-off between modulation efficiency and optical bandwidth. And given a determined N_p , the modulation efficiency linearly

increases with N_r but shows square-root dependency on the voltage (Figure S2e). In order to ensure a sufficiently high optical bandwidth for ultrahigh-speed communication and enough modulation depth for multi-level separation simultaneously, the design of N_p 20 and N_r 20 is selected here, resulting in the optical bandwidth and efficiency factor of $0.028 \pi/V$ and 95 GHz, respectively.

We hope the above analysis could make it clearer. We have also added more design details and discussions about the relationship between these parameters in Supplementary Section 2.

(Supplementary Section 2) In the modulation arms, a coupled-resonator optical waveguide (CROW) is utilized³ to generate the slow-light effect, in which a complete resonator is constructed by a $\lambda/4$ phase shifter region with broader width and several Bragg gratings on both sides with a period of around 300 nm. The structure of slow-light waveguides is demonstrated in Fig. S2a. Here, the phase shifter is in the middle position of each resonator, and a certain equal number of period (N_p) of gratings on each side (narrower Bragg gratings) construct one side beam of a resonator, then two beams are connected with each other by a $\lambda/4$ phase shifter region (broader part) to form a complete CROW resonator, along the direction of light propagation in the waveguide. Thus, through the $\lambda/4$ phase shifter region, the supercell can generate a topological mid-gap mode embedded in the photonic bandgap, between the antisymmetric and symmetric transverse electric bands, as the mode adopted for multi-wavelength communications in C-band, shown in Fig. S2b. Furthermore, a finite number of resonators (N_r) are cascaded together to construct a modulation arm. As for the physical mechanism of modulation, the modulation process is still based on the plasma dispersion effect of the depletion-mode PN junction, which is suitable for industrial applications. Also, the PN junction is based on the CROW design, illustrating a periodic structure. The part above the axis of the slow-light waveguide in Fig. S2a is P-doped ($5.0 \times 10^{17}/\text{cm}^3$), and the bottom part is N-doped ($5.0 \times 10^{17}/\text{cm}^3$). Here, the relatively low doping concentration of PN junction is selected, which is conducive to the realization of sufficiently high electrical bandwidth due to the decrease of junction capacitance. Meanwhile, the efficiency factor (define as $\varphi/\pi V$ to characterize the phase change per voltage) can be effectively compensated and controlled by introducing the slow-light waveguide structure. The zoom-in SEM image of the fabricated slow-light waveguide is also demonstrated in Fig. S2a (part of one resonator is shown here, including a $\lambda/4$ phase shifter region and six periods of Bragg gratings on both sides), with actual measured dimensions (the core and overall width of the waveguide, the period and width of the gratings, the width of the phase shifter). As a one-dimensional waveguide grating structure, silicon CROW possesses high CMOS compatibility, and all feature sizes are suitable for commercial silicon photonic foundry.

The slow-light scheme provides a high degree of design freedom, and the structure parameters can be flexibly selected according to the specific application scenarios. In the slow-light design, the selection of group index is critical to the overall performance of the device. For the ultrahigh-speed applications here, if the slow-light effect is set to be over high by adjusting the grating structure, the insufficient optical bandwidth will reduce the EO bandwidth significantly, which will become the primary bottleneck. Simultaneously, the group index cannot be too low because of the limitation in modulation efficiency. Therefore, based on our optimized waveguide grating structure, the designed group index of $n_g=6.1$ is a carefully optimal value to balance bandwidth and modulation efficiency, which is the key point for ultrahigh-speed applications in our work. Considering the group index of the typical silicon waveguide is around 4.0, the slow-light factor in the designed slow-light waveguide is 1.525. In fact, the values of N_r and N_p will directly affect the optical performance of the device, thus we can build a reconfigurable slow-light model to describe the relationships. Fig. S2c-e demonstrates the variation of optical

bandwidth, efficiency factor and phase accumulation with N_r and N_p . In general, the increase of N_r and N_p , which means the enhancement of slow-light effect, will tune up efficiency factor and Q factor but reduce optical bandwidth. Therefore, it is essential to select favorable N_p and N_r to balance the parameters according to the application scenarios. For a certain N_p value of 20, Fig. S2c shows the change of optical bandwidth and efficiency factor with N_r . When N_r increases, the optical bandwidth will decrease gradually and the efficiency factor basically linearly. In contrast, for a certain N_r value of 20, the increasing N_p will enhance the efficiency factor quite obviously, especially for the region over 30 (Fig. S2d). However, higher N_p will constrain the optical bandwidth, and this will restrict the modulator performance at ultrahigh-speed applications.

Comment 2: “The driving voltage V_{pp} is not shown in this paper. Please write it for each result. The modulation efficiency is written to be $0.82 V_{cm}$, and the phase shifter length is 249 microns, the $V_{pi} = 33 V$. If I assume $V_{pp} = 5 V$ written in Ref. 43, the phase shift is only 0.15π . If the operating is centered at the quadrature point, the nonlinearity is minimized. Actually, the IMD3 is more than 40 dB lower than the fundamental wave, as shown in Fig. 2e. In such a situation, why was the optimizing of the equalizer so effective, particularly for PAM-4?”

Our reply: Thanks for the reviewer’s suggestions and comments. We have already added the driving voltage V_{pp} values in the method for the experimental results and emphasized that the adopted differential signals work in a dual-drive push-pull mode. For differential signals, the driving voltage in research generally refers to the V_{pp} applied to each modulation arm [*Nat. Electron.*, 6(11), 910-921 (2023)], [*Optica*, 11(9), 1212-1219 (2024)], [*Opt. Exp.*, 27(10), 14321-14327 (2019)], [*Opt. Lett.*, 39(16), 4839-4842, (2014)], [*J. Light. Technol.*, 32(12), 2240-2247 (2014)]. In the tests, the differential signals are amplified by a commercial SHF single-ended driver to obtain a 5-V V_{pp} . Meanwhile, the designed modulator works under the push-pull configuration with a dual-drive architecture of GSGSG-type RF electrodes. Here, the driving voltage of differential signals for each arm is both 5-V V_{pp} , so the overall swing is 10 V. Considering the V_{π} of 33 V ($V_{\pi L}$ is $0.82 V \cdot cm$ and L is 249 μm), $\sim 30\% V_{\pi}$ is reached, which is basically enough for modulation with the equalizers. For example, in [*Optica*, 7(11), 1514-1516 (2020)], the modulation efficiency of the silicon modulator is $1.5 V \cdot cm$, considering the modulation arm of 2.5 mm, the V_{π} is about 6 V. And the overall swing is 1 V (0.5-V V_{pp} for each arm), which has reached $\sim 17\%$ of V_{π} . Meanwhile, the slow-light design model is a reconfigurable method for different application orientations. By adjusting the structure parameters and increasing the slow-light effect, the modulation efficiency can be enhanced further, thus reducing the requirements for driving voltages.

In fact, the AI equalizer is used at the back end of the system, so it is a full-link nonlinear distortion optimization for both optical and electrical devices, not only compensating for the insufficient linearity of the modulator. For the modulator nonlinearity, silicon modulators employ plasma dispersion effects by forming PN junctions. During ultrahigh-speed modulation, carriers cannot fully recover to their steady state before the next cycle, causing the accumulation of nonlinear effects. Also, the Mach-Zehnder modulator inherently exhibits a cosine transfer function, which will also introduce nonlinearities, deviating the output from a linear mapping of the input and thus degrading the modulation performance. For the system, the primary electrical nonlinear sources originate from RF components, attributed to the inherent nonlinear behavior of charge transport at the transistor junction which is predominantly observed in the transistor driver. And nonlinear effects such as amplitude-to-phase conversion, in-band and base-band memory effects in amplifiers, the constrained operation

range of the analog-to-digital converter introduced nonlinear vibrations cannot be disregarded.

These optical and electrical nonlinearities will be a problem for multi-level formats like PAM-4 and PAM-8. Compensating for these nonlinear distortions poses a challenge for effective compensation. Digital pre-distortion is one of the frequently employed methods, but pre-compensation algorithms pose complexity in implementation as they necessitate back-propagated signals to determine the nonlinear filter coefficients in the transmitters. Other channel impairments compensation methods such as Volterra equalizer and digital backpropagation encounter exorbitant computational complexity owing to the considerable number of processing steps involved and suffered imperfect performance.

Artificial neural network equalizers based on machine learning have been proposed to address the nonlinear challenges. Neural network equalizers exhibit potency in estimating parameters within a noisy environment and recognizing complex mappings with nonlinear boundaries between input and output data, it views the equalization as multi-class problems and provides significantly better performances than the conventional linear and nonlinear equalizers such as feed-forward equalizer, decision feedback equalizer and Volterra equalizer etc. By training on large datasets, neural network can better capture the underlying distribution of the optical signal and learn the subtle relationships between different symbols, this enables them to effectively handle cases where traditional equalizers might fail. Neural networks can learn to directly map the distorted signal back to the transmitted symbols without manually decomposing the problem into individual components. This simplifies the process and often leads to better performance because the network learns an optimal transformation of the received signal, considering all distortions jointly. Based on this consideration, we employ ANN algorithms to equalize the high-speed signal generated by the silicon slow-light modulators.

We hope the above discussion could clear the reviewer's concerns.

(Main part, line 572) then the differential signals are amplified by a commercial SHF single-ended driver to obtain a 5-V V_{pp} and then injected into the modulator working under the dual-drive push-pull configuration.

Comment 3: *"The authors mention that the nonlinearity of the slow light modulator is similar to that of standard Si modulators operated by carrier plasma dispersion. Does it mean that AI-assisted equalizer can improve the signal quality not only of the slow light modulator but also of other modulators? If so, this study includes two separate topics on slow light modulator and AI-assisted equalizer, and the sentence "AI-powered silicon photonic slow light technology" in the title is misleading."*

Our reply: Thanks for the comments. We agree with the reviewer that the AI equalizer can improve the signal quality not only for the slow-light modulators but also for other modulators. However, there is a strong match and correlation between the two aspects, and this is also the starting point of our research. Sorry for not explaining it clearly in the initial manuscript. The AI equalizer is particularly suitable for silicon slow-light modulators for the following reasons:

Nonlinear distortions: Improving the indicators of modulators, especially bandwidth and modulation efficiency, is a long-term goal in this field, which is also the device-level guarantee for ultrahigh-speed transmission. By utilizing the slow-light approach, the bandwidth and modulation efficiency can be improved. However, the insufficient linearity is not enhanced, which becomes the bottleneck for high-order transmission, and the lack of linearity brought by silicon material can be compensated by AI equalization exactly. For conventional silicon modulators, in the process of ultrahigh-speed transmission,

low EO bandwidth is the primary limitation, thus it is no longer important whether nonlinear equalization is used or not. High bandwidth is the basis for achieving ultrahigh-speed transmission, which is the advantage of slow-light modulators compared to conventional silicon modulators. The adoption of AI equalizer for nonlinear distortion is based on the situation that slow-light modulator can support ultrahigh-speed transmission with high-bandwidth performance.

Fabrication deviations: At present, the main devices used in ultrahigh-speed transmission are lithium niobate thin film modulators. The electro-optic coefficient of lithium niobate can be basically kept consistent, which makes the modulator uniform. However, pure silicon modulation is based on the plasma dispersion effect, and it is difficult to achieve complete consistency in the doping process. Also, the introduction of slow-light effect here is based on the grating waveguide structure. Compared to the two-dimensional photonic crystal, although the silicon waveguide grating structure has greatly improved the process stability with CMOS compatibility, the unavoidable structure fluctuations of gratings still exist due to the fabrication limits. After introducing AI enhancement, the differences between grating devices can be minimized. Therefore, through the AI equalization at the back end, the process limitations of the slow-light structure can be eliminated on the same wafer.

System design: Based on the points above, the AI equalizer is especially suitable for doped-silicon slow-light modulators for ultrahigh-speed applications. Therefore, the differences between doped-silicon slow-light modulators can be eliminated by AI equalization, making the deployment of large-scale slow-light modulator array possible. The modulator array and the WDM system can maximize the performance advantages by introducing the AI approach.

Sorry for not clarifying the matching between the silicon slow-light modulator and the AI equalizer clearly in the initial manuscript. Since the adaptive characteristics of AI equalizers help mitigate the impact of fabrication sensitivity and nonlinearity in the slow light structure, this approach further enhances the performance of the slow light technology, which is currently the only viable method to achieve bandwidths greater than 100 GHz in silicon photonics. We have already supplemented the key points above in the main part. Meanwhile, to avoid potential misunderstanding, we have changed the title to “Exploring 400 Gbps/ λ and beyond with **AI-driven** silicon photonic slow-light technology”

(Main part, line 86) Therefore, by using an ANN equalizer, the nonlinear distortion of Si-SLMs can be reduced. At the manufacturing level, achieving complete uniformity remains challenging in the doping process, while the PN junction is the modulation basis for Si-SLMs^{8,9}. Simultaneously, although one-dimensional waveguide grating structure has improved the process stability compared to two-dimensional photonic crystals^{40,42}, the structural fluctuations of doped silicon gratings still exist due to the fabrication limitations. Therefore, Si-SLMs require an adaptive technical approach at the back end to eliminate the variations caused by process deviations, thus making the deployment of large-scale high-density Si-SLM systems possible.

(Main part, line 242) Meanwhile, the relatively high passband conformity between devices can demonstrate that the fabricated Si-SLMs have a favorable process consistency in general. However, from the spectral results it can be seen that a wavelength shift still occurs, due to the unavoidable fabrication deviations. Although the manufacturing uniformity of Si-SLMs has been improved by introducing one-dimensional CROW compared to two-dimensional photonic crystals, the performance fluctuations between doped silicon grating structures still exist. Therefore, for large-scale Si-SLM systems, the AI equalization at the back end is necessary to minimize device performance fluctuations caused by

process errors.

(Main part, line 506) At the manufacturing level, since silicon material does not possess a high and consistent EO coefficient, forming PN junction is the basis for realizing pure silicon modulation, while achieving complete uniformity in doping process is still challenging. Simultaneously, although the CMOS-compatible waveguide grating structure has already improved process stability compared to photonic crystals, the structural fluctuations of doped silicon gratings still exist due to the unavoidable fabrication deviations. Therefore, through AI equalization at the back end, the variations between Si-SLMs caused by process errors can be eliminated, thereby making the deployment of large-scale transmission systems possible.

Comment 4: *“In the acquisition of eye diagrams, the electrical signal was amplified using SHF S807C whose electrical bandwidth was limited to 55 GHz. I wonder if the optimized equalizer compensated for the nonlinearity of these electrical devices used.”*

Our reply: Thanks for the reviewer’s comments. For the electrical amplifiers, restricted by experimental conditions, the electrical amplifiers we used in the experiments are commercial SHF single-ended drivers, including SHF S807C and SHF S804B (the experiments for driver replacement as well as optical amplifiers are also conducted for evaluating the retraining frequency of the neural network in Supplementary Section 5). The high frequency 3-dB point is around 55-60 GHz, and the gain is around 22-23 dB [<https://www.shf-communication.com/products/rf-broadband-amplifiers/>]. Nevertheless, for higher frequency, the electrical amplifiers are still functional and can work in practical scenarios, while some degree of waveform degradation will occur indeed at ultra-high frequency region. In the experiments, the optimized AI equalizer is adopted at the back end of the system, so this is a full-link nonlinear distortion compensation for both the optical and electrical devices used in the system. For the nonlinear signal distortions at the system level, the electrical nonlinear sources originate from RF components due to the inherent nonlinear behavior of charge transport at the transistor junction, which is predominantly observed in the drivers. Also, the utilization of EDFA could introduce nonlinear Kerr impairments to the optical link. In the experiments, we adopt the AI equalizer compensation to process the modulated signal after it is output from the modulator and received by the oscilloscope. Therefore, this is a full-link compensation method for nonlinear distortions, including the nonlinearity of all the optical and electrical components we used in the system.

We hope the above analysis could make it clearer.

Comment 5: *“Considering the small phase shift noted in Comment 2, the extinction ratio must be small, although it is not shown in this paper. Please show this and give some discussion.”*

Our reply: Thanks for the comments. We have extracted the values of the extinction ratio (ER) from the transmission result of the silicon slow-light modulator by tuning the applied voltage. An ER of 36 dB is obtained, which is relatively small compared to conventional long silicon Mach-Zehnder modulators (with 2-3 mm modulation arms) indeed [*Adv. Photon.*, 3(2), 024003 (2021)], due to the ultra-compact footprint. However, compared to silicon micro-ring modulators, the ERs of which are generally from 20-30 dB or even smaller [*Nat. Commun.*, 15(1), 918 (2024)], [*Opt. Lett.* 48(4), 1036-1039 (2023)], [*J. Light. Technol.*, 37(1), 110 - 115 (2019)], [*Photon. Res.*, 10(4), 1127 - 1133 (2022)], the designed silicon slow-light modulator possesses a sufficient ER for achieving enough modulation depth. In fact, during the design process, considering the issue of insufficient ER for small-footprint modulators, we chose to

embed the slow-light waveguide structure into the Mach-Zehnder interferometer (MZI) and adopt a dual-drive push-pull working mode. The solution of embedding the resonant cavity into MZI architecture can leverage the advantage of MZI to maintain a basically sufficient ER while reducing the device footprint. We have already added the ER value in the main part.

(Main part, line 203) Moreover, by embedding CROW into the modulation arms, the advantages of MZI can be leveraged to maintain a sufficient extinction ratio (ER) while reducing the device footprint. The ER of the compact modulator is measured to be 36 dB, provided by the thermal-insensitive resonator-assisted MZI architecture.

Comment 6: “The working spectral width is shown to be 7 nm at 1550 nm wavelengths, which corresponds to a frequency range of 0.87 THz. It is too narrow for 3.2 Tbps transmission using 8 wavelengths WDM, isn't it?”

Our reply: Thanks for the question. The signal frequency broadening is an important factor for IM/DD signal transmission in WDM systems. In high-speed signal transmission, the signal broadening on spectrum for one side is as follows:

$$\Delta\nu = R_s \times \frac{1}{2} \times (1 + \alpha)$$

in which R_s is the baud rate of the signal, α is the roll-off factor (0.1 in the experiment), $\Delta\nu$ is the signal broadening on one side, and the total width of the signal is $2\Delta\nu$.

For the operating wavelength range, the passband of a single device is around 7 nm, but for an 8-channel modulator array, the appropriate slight wavelength shift can extend the available passband to 10 nm on one chip, as shown in Fig. 2(g). Therefore, the wavelength interval can be further expanded based on the 10-nm passband (The spacing between two on-chip adjacent channels can be expanded to 1.25 nm).

For WDM transmission experiments of DNN, the signal rate is up to 224 Gbps PAM-4 (112 Gbaud), the total frequency width for one channel is 123 GHz, and the frequency width for 200 Gbps PAM-4 is 110 GHz. The wavelength interval between two adjacent channels is 1 nm, which corresponds to 125 GHz frequency range, is basically enough.

However, for 400 Gbps (200 Gbaud) PAM-4, the total frequency width for one channel is 220 GHz. In the practical WDM experiment of GRU, considering the crosstalk caused by signal broadening, we select channel 1,3,5,7 (odd number) as one WDM transmission path, while channel 2,4,6,8 (even number) as the other WDM transmission path. After WDM of each 4 channels, the two WDM paths are then transmitted in parallel to reach 3.2 Tbps capacity, which is similar to the extensively deployed transceiver module scheme (2×FR4) [OFC, M4H.1, (2022)]. At present, the wavelength interval between two adjacent channels is 1 nm. Therefore, for the two on-chip channels separated by one (e.g. Ch2 and Ch4), the wavelength interval is 2 nm, and the corresponding frequency range is 250 GHz, which is larger than the total signal frequency width 220 GHz of one channel. To explain the practical WDM experiment more clearly, we have added a detailed description of the experimental system in the main part.

(Main part, line 377) Here, our goal is to increase the single-lane data rate of Si-SLMs significantly by the AI approach. However, the frequency broadening brought by IM/DD signals cannot be ignored at ultrahigh-speed data rates. Therefore, in the practical GRU WDM transmission experiment, we select on-chip channel Ch1, Ch3, Ch5, Ch7 (odd channels) as one WDM transmission path and Ch2, Ch4, Ch6,

Ch8 (even channels) as the other WDM transmission path to reduce modulation crosstalk by increasing the spectral separation between channels. After assembling each group of on-chip signals through WDM, the two paths are then transmitted in parallel to reach a total capacity of 3.2 Tbps, similar to the solution adopted by commercial optical module manufacturers⁷¹.

Comment 7: *“In addition, the paper seems to me long and very redundant. Please simplify well-known backgrounds, and remove repeated arguments.”*

Our reply: Thanks for the suggestions. We have already significantly streamlined the paper, simplified the well-known backgrounds and removed the repeated arguments to make it easier for readers to understand and get the main points. Meanwhile, we have made sufficient improvements to the manuscript, optimized figure design and full calibration of the manuscript text.

Response to the report from Referee #2

General comments: *"The submission represents an interesting improvement over existing similar technologies, and it would be well suited to be published in a technically oriented journal. Papers published by Nature Communications "aim to represent important advances of significance to specialists within each field", and its article format requires "a novel and important research study of high quality and of interest to that specific research community".*

On the other hand, the Journal of Lightwave Technology, for example, invites "experimental results which advance the technological base of guided-wave technology". While the scopes of these journals have some overlap, the present submission advances well-known technologies incrementally, while it lacks groundbreaking results.

Therefore, I cannot recommend acceptance of the submission for publication, even after an urgently required major revision.

However, I want to list a few comments which could guide the authors toward an improved version of their report.

The submission consists of two separate parts which could stand for themselves. Certainly, the experiments combine both parts by employing a slow-light modulator (A) with AI-supported reception (B). The title tries to forge a connection between these basically unrelated parts and suggests that the slow-light technology is achieved by artificial intelligence - which is not the case. In the following, I shall comment on both parts (A and B) separately."

Our reply: Thanks for the reviewer's comments. Sorry for not explaining the innovation of our work fully and not describing the relationship between slow-light modulators and AI equalizers clearly in the initial manuscript. Below we summarize several important innovations of our research:

High adaptability between slow-light modulators and AI equalizers: There exists a high correlation between the slow-light modulators and AI equalizers, and this is also the starting point of our research. By utilizing the slow-light approach, the bandwidth and modulation efficiency can be improved simultaneously. However, the insufficient linearity is not enhanced, and this can be compensated by AI equalization exactly. For conventional silicon modulators, low EO bandwidth is the primary bottleneck in ultrahigh-speed applications, and the adoption of AI equalizer for nonlinear distortion is based on the circumstance that slow-light modulators can support ultrahigh-speed signal transmission with high bandwidth. Meanwhile, pure silicon modulation is based on the plasma dispersion effect, and it is difficult to achieve complete consistency in the doping process. Simultaneously, although the silicon waveguide grating structure has improved the process stability compared to two-dimensional photonic crystals, the unavoidable structure fluctuations of gratings still exist. Therefore, silicon slow-light modulators require an adaptive technical approach to minimize the difference caused by process errors. Through AI equalization at the back end, the process limitations of the doped-silicon slow-light structure can be eliminated on the same wafer, making the deployment of large-scale slow-light modulator array possible. Since the adaptive characteristics of AI equalizers help mitigate the impact of fabrication sensitivity and nonlinearity in the slow light structure, this approach further enhances the performance of the slow light technology, which is currently the only viable method to achieve bandwidths greater than 100 GHz in silicon photonics.

Implementation of 400 Gbps per lane: Recently, for ultrahigh-speed transmission around 400 Gbps, the mainstream technical solution is heterogeneous integration technology. Although the adoption of heterogeneous integration requires a substantial update of the fabrication process (which is very costly), driven by the demand for technologies such as AI, academia and industry regard it as a key research direction. However, limited by the plasma dispersion effect of silicon, whether silicon modulators can achieve 400 Gbps signal transmission will affect their future development path and deployment. Since the advantage of silicon photonics is not high speed (but CMOS process compatibility), there is relatively little discussion or prospect of silicon modulators with transmission rates close to 400 Gbps. In contrast, there is a lot of research on high-speed heterogeneous integrated modulators nowadays. Therefore, based on the standard silicon photonic platform, it is important to conduct various optimizations to increase the transmission rate of pure silicon modulators to 400 Gbps for the next-generation standards. We believe the successful implementation of single-channel 400 Gbps with pure silicon modulator will greatly boost the industry's confidence in silicon photonics.

We have supplemented the discussion points above in the main part of the manuscript. In the meanwhile, to avoid potential misunderstanding, we have changed the title to “Exploring 400 Gbps/ λ and beyond with **AI-driven** silicon photonic slow-light technology”.

(Main part, line 86) Therefore, by using an ANN equalizer, the nonlinear distortion of Si-SLMs can be reduced. At the manufacturing level, achieving complete uniformity remains challenging in the doping process, while the PN junction is the modulation basis for Si-SLMs^{8,9}. Simultaneously, although one-dimensional waveguide grating structure has improved the process stability compared to two-dimensional photonic crystals^{40,42}, the structural fluctuations of doped silicon gratings still exist due to the fabrication limitations. Therefore, Si-SLMs require an adaptive technical approach at the back end to eliminate the variations caused by process deviations, thus making the deployment of large-scale high-density Si-SLM systems possible.

(Main part, line 242) Meanwhile, the relatively high passband conformity between devices can demonstrate that the fabricated Si-SLMs have a favorable process consistency in general. However, from the spectral results it can be seen that a wavelength shift still occurs, due to the unavoidable fabrication deviations. Although the manufacturing uniformity of Si-SLMs has been improved by introducing one-dimensional CROW compared to two-dimensional photonic crystals, the performance fluctuations between doped silicon grating structures still exist. Therefore, for large-scale Si-SLM systems, the AI equalization at the back end is necessary to minimize device performance fluctuations caused by process errors.

(Main part, line 506) At the manufacturing level, since silicon material does not possess a high and consistent EO coefficient, forming PN junction is the basis for realizing pure silicon modulation, while achieving complete uniformity in doping process is still challenging. Simultaneously, although the CMOS-compatible waveguide grating structure has already improved process stability compared to photonic crystals, the structural fluctuations of doped silicon gratings still exist due to the unavoidable fabrication deviations. Therefore, through AI equalization at the back end, the variations between Si-SLMs caused by process errors can be eliminated, thereby making the deployment of large-scale transmission systems possible.

To further address the raised questions, we made the point-by-point response as follows.

Comment 1: *“A1. The paper closest in topic to the present submission is (let aside the segmentation for avoiding a DAC):*

[39] O. Jafari, S. Zhalehpour, W. Shi, S. LaRoche: DAC-Less PAM-4 Slow-Light Silicon Photonic Modulator Providing High Efficiency and Stability. JLT 39 (2021) 5074-5082. doi:10.1109/JLT.2021.3083140

This paper is an excellent example of the way results should be presented: The basic physical mechanisms are clearly explained, measurement setups are elaborated, experimental results are well discussed, and the achievements are put in context with the state of the art.

A2. Reference [39] discusses exactly the same modulator structure as in the submission. Therefore I do not see the novelty. The improvements are incremental.”

Our reply: Thanks for the valuable comment. We fully agree that this paper is a distinguished representative work in the field of slow-light modulators, especially for the waveguide grating structure. High-speed PAM-4 signals up to 90 Gbps are realized based on the segmented slow-light modulators and all the analysis is complete including the physical principals, measurement systems and experimental results. Therefore, through in-depth study of this article, we further optimized the expression and analysis of the manuscript in detail to make the content more comprehensive.

For the novelty and improvement in our design, we summarized the innovation points as follows:

Firstly, the proper selection of slow-light effect is critical to device performance. In previous research, although the slow-light effect is widely considered to be an effective method to improve modulation efficiency, its side effect on optical bandwidth is usually ignored. If the slow-light effect is set to be too strong, the limited optical bandwidth will reduce the EO bandwidth significantly (as well as the narrow optical passband), which will become the primary bottleneck in ultrahigh-speed applications. On the other hand, the group index cannot be too low because of the limitation of practical driving ability. Therefore, based on our optimized waveguide grating structure, the designed moderate group index $n_g=6.1$ is actually the optimal choice to balance bandwidth and modulation efficiency.

Secondly, on the level of device architecture, the modulator is designed to work in push-pull drive mode, using RF electrodes with GSGSG structure in dual-drive operation. In the slow-light design, it is necessary to shorten the device length. The bandwidth can be increased by shortening the electrode length, and excessive length will also increase the insertion loss. Therefore, considering the phase accumulation requirements in the meanwhile, it is important to maximize the phase shift within a limited footprint by adopting the dual-drive push-pull drive architecture.

Thirdly, for design concept, the slow-light design and AI equalization are highly applicable, which is the premise support in our structure design. By AI equalization, the limited linearity brought by silicon material can be compensated. For the performance orientation (determined by slow-light structure parameters), considering that the AI equalizer will improve signal quality, we prioritize ultra-high bandwidth to ensure that the signal is transmitted with minimal loss, which provides the guarantee for AI decoding. Meanwhile, after introducing AI enhancement, the differences between devices due to fabrication fluctuations can be eliminated through the AI equalization, making the deployment of large-scale slow-light modulator array possible.

We hope the reasons above can clear the reviewer's concerns.

Comment 2: *"As an aside: The design of a slow-light 100Gbit/s modulator with an optical bandwidth of 1THz was published as early as 2008: Brosi, J.-M.; Koos, Ch.; Andreani, L. C.; Waldow, M.; Leuthold, J.; Freude, W.: High-speed low-voltage electro-optic modulator with a polymer-infiltrated silicon photonic crystal waveguide, Opt. Express 16(2008) 4177 - 4191. doi:10.1364/OE.16.004177"*

Our reply: Thanks for the comment and suggestion. We have read and analyzed this article in detail. This research is an excellent example, especially for silicon-based polymer modulators, and provides a remarkable reference in slow-light field and we have already cited this paper in main part as Ref[59].

However, our work still has many different points compared to this work. First, the material platform and modulation mechanisms are different. Compared with the pure silicon platform (exploiting the plasmon dispersion effect of silicon) that can be applied in large-scale industries, the polymer platform (with excellent electro-optical effect) is a natural platform suitable for high-speed modulation. However, the stability of polymers still needs to be explored. On the other hand, a two-dimensional photonic crystal structure is adopted, which is conducive to achieving a larger optical bandwidth. With carefully shifted lattices, these designs can broaden the range of the band edge with proper group index. Here, we chose one-dimensional CROW as the slow-light scheme because it is highly CMOS-compatible and can be fabricated in the standard silicon photonic process.

Meanwhile, in order to more clearly present the research in this field, we have also added several early explorations of photonic crystal modulators similar to this paper, such as Ref[60] [*Appl. Phys. Lett.*, 87(22), 221105 (2005)] and Ref[61] [*Appl. Phys. Lett.*, 97(9), 093304 (2010)]. These works prove the feasibility of slow-light modulation using photonic crystals. We thank these researchers for their early exploration of the field of photonic crystal modulation.

Comment 3: *"The claim "Our approach breaks the transmission limits of silicon photonics" cannot be interpreted as a "first" in view of [39]. However, this is suggested by the quoted sentence."*

Our reply: Thanks for the comment. Sorry for the misunderstanding, our original intention is to describe that this approach can increase the IM/DD transmission rate of silicon modulators to 400 Gbps. We have already deleted this claim "breaks the transmission limits" in the whole manuscript.

Comment 4: *"In contrast to [39] the submission lacks an explanation of the structural design."*

Our reply: We further studied Ref[39] and added more structural design and descriptions in our manuscript. For the explanation of the structural design, we have made detailed supplements on the following two aspects:

Overall architecture of the device: For the overall device design, the compact silicon slow-light modulator is under the MZI structure, with a dual-drive architecture of GSGSG-type RF electrodes. We have further improved the description of the device architecture in the main part as follows:

(Main part, line 138) For the single device, the architecture and morphology of one Si-SLM can be seen in Fig. 2b. Under a compact Mach-Zehnder (MZ) interferometer structure, the modulation arms can be shrunk to only 249 μm due to the slow-light effect, which is an order of magnitude shorter than conventional Si-MZMs, thus significantly enhancing system integration density. Simultaneously, to

enlarge the phase change, also for the better photoelectric integration and low chirp, a dual-drive architecture of GSGSG-type radio-frequency (RF) electrodes is adopted here, which enables the modulator to operate under the push-pull driving mode. At the remote end of the RF electrodes, on-chip resistors are integrated to reduce microwave reflection. Near the waveguide output port, TiN heaters are adopted on the waveguides to control the operating point of the modulator. (See Supplementary Section 1 for more details)

Structure and design of the slow-light waveguide: For the description of slow-light structure, we added a lot of description of the CROW structure in the main part and Supplementary Section 2 and optimized the layout of Figure S2a. This part of the content is related to the following Comments 8-10 below. Therefore, we have listed the revised text of the manuscript in the response to Comment 8.

Comment 5: "Fig.1a is unclear. What is meant with "shadow areas"? Covered areas with an unrelated design? Then say so!"

Our reply: Thanks for the suggestions. (Figure 2 describes the chip design and device structure. We suppose that the comment refers to Fig.2, and the following comments are the same) For the chip optical micrograph in Fig.2(a), we have made arrangement optimizations to improve the expressiveness of the images. The "shadow areas" here means the covered area that is not relevant to the design indeed. We have already changed the term to "unrelated area" to make the description clearer. The new Fig.2(a) together with Fig.2(b) is shown in Comment 6 below.

Comment 6: "The inset "Photonics chip photograph" in Fig.1 "(a) Optical micrograph of the Si-SLM transmitter chip, with the photograph of the overall chip as the inset." is irritating."

Our reply: Sorry for the misunderstanding due to the picture distribution. We have already removed the overall chip photograph and now Fig.2(a) is only the optical micrograph of the transmitter chip.

Figure 2. Design and characterization of the Si-SLM chip (a) Optical micrograph of the Si-SLM chip. (b) Device morphology of the fabricated Si-SLM.

Comment 7: "The explanation Fig.1 "(b) Device morphology of the fabricated Si-SLM. A dual-drive architecture of GSGSG-type RF electrode under the Mach-Zehnder interferometer is adopted. The modulation arms are only 249 μm long and the SEM image of the fabricated slow-light waveguide is shown as the inset below." does not help in understanding."

Our reply: To better describe the device structure comprehensively, we have substantially revised the

explanation of Fig. 2(b) as follows:

(Figure 2b) Device morphology of the fabricated Si-SLM. The modulator is under a Mach-Zehnder interferometer structure, with the compact modulation arm of only 249 μm . A dual-drive architecture of GSGSG-type RF electrodes is adopted, which enables the modulator to operate under the push-pull driving mode to enlarge the phase change. The SEM image of the fabricated slow-light waveguide is shown as the inset below. The waveguides are fabricated under a standard silicon photonic process and all feature sizes are suitable for the commercial silicon photonic foundry.

Comment 8: *"Neither does the main text provide a clear explanation (line 150-156): "Based on 151 the coupled-resonator optical waveguide (CROW), the depletion-mode PN junction has a periodic structure and the SEM image of the fabricated slow-light waveguide is illustrated below the device photograph, in which a complete resonator is constructed by a phase shifter and Bragg gratings with a period of around 300 nm on both sides. In our reconfigurable slow-light model, several resonators are cascaded to construct the modulation arm, and the parameters can be selected flexibly, according to specific application scenarios."*

Our reply: We have enhanced the description of the slow-light structure in the main part and rewritten this section fully. Meanwhile, to display the slow-light structure more comprehensively, we added a lot of structural and design details in Supplementary Section 2, and also moved the CROW structure diagram as well as SEM image with actual manufacturing parameters from Figure S1c to new Figure S2a. We have also added the photonic bandgap as the new Figure S2b.

(Main part, line 148) In the modulation arms, the slow-light effect is generated by the coupled-resonator optical waveguide (CROW), which is a one-dimensional waveguide grating structure. The SEM image of the fabricated slow-light waveguide is illustrated below the device photograph in Fig. 2b. Here, a complete resonator is constructed by a $\lambda/4$ phase shifter region with broader width in the middle position and an equal number of Bragg gratings on both sides with a period of around 300 nm, along the direction of light propagation in the waveguide. Each certain number of gratings construct one side beam of a resonator, and the two beams are separated by the phase shifter from their adjacent one. Therefore, the supercell is created through the phase shifter region, which leads to a mid-gap mode embedded in the bandgap. In our reconfigurable slow-light model, a finite number of resonators are cascaded to construct the modulation arm, and the structure parameters can be selected flexibly, according to specific application scenarios. Based on the grating waveguide configuration, the PN junction with a periodic structure is formed, which will work in the depletion mode under reverse bias voltage.

(Supplementary Section 2) In the modulation arms, a coupled-resonator optical waveguide (CROW) is utilized³ to generate the slow-light effect, in which a complete resonator is constructed by a $\lambda/4$ phase shifter region with broader width and several Bragg gratings on both sides with a period of around 300 nm. The structure of slow-light waveguides is demonstrated in Fig. S2a. Here, the phase shifter is in the middle position of each resonator, and a certain equal number of period (N_p) of gratings on each side (narrower Bragg gratings) construct one side beam of a resonator, then two beams are connected with each other by a $\lambda/4$ phase shifter region (broader part) to form a complete CROW resonator, along the direction of light propagation in the waveguide. Thus, through the $\lambda/4$ phase shifter region, the supercell can generate a topological mid-gap mode embedded in the photonic bandgap, between the

antisymmetric and symmetric transverse electric bands, as the mode adopted for multi-wavelength communications in C-band, shown in Fig. S2b. Furthermore, a finite number of resonators (N_r) are cascaded together to construct a modulation arm. As for the physical mechanism of modulation, the modulation process is still based on the plasma dispersion effect of the depletion-mode PN junction, which is suitable for industrial applications. Also, the PN junction is based on the CROW design, illustrating a periodic structure. The part above the axis of the slow-light waveguide in Fig. S2a is P-doped ($5.0 \times 10^{17}/\text{cm}^3$), and the bottom part is N-doped ($5.0 \times 10^{17}/\text{cm}^3$). Here, the relatively low doping concentration of PN junction is selected, which is conducive to the realization of sufficiently high electrical bandwidth due to the decrease of junction capacitance. Meanwhile, the efficiency factor (define as $\phi/\pi V$ to characterize the phase change per voltage) can be effectively compensated and controlled by introducing the slow-light waveguide structure. The zoom-in SEM image of the fabricated slow-light waveguide is also demonstrated in Fig. S2a (part of one resonator is shown here, including a $\lambda/4$ phase shifter region and six periods of Bragg gratings on both sides), with actual measured dimensions (the core and overall width of the waveguide, the period and width of the gratings, the width of the phase shifter). As a one-dimensional waveguide grating structure, silicon CROW possesses high CMOS compatibility, and all feature sizes are suitable for commercial silicon photonic foundry.

The slow-light scheme provides a high degree of design freedom, and the structure parameters can be flexibly selected according to the specific application scenarios. In the slow-light design, the selection of group index is critical to the overall performance of the device. For the ultrahigh-speed applications here, if the slow-light effect is set to be over high by adjusting the grating structure, the insufficient optical bandwidth will reduce the EO bandwidth significantly, which will become the primary bottleneck. Simultaneously, the group index cannot be too low because of the limitation in modulation efficiency. Therefore, based on our optimized waveguide grating structure, the designed group index of $n_g=6.1$ is a carefully optimal value to balance bandwidth and modulation efficiency, which is the key point for ultrahigh-speed applications in our work. Considering the group index of the typical silicon waveguide is around 4.0, the slow-light factor in the designed slow-light waveguide is 1.525. In fact, the values of N_r and N_p will directly affect the optical performance of the device, thus we can build a reconfigurable slow-light model to describe the relationships. Fig. S2c-e demonstrates the variation of optical bandwidth, efficiency factor and phase accumulation with N_r and N_p . In general, the increase of N_r and N_p , which means the enhancement of slow-light effect, will tune up efficiency factor and Q factor but reduce optical bandwidth. Therefore, it is essential to select favorable N_p and N_r to balance the parameters according to the application scenarios. For a certain N_p value of 20, Fig. S2c shows the change of optical bandwidth and efficiency factor with N_r . When N_r increases, the optical bandwidth will decrease gradually and the efficiency factor basically linearly. In contrast, for a certain N_r value of 20, the increasing N_p will enhance the efficiency factor quite obviously, especially for the region over 30 (Fig. S2d). However, higher N_p will constrain the optical bandwidth, and this will restrict the modulator performance at ultrahigh-speed applications.

Figure S2. Slow-light design model (a) Slow-light waveguide structure in the modulation arm. The SEM image is illustrated with actual fabricated parameters. (b) Photonic bandgap in the designed slow-light structure. The mid-gap mode is generated around 1550 nm.

Comment 9: *“Is the phase shifter only in the centre, or are not the Bragg gratings part of the junction structure, too?”*

Our reply: The phase shifter ($\lambda/4$ phase shifter region) here refers to the defect structure with wider width in the middle position of each CROW resonator. Each resonator has a phase shifter at its own center (in other words, the number of phase shifters is the same as the number of resonators). The structure of modulation arms composed by resonators with more details are shown in Fig. S2(a). In each resonator, a phase shifter (for the generation of mid-gap mode, which is also called defect mode) and an equal number of Bragg grating on both sides (along the light propagation direction) are connected to construct one resonant cavity. For the PN junction area, the entire resonator structure is implanted with P-type or N-type doping (all the Bragg grating part is the PN junction structure) and several resonators are cascaded together to form a modulation arm.

At the same time, in order to avoid potential misunderstandings, we hereby further clarify: the term "**phase shifter**" in the whole manuscript refers to the defect structure (for the generation of mid-gap mode) in the middle position of each CROW resonator, which is actually a $\lambda/4$ phase shifter region; and the phase shift arms in the MZ interferometer structure for achieving phase modulation is referred to as the term "**modulation arms**" in the whole manuscript. These are two different concepts and structures. To ensure that there is no ambiguity, we emphasize the first occurrence of phase shifter as " **$\lambda/4$ phase shifter region**" in the main part and explain its function in the meanwhile.

We hope the above statements could make it clearer.

(Main part, line 151) a complete resonator is constructed by a $\lambda/4$ phase shifter region with broader width in the middle position and an equal number of Bragg gratings

Comment 10: *“Is only the broader part between the mirrors regarded as a resonator? I would understand that a resonator consist of two mirrors and a light propagation region in-between.”*

Our reply: In the CROW design, a certain number of period (N_p) of gratings (narrower Bragg gratings) construct one beam of a resonator, and two beams are connected with each other by a $\lambda/4$ phase shifter region (broader part) to form a complete CROW resonator (Fig.S2(a)). Furthermore, a finite number of resonators (N_r) are cascaded together to construct a modulation arm, and the structure parameters can be selected according to the application requirements (Supplementary Section 2). In other words, one

modulation arm has N_r resonators, one resonator has one phase shifter region together with two side beams, and one beam has N_p gratings. As mentioned in Comment 8 and 9, we have added a lot of detailed descriptions about the CROW structure in the main text as well as Supplementary Section 2.

Comment 11: "On chip resistors (which value?) can only be guessed, but not seen in Fig.1b."

Our reply: Thanks for the comments. The on-chip resistors at the remote end of electrodes are used to achieve better impedance matching and made of TiN material. The theoretical value of the designed resistor is 50Ω . For the specific on-chip resistor measurements, we have drawn a resistor structure on the layout specifically for testing the on-chip resistors accurately, as shown in Figure R2-1 below.

Figure R2-1. On-chip resistors for measuring. Design layout and optical micrograph.

Then, we tested the resistance value under different voltages, shown in Figure R2-2. From the measured results, it can be seen that as the voltage increases, the resistance value also increases slightly. For the 3-V bias voltage we mainly adopted in the practical experiments, the on-chip resistor is 56Ω .

Figure R2-2. Resistance values of on-chip resistors under different voltages.

The experimental value is basically consistent with the theoretical design. To clarify this point, we have modified Figure S1b and drawn the on-chip resistors to the device schematic. For the specific resistance value, we added the measurement results 56Ω of on-chip resistors into the Supplementary Section 1.

(Supplementary Section 1) Meanwhile, on-chip matching resistors are integrated at the remote end of RF electrodes, with a measured resistance of 56Ω , to achieve better impedance matching and further reduce microwave reflection.

Comment 12: "What is the meaning of the RHS of Fig.1b labeled "TiN heater"?"

Our reply: In the photograph of Fig. 2(b), the overall structure of the device is shown, including the main modulation structure and DC control area. The "TiN heater" on the RHS of the device is used to control the DC operating point of the modulator, which is a commonly used component in silicon modulators. By applying a voltage on the TiN heater (the corresponding array DC electrodes are illustrated in the

right side of Fig. 2(a)), the basic refractive index of silicon waveguides will be adjusted by the DC voltage due to the thermo-optical effect of silicon. Therefore, the operating point of the silicon modulators can be controlled effectively to be centered at the quadrature point. To clarify this point, we have added the function of TiN heater and its adjustment method to the main part and Supplementary Section 1.

(Main part, line 137) Meanwhile, the direct-current (DC) pad array (right) is applied to control the modulator operating point effectively through the TiN heater of each device.

(Main part, line 146) Near the waveguide output port, TiN heaters are adopted on the waveguides to control the operating point of the modulator.

(Supplementary Section 1) Near the output of the modulator, TiN heaters are adopted on the waveguides to precisely control the operating point of the modulator to be centered at the quadrature point.

Comment 13: *"The term "modulation efficiency" is frequently used, but is nowhere defined and can be guessed only from the units."*

Our reply: Thanks for this comment. Modulation efficiency is an important common indicator to measure the modulation capability of the modulator, which is defined as $V_{\pi}L$, the product of the modulator half-wave voltage and the modulation arm length, where V_{π} is the half-wave voltage, defined as the voltage required when the optical signal changes the phase by π in the modulation arm, and L is the modulation arm length of the modulator. Higher modulation efficiency means that the value of $V_{\pi}L$ is smaller, that is, the half-wave voltage of the device is smaller, or the required modulation arm length is shorter. A higher modulation efficiency will facilitate the realization of low driving voltage of the modulator and miniaturization of the device, thereby realizing low power consumption and high-density integration in practical applications. In the device design, optimizing the structure of modulators to increase the overlap area of the optical and electrical fields can increase modulation efficiency.

Meanwhile, regarding the phase modulation capability of the slow-light modulator, we define the term "efficiency factor" as $\varphi/\pi V$ to characterize the phase change per voltage, which is in the Supplementary Section 2. Here, the phase accumulation can be enhanced by the slow-light effect. To make it easier to understand, we have added an explanation of this term to the text.

(Supplementary Section 2) Meanwhile, the efficiency factor (define as $\varphi/\pi V$ to characterize the phase change per voltage) can be effectively compensated

Comment 14: *"In this context: What was the actual drive voltage swing during operation? With $V_{\pi}L = 8.2Vmm$ and $L = 250\mu m$, $V_{\pi} = 33V$ would result, an extrordinarily large value for a drive frequency of 90GHz and a symbol rate of 200GBd."*

Our reply: Thanks for the question. For the driving voltage in the main experiments, the high-speed differential signals are amplified by a commercial SHF single-ended driver to obtain a 5-V V_{pp} . We have already added the value of V_{pp} in the manuscript (for differential signals, the V_{pp} generally refers to the voltage value applied on each modulation arm [*Nat. Electron.*, 6(11), 910-921 (2023)], [*Optica*, 11(9), 1212-1219 (2024)], [*Opt. Exp.*, 27(10), 14321-14327 (2019)], [*Opt. Lett.*, 39(16), 4839-4842, (2014)], [*J. Light. Technol.*, 32(12), 2240-2247 (2014)]). Meanwhile, for $V_{\pi}L$ is 0.82 V·cm and L is 249 μm , the $V_{\pi}L$ is 33 V indeed. Therefore, to achieve the maximum phase change within a limited modulation arm, we

choose the push-pull working mode with dual-drive GSGSG-type RF electrodes. In this way, based on the differential signals of 5-V V_{pp} for each modulation arm, the overall swing is 10 V, reaching $\sim 30\%$ of V_{π} , which is basically enough for transmission with the equalizers. In fact, in previous works, to achieve effective signal loading, the driving voltage does not necessarily have to reach the half-wave voltage, and the modulation depth will be basically sufficient. For instance, in [*Optica*, 7(11), 1514-1516 (2020)], the modulation efficiency of the silicon MZM is 1.5 V-cm, considering the 2.5 mm modulation arm, the V_{π} is about 6 V. And the overall swing is 1 V (0.5-V V_{pp} for each arm), which has reached $\sim 17\%$ of V_{π} . We hope the analysis can clear the reviewer's concerns.

(Main part, line 571) then the differential signals are amplified by a commercial SHF single-ended driver to obtain a 5-V V_{pp} and then injected into the modulator working under the dual-drive push-pull configuration.

Comment 15: "No insertion loss of the modulator was provided, neither fibre-to-fibre, nor for the modulator alone."

Our reply: Thanks for the comments. We have supplemented the insertion loss of the modulator and the coupling loss to the manuscript. The insertion loss of the modulator is measured to be 10.5 dB, in which 9.1 dB comes from the modulation arms and the remaining from directional couplers and routing waveguides. For the coupling loss of input and output edge couplers, the coupling loss by a pair of edge couplers is 10 dB with 5-dB loss for one side with the optical fiber.

(Main part, line 198) For the optical loss of the modulator, the unoptimized coupling loss between a pair of edge couplers and the fibers is 10 dB (5-dB loss for each one), and the insertion loss of the Si-SLMs is measured to be 10.5 dB (9.1 dB from the modulation arms, remaining from directional couplers and routing waveguides). The optical loss can be further reduced by improving the manufacturing process or introducing a transition structure between the conventional waveguide and the slow light waveguide.

Comment 16: "What was the modulator chirp and the interdependence of phase and amplitude modulation of the pn junctions?"

Our reply: The chirp effect refers to the nonlinear change of the frequency or phase of the signal during the modulation process. In the plasma dispersion effect of silicon, both the real (refraction) and imaginary (absorption) parts of the refractive index change with the carrier concentration. Therefore, although the carrier absorption effect is relatively weak, the absorption effect would still raise some chirp effect by introducing additional intensity modulation into the normal main phase modulation process under Mach-Zehnder interferometric (MZI) architecture, resulting in nonlinear distortions.

First, for the refractive index changes induced by free carriers, we may use the complex refractive index to represent the variation in the refractive index as:

$$\tilde{\Delta n} = \Delta n + i\Delta k$$

Here, Δn denotes the change in the real part of the refractive index, and Δk denotes the change in the imaginary part of the refractive index. We define the changes in free-carrier concentrations as ΔN_e and ΔN_h , respectively. According to the empirical model proposed by Soref and Bennett [*IEEE J. Quant. Electron.* 23, 123-129 (1987)], in the telecommunication wavelengths, the change in the refractive index induced by free carriers can be approximated as:

$$\Delta n(\Delta N_e, \Delta N_h) = -(8.8 \times 10^{-22} \Delta N_e + 8.5 \times 10^{-18} (\Delta N_h)^{0.8})$$

$$\Delta \alpha(\Delta N_e, \Delta N_h) = 8.5 \times 10^{-18} \Delta N_e + 6.0 \times 10^{-18} \Delta N_h$$

In this expression, $\Delta \alpha$ represents the additional absorption coefficient induced by free carriers. The imaginary part of the refractive index, Δk and the absorption coefficient $\Delta \alpha$, are related as:

$$\Delta k = \frac{\lambda}{4\pi} \Delta \alpha$$

Consider a modulator under MZI with a length of L . Suppose the input optical field is given by:

$$E_{in}(t) = E_0 \exp(i\omega t)$$

After passing through a multi-mode interference, the input field is evenly split into the two arms. Let one arm serve as the modulation arm, where the injection of free carriers induces changes in both the phase change and absorption, while the other arm remains unchanged. Denoting the refractive index of the unmodulated arm as n_0 the output field recombined at the coupler can be expressed as:

$$E_{out}(t) = \frac{E_0}{\sqrt{2}} \exp(i\omega t) \left[\exp\left(\frac{i2\pi n_0 L}{\lambda}\right) + \exp\left(\frac{i2\pi(n_0 + \Delta n)L}{\lambda} - \frac{\Delta \alpha L}{2}\right) \right]$$

In this expression, the Δn and $\Delta \alpha$ terms that appear in the second exponential reflect the phase and absorption modulation introduced by the free carriers, respectively. When both phase and amplitude changes are present, the instantaneous frequency of the optical field experiences a shift, commonly referred to the chirp. Due to the coupling between Δn and $\Delta \alpha$, the output spectrum may undergo broadening and nonlinear distortion. This outcome highlights that even when the absorption effect is comparatively weak, the combined influence of amplitude variation and phase modulation can still result in noticeable chirp and associated frequency broadening.

By implementing a differential signal under a push-pull operation mode, two opposite RF signals with equal amplitudes are applied to the two modulation arms of the MZI, thereby enhancing the overall phase change. In this approach, the chirp can be further suppressed, reducing the adverse impact induced by the absorption.

Under the push-pull drive, the upper and lower arms of the MZI experience opposite refractive index changes. As a result, the output optical field can be expressed as:

$$E_{out}(t) = \sqrt{2} E_0 \exp\left(i \frac{2\pi n_0 L}{\lambda}\right) \cdot \cos\left(\frac{2\pi \Delta n(t) L}{\lambda}\right)$$

It is evident from this equation that what was originally pure phase modulation is effectively converted into intensity modulation. In this scenario, the instantaneous frequency variation of the output optical field is determined solely by the linear portion of the phase response. Consequently, near-zero chirp modulation can be achieved, thereby mitigating the deleterious effects associated with the absorption in the silicon material.

Comment 17: "How would this "small" chirp influence the transmission rate through a 300m long fibre?"

Our reply: The silicon slow-light modulator is based on a MZI architecture, inherently featuring low chirp

[*J. Light. Technol.*, 6, 87–93 (1988)]. By adopting the push-pull modulation scheme as described,

$$E_{\text{out}}(t) = \sqrt{2}E_0 \exp\left(i \frac{2\pi n_0 L}{\lambda}\right) \cdot \cos\left(\frac{2\pi \Delta n(t)L}{\lambda}\right)$$

the device can achieve near-zero chirp operation [*J. Light. Technol.*, 29, 1011–1017 (2011)]. Even when minimal chirp is present in our experiments, its influence on the bit error rate over high data rates and distances on the order of hundreds of meters is secondary compared to that of dispersion effects.

For high-speed signals in the range of several hundred Gbps, transmission over a 300 m fiber is primarily impaired by pulse broadening, inter-symbol interference, and waveform distortion induced by group velocity dispersion (GVD). Given fixed dispersion parameter D and transmission length L , the degree of pulse broadening and symbol overlap is determined by the product of the signal's bandwidth and the fiber's dispersion. Under high data-rate conditions, the symbol duration is inherently short, and even a slight amount of GVD can cause severe symbol distortion. This distortion level far exceeds the impact of any minor chirp, making GVD the main contributor to BER degradation.

To mitigate these effects, we employ a neural network-based equalizer at the receiver. Such a neural network equalizer can adaptively learn complex nonlinear mappings and effectively compensate for dispersion-induced errors. Unlike traditional linear equalizers or fixed-parameter DSP methods, a neural network equalizer extracts features from the received waveform and updates its weights via gradient descent, performing an inverse mapping to restore the original signal pattern. Building upon the inherently low-chirp output of the modulator, the neural network significantly alleviates dispersion-induced distortions as well as minor chirp effects. The BERs of PAM-4 and PAM-8 signal under B2B, 100m 200m and 300m optical fiber (Corning's SMF-28) transmission are also measured, shown in the new Figure 6 in Comment 26 below, while the new text content is added in the main part. Our experimental data at various distances consistently demonstrate the NN equalizer's effectiveness.

Comment 18: *"What was the light power injected into the modulator?"*

Our reply: In the experiment, the output optical power of the laser is measured to be 13 dBm. Also, no EDFA is placed before the modulator. Therefore, considering the 5-dB coupling loss between the input edge coupler and optical fiber, the light power actually injected into the modulator is around 8 dBm. We have added the output power of the laser source in the Supplementary Section 6, so the light power injected into the modulator can be obtained by considering the coupling loss which is also provided in the main part in Comment 15.

(Supplementary Section 6) In the experiment, the laser source provides the optical carrier signal with the output power of 13 dBm.

Comment 19: *"What was the energy requirement per bit?"*

Our reply: Thanks for the question. We have calculated the energy consumption per bit for both PAM-4 and PAM-8 using the equation below. The energy consumption per bit is calculated to be 129.1 fJ/bit for PAM-4 and 66.4 fJ/bit for PAM-8. Although the relatively large driving voltage together with the dual-drive push-pull configuration increase the power consumption to some extent, the low doping and compact footprint lead to small junction capacitance, thus controlling the energy consumption to a favorable level. We have already added the energy consumption per bit in Supplementary Section 6.

(Supplementary Section 6) To evaluate the optical energy consumption level of the modulator (excluding systems), we have calculated the energy consumption per bit of the optical modulator for both PAM-4 and PAM-8 transmission. For PAM signals, the energy consumption per bit can be calculated using the equation below^{4,5}:

$$E_b = \frac{C V_{pp}^2}{M^2 \log_2(M)} \sum_{i=0}^{M-1} (M-i) \left(\frac{i}{M-1}\right)^2$$

in which M is the modulation level, C is the junction capacitance and V_{pp} is the driving voltage.

In the device design, the relatively low doping concentration of the PN junction leads to a smaller junction capacitance. Under the bias voltage of 3 V, the designed PN junction model illustrates a junction capacitance of 1.5 pF/cm, which gives the total capacitance of 37.2 fF in a device length of 249 μm . Therefore, under a dual-drive differential signal with a 5-V V_{pp} for each modulation arm, the energy consumption per bit is calculated to be 129.1 fJ/bit for PAM-4 and 66.4 fJ/bit for PAM-8. Although the relatively large driving voltage together with the dual-drive push-pull configuration increase the energy consumption to some extent, the low doping and compact footprint lead to a small junction capacitance, thus controlling the energy consumption to a favorable level.

Comment 20: “What was the slow-down factor in the slow-light waveguide?”

Our reply: In our design, the designed group index under the slow-light effect is 6.1. Therefore, considering the group index of conventional silicon rib waveguide is 4, the slow-down factor is 1.525. In the slow-light design, the precise selection of the slow-light factor is important. If the slow-light factor is over high, the limited optical bandwidth will reduce the EO bandwidth significantly as well as result in the narrow optical passband, which will limit multi-wavelength ultrahigh-speed communications. On the other hand, if the slow-light factor is too low (or without the slow-light effect), the modulation efficiency is insufficient to support the enough modulation depth, which will also limit the footprint. Therefore, based on the optimization of the waveguide grating structure, the designed moderate group index $n_g=6.1$ is actually the optimal choice to balance bandwidth and efficiency for ultrahigh-speed applications. We have added the group index and the slow-light factor in Supplementary Section 2.

(Supplementary Section 2) Therefore, based on our optimized waveguide grating structure, the designed group index of $n_g=6.1$ is a carefully optimal value to balance bandwidth and modulation efficiency, which is the key point for ultrahigh-speed applications in our work. Considering the group index of the typical silicon waveguide is around 4.0, the slow-light factor in the designed slow-light waveguide is 1.525.

Comment 21: “What was the wave impedance of the drive electrodes?”

Our reply: Thanks for the question. For the electrode design, the RF electrodes are simulated and optimized in Ansoft HFSS (High Frequency Structure Simulator) software and the structure as well as the parameters are shown in Figure R2-3 below.

Figure R2-3. The structure of the RF electrodes. (a) Cross-section view. (b) Top view.

Based on the model, we have extracted the impedance as well as the microwave refractive index for the RF electrodes. For the impedance, the value is slightly higher than 50Ω , which is around 53Ω . This is basically the same level as the on-chip resistors demonstrated above, illustrating the favorable impedance matching of the modulator. The microwave electrode refractive index of the electrodes is around 2.1. The matching degree of microwave and light is an important factor affecting the working bandwidth, so this will limit the length of the electrode and the group index selection in the design. We have added the value of impedance and microwave refractive index in Supplementary Section 1.

(Supplementary Section 1) In order to achieve characteristic impedance matching, the signal electrode width is set to $10 \mu\text{m}$, and the distance between the ground and signal electrodes is $6.4 \mu\text{m}$, resulting in an impedance of 53Ω .

(Supplementary Section 1) In fact, at the electrode design level, there still exists a lot of space for optimization. At present, the microwave refractive index of the electrodes is around 2.1. If the segmented slow-wave electrodes can be adopted, the phase matching with the slow-light waveguide can be improved, thereby further increasing the EO modulation bandwidth of the device^{1,2}.

Comment 22: "Fig.1g and Table 1 name a "Single passband of 7nm" (corresponding to about 875GHz), but the (unexplained) Fig.S2a says that with "5 resonators" the optical bandwidth would be at best close to 250GHz."

Our reply: The calculated bandwidth from the passband and the optical bandwidth are two different concepts. If the optical bandwidth is derived from the passband width, this is based on the relationship between Q factor, optical bandwidth and passband width in an isolated single resonator. Specifically, this is assumed that the passband width $\Delta\lambda$ and the Q factor are related to each other as:

$$Q = \Delta\lambda/\lambda_0$$

However, this relationship does not hold in CROW model adopted here.

In our slow-light model, the Q factor and optical bandwidth are still related to each other through the photon lifetime τ , which is:

$$BW_0 = 1/2\pi\tau = c/(\lambda_0 Q)$$

But the relationship between optical passband width and Q has already changed. In CROW [Opt. Lett.

24, 711–713 (1999)], [*JOSA B* 21, 1665–1673 (2004)], multiple microcavities are cascaded, which will cause the passband to no longer correspond to a single resonant mode, but to a whole CROW band. Therefore, the passband width is no longer determined by the Q factor, but by the width of the CROW band. In our model, the key parameter used to determine the passband width or CROW band width instead of the Q factor is the coupling coefficient κ , which is shown as:

$$\begin{aligned}\kappa &= \int d^3\mathbf{r}[\varepsilon_0(\mathbf{r}) - \varepsilon(\mathbf{r})]\mathbf{E}_\Omega(\mathbf{r}) \cdot \mathbf{E}_\Omega(\mathbf{r} - 2N_p a \mathbf{e}_x) \\ &= 2\kappa_1(ff) \Delta \varepsilon E_0^2 \int_0^\infty e^{-\Delta \cdot x} \cdot e^{-\Delta \cdot (x+2N_p a)} dx = \kappa_0 e^{-2N_p a \Delta} / N_p\end{aligned}$$

in which E_Ω is the normalized electric field of modes.

As a result, the couplings of supercell modes bring several continuous supercell bands, and the supercell bands open several transmission windows in the spectra. Here, the spectrum width follows:

$$\Delta \lambda \propto 2 \kappa \lambda_0$$

in which λ_0 is the central wavelength of the modes.

Although the overall Q factor can still be derived from the parameters of the isolated resonant cavity and the number of cascades, the relationship between it and the passband width is no longer simple as one cavity. Based on the above reasons, the calculated bandwidth from the passband is not the same one as the optical bandwidth in the slow-light model.

We hope the analysis above can make it clearer.

Comment 23: “B1. The AI equalizer uses deep neural networks (DNN) and gated recurrent units (GRU) as a gating mechanism in recurrent neural networks. Both techniques were extensively treated in numerous publications, and even Wikipedia has articles with detailed descriptions.”

Our reply: Thanks for the comments. In this paper, compared to the traditional DNN and GRU, the employed DNN and GRU in this work are modified and extended to better adapt the high signal modulation level and speed in the IM/DD system that employs silicon slow-light modulators.

In the DNN, a specially designed multi-level real-value *sigmoid* activation function is employed. The activation functions traditionally employed in artificial neural networks for signal equalization typically exhibit two saturation regions (-1, +1 or 0, 1), such as *sigmoid*, *tanh*, and *softmax*, as shown in Figure S3a. However, in scenarios involving high-order modulation, such as PAM-4/PAM-8, where the modulation level exceeds two, the effectiveness of ANN-based equalizers tends to be inadequate. The signal equalization process can be conceptualized as a classification problem, wherein signals are categorized into different classes, such as (-3, -1, 1, 3) in PAM-4 or (-7, -5, -3, 1, 3, 5, 7) in PAM-8, respectively. It is crucial that the output values of the neurons in the ANN equalizer lie within the saturation region. For a high-order modulation signal, it should be classified into multiple categories at the output of the ANN equalizer. It is feasible to find an activation function featuring multiple saturation regions within a multi-level neuron. Here an activation function with four and eight saturation level regions are implemented through function $f(x)=2\eta_2/(1+e^{-\eta_1(x-2\alpha)})-\eta_2+2\alpha$, where $\eta_2=(1+e^{-\eta_1})/(1-e^{-\eta_1})$, α equals to -1, 0 and 1 when $x \leq -1$, $-1 < x \leq 1$, and $x > 1$, respectively. η_1 represents the gradient factor, continuity of the function is kept by η_2 . The constructed $f(x)$ has four saturation regions which are close

to the amplitude of PAM-4 (-3, -1, 1, 3), as shown in Figure S3b. Similarly, it can be extended to 8-level *sigmoid* function as shown in Figure S3c which can be used to equalize the modulated signals such as PAM-8, 8-ASK. The multi-level characteristic of the activation function $f(x)$ makes it suitable for PAM-4 and PAM-8 equalization, which has not been explored in the form of deep neural network with multiple hidden layers in the scenario of IM/DD systems that exceeding 400 Gbps data rate.

Figure S3. *Sigmoid* functions with (a) two-level, (b) four-level, (c) eight-level saturation regions. As a comparison, the system performance employing *tanh* and two-level *sigmoid* as the activation function are also measured in Figure S4. Two groups of data with different data rates of PAM-4 are selected for the measurements, the results indicate that the performance is relatively poor when using either *tanh* or two level *sigmoid* as activation functions. However, when a four-level *sigmoid* function is employed, it outperforms both the *tanh* and two-level *sigmoid* functions. Both the *tanh* and *sigmoid* functions are commonly used to introduce non-linearity in deep neural networks, *tanh* is generally favored in hidden layers for its better gradient properties, it typically has a stronger gradient compared to the *sigmoid* function, making it more suitable for training deep networks with many hidden layers.

Figure S4. System performance of employing *tanh*, two-level *sigmoid* and multilevel *sigmoid* for two groups of data with different data rates: (a) data group 1; (b) data group 2.

Long short-term memory (LSTM) is an advanced type of RNN. While RNNs suffer from short-term memory issues, the LSTM network has the ability to learn long term dependencies between time steps, insofar as it was specifically designed to address the gradient problems encountered in RNNs. Similar to the LSTM, the gated recurrent units (GRU) was designed to overcome the short-term memory issues of RNNs. GRU networks are made up of GRU cells, which are units that contain a series of gates that can control the flow of information into and out of the cell. The gates can learn to keep relevant information and discard irrelevant information, allowing the cell to remember important information for long periods of time. However, the GRU is less complex than the LSTM, as it has only two types of gates: the reset and update gates. The reset gate is used to handle short-term memory, whereas the update gate is responsible for long-term memory. Models with bidirectional structure are capable of learning information from both preceding and following data when processing the current data. The

bidirectional GRU (bi-GRU) model comprises two unidirectional GRU layers operating in opposite directions. One GRU processes the input sequence in the forward direction, starting from the beginning, while the other operates in reverse, processing the sequence from the end toward the start. By combining forward and backward GRU processing, the model incorporates information from both the future and the past to influence its current states. Some reports employ bi-GRU to compensate the fiber nonlinear impairment during the long-distance transmission in coherent system such as [Opt. Exp., 29(4), 5923, 2021], [J. Light. Technol., 41(11), 3522, 2023], but there still less bi-GRU performance evaluation reports in short reach IM/DD systems, especially for systems that employ silicon devices with data rates around 400 Gbps in a high-density data-rate multichannel wavelength-division-multiplexed chip. In this work, we employ the bi-GRU as illustrated in Figure 4b to promote the performance. Furthermore, in the revised version of the manuscript, a novel three-layer GRU (T-biGRU) equalizer is proposed and implemented, as shown in Figure 7a. The first layer of the T-biGRU equalizer is the input layer, the current symbol r_i with its k preceding and k succeeding symbols together are used as the input sequence of the network. The T-biGRU model is determined based on the state of three GRU layers. The first GRU layer processes the data in a forward direction, starting from the beginning of the sequence. The second GRU layer processes data in a backward direction, starting from the end of the sequence, to capture reverse temporal dependencies. The third GRU layer, like the first, processes data in a forward direction from the beginning of the sequence. The hidden state encapsulates the flow of symbolic information across recurrent time steps, ensuring continuity and context throughout the sequence. The outputs of the *ReLU* layer represent the probabilities that the current symbol r_i maps to each class. The bi-GRU model relies on the states of two GRUs, whereas the T-biGRU model utilizes the states of three GRUs. By integrating forward, backward, and repeated forward GRU processing, the T-biGRU model more comprehensively extracts both global and local features of the sequence, thereby significantly enhancing equalization performance.

The performance of the T-biGRU is further evaluated in new Figure 7. The results indicate that by employing T-biGRU the BER performance for around 400 Gbps signal can be further improved to smaller than 10^{-3} , which is slightly better than the bi-GRU case, proving the effectiveness of the equalizer. It is worth noting that the computational complexity of the T-biGRU is 1.5 times that of the bi-GRU, leading to increased multiplications and higher power consumption (as shown in Comment 25 below). In the equalizer choosing, it is essential to strike a balance among performance, computational complexity, and power consumption to achieve the desired system performance effectively.

Fig. 7: T-biGRU transmission results. (a) The architecture of T-biGRU equalizer. By integrating forward, backward and repeated forward GRU processing, the T-biGRU model extracts global and local features of the sequence more comprehensively, thereby further enhancing equalization performance. (b) T-biGRU PAM-4 eye diagram and constellation at 400 Gbps. (c) T-biGRU PAM-8 eye diagram and constellation at 390 Gbps. (d) BER curves of PAM-4 signal with increasing data rates (280 Gbps, 320 Gbps, 360 Gbps and 400 Gbps) of different channels. (e) BER curves of PAM-8 signal with increasing data rates (300 Gbps, 330 Gbps, 360 Gbps and 390 Gbps) of different channels. Ch2, Ch4, Ch6 and Ch8 are shown here, with all BERs lower than 10^{-3} .

We have already added the analysis of DNN in Supplementary Section 3 (new Figure S3, new Figure S4) and the new model and experimental results of T-biGRU in main part and Supplementary Section 4 (new Figure 7, new Figure S8, new Figure S9)

(Supplementary Section 3) In the DNN, a specially designed multi-level real-value *sigmoid* activation function is employed. The activation functions traditionally employed in artificial neural networks for signal equalization typically exhibit two saturation regions (-1, +1 or 0, 1), such as *sigmoid*, *tanh*, and *softmax*, as shown in Fig.S3a. However, in scenarios involving high-order modulation, such as PAM-4/PAM-8 in IM/DD systems, where the modulation level exceeds two, the effectiveness of ANN-based equalizers tends to be inadequate. The signal equalization process can be conceptualized as a classification problem, wherein signals are categorized into different classes, such as (-3, -1, 1, 3) in PAM-4 or (-7, -5, -3, 1, 3, 5, 7) in PAM-8, respectively. It is crucial that the output values of the neurons in the ANN equalizer lie within the saturation region. For a high-order modulation signal, it should be classified into multiple categories at the output of the ANN equalizer. It is feasible to find an activation function featuring multiple saturation regions within a multi-level neuron. Here an activation function with four and eight saturation level regions are implemented through function $f(x)=2\eta_2/(1+e^{-\eta_1(x-2\alpha)})-\eta_2+2\alpha$, where $\eta_2=(1+e^{-\eta_1})/(1-e^{-\eta_1})$, α equals to -1, 0 and 1 when $x\leq-1$, $-1<x\leq 1$, and $x>1$, respectively. η_1 represents the gradient factor, continuity of the function is kept by η_2 . The constructed $f(x)$ has four saturation regions which are close to the amplitude of PAM-4 (-3, -1, 1, 3), as shown in Fig.S3b. Similarly, it can be extended to 8-level sigmoid function as shown in Fig.S3c which can be used to equalize the modulated signals such as PAM-8, 8-ASK. The multi-level characteristic of the activation function $f(x)$ makes it suitable for PAM-4 and PAM-8 equalization, which has not been explored in the form of deep

neural network with multiple hidden layers in the scenario of IM/DD systems.

As a comparison, the system performance employing *tanh* and two-level *sigmoid* as the activation function are also measured, as shown in Fig.S4. Two groups of data (group 1 in Fig.S4a and group 2 in Fig.S4b) with different data rates (130 Gbps and 170 Gbps) of PAM-4 are selected for the measurements, the results indicate that the performance is relatively poor when using either *tanh* or two level *sigmoid* as activation functions. However, when a four-level *sigmoid* function is employed, it outperforms both the *tanh* and two-level *sigmoid* functions. Both the *tanh* and *sigmoid* functions are commonly used to introduce non-linearity in deep neural networks, *tanh* is generally favored in hidden layers for its better gradient properties, it typically has a stronger gradient compared to the *sigmoid* function, making it more suitable for training deep networks with many hidden layers.

(Supplementary Section 4) GRU networks are made up of GRU cells, which are units that contain a series of gates that can control the flow of information into and out of the cell. The gates can learn to keep relevant information and discard irrelevant information, allowing the cell to remember important information for long periods of time. However, the GRU is less complex than the LSTM, as it has only two types of gates: the reset and update gates. The reset gate is used to handle short-term memory, whereas the update gate is responsible for long-term memory. Models with bidirectional structure are capable of learning information from both preceding and following data when processing the current data. The bi-GRU model comprises two unidirectional GRU layers operating in opposite directions. One GRU processes the input sequence in the forward direction, starting from the beginning, while the other operates in reverse, processing the sequence from the end toward the start. By combining forward and backward GRU processing, the model incorporates information from both the future and the past to influence its current states. The bi-GRU model relies on the states of two GRUs, whereas the T-biGRU model utilizes the states of three GRUs. By integrating forward, backward, and repeated forward GRU processing, the T-biGRU model more comprehensively extracts both global and local features of the sequence.

(Supplementary Section 4) Furthermore, the T-biGRU transmission results are evaluated for both PAM-4 and PAM-8 signals. The T-biGRU PAM-4 results of eye diagrams, constellations and BERs for Ch2, Ch4, Ch6 and Ch8 are demonstrated in Fig.~S8, from 280 Gbps to 400 Gbps. Meanwhile, the T-biGRU PAM-8 results from 300 Gbps to 390 Gbps are demonstrated in Fig.~S9. The experimental results illustrate that the quality of eye diagrams and constellations is further enhanced, and the BERs for all channels can be improved to be smaller than 10^{-3} , even up to around 400 Gbps, while the high consistency is maintained between different channels.

(Main part, line 360) Some reports employ bi-GRU to compensate the fiber nonlinear impairment during the long-distance transmission in coherent system^{67,70}, but there still less bi-GRU performance evaluation reports in short-reach IM/DD systems, especially for systems that employ silicon devices with data rates around 400 Gbps in a high-density chip. In this work, we employ the bi-GRU as illustrated in Fig.~4b to promote the transmission performance. Meanwhile, based on bi-GRU, a new three-layer GRU equalizer (T-biGRU) is further proposed and implemented, as shown in Fig.~7a. The T-biGRU model is determined based on the state of three GRU layers. The first GRU layer processes the data in a forward direction, starting from the beginning of the sequence. The second GRU layer processes data in a backward direction to capture reverse temporal dependencies. The third GRU layer, like the first, processes data in a forward direction from the beginning of the sequence (see Methods). The hidden

state encapsulates the flow of symbolic information across recurrent time steps, ensuring continuity and context throughout the sequence. The bi-GRU model relies on the states of two GRU layers, whereas the T-biGRU model utilizes the states of three GRU layers. By integrating forward, backward, and repeated forward GRU processing, the T-biGRU model more comprehensively extracts both global and local features of the sequence, thereby further enhancing equalization performance.

(Main part, line 458) Furthermore, the transmission results of employing T-biGRU equalizer (Fig.~7a) are evaluated for both PAM-4 (400 Gbps in Fig.~7b) and PAM-8 (390 Gbps in Fig.~7c). For BER curves with increasing data rates of different channels (Ch2, Ch4, Ch6 and Ch8 are shown for example here), PAM-4 results (280 Gbps, 320 Gbps, 360 Gbps, 400 Gbps) are shown in Fig.~7d, while PAM-8 results (300 Gbps, 330 Gbps, 360 Gbps, 390 Gbps) are shown in Fig.~7e. The results indicate that by employing T-biGRU equalizer, the BER performance for around 400 Gbps transmission of all channels can be improved to be smaller than 10^{-3} , which is slightly better than the bi-GRU case. The T-biGRU results for all eye diagrams, constellations and BERs corresponding to Fig.~7 are demonstrated in Supplementary Section 4. However, it is worth noting that the computational complexity of the T-biGRU is 1.5 times that of the bi-GRU (see Methods), leading to increased multiplications and higher power consumption (Supplementary Section 6). In the equalizer selection, it is important to strike a balance among performance, computational complexity, and power consumption to achieve the desired system performance effectively.

Comment 24: *"B2. I do not see the novelty in providing "Algorithm details" instead of specifying the energy requirements, and how often the learning process must be repeated if any of the optical or electronic components has to be exchanged, or if the modulator ages and/or its bias drifts."*

Our reply: For the energy requirements, we have discussed this in detail in Comment 25 below. For the repeated learning process, in optical equalization using neural network-based methods, the frequency of retraining (learning process) depends on the stability and variations of the system components. Specifically: 1) Component replacement. When components such as photodetectors, electrical/optical amplifiers, or modulators are exchanged, the learning process must typically be repeated. This is because such changes can significantly alter the signal characteristics due to the differentiated device response, rendering the previously trained model ineffective. 2) Aging effects. As optical modulators or other components age, their performance metrics including insertion loss, bandwidth, or nonlinearity may degrade over time. This gradual change requires periodic re-training to adapt to the new system state. 3) Bias drift. If the modulator's bias drifts due to temperature fluctuations or voltage instability, the neural network model may fail to compensate for the resulting signal distortion. In this case, re-training is needed when the drift exceeds the model's tolerance level (resulting in significant deviations in performances like BER, SNR, etc.). 4) Noise characteristics. Changes in the noise environment, such as increased crosstalk or interference, might necessitate model updates.

In the experiments, we replace the electrical driver and EDFA with different models and evaluate the resulting system performance variations. In the case of driver replacement experiment, the neural network is first trained using data collected from the system that employs driver 1. The trained network is then tested using data collected from the system employs driver 1 and replaced driver 2 (another model) and the results are presented in Figure S10a. The results show that using the trained network model of driver 1 is not suitable for the testing of employing Driver 2, there is obvious performance degradation. The results of training the network using replaced driver 2 and testing with the same driver

are also presented. The situation is similar for the case of EDFA replacement, as shown in Figure S10b. We manually change the modulator DC bias to simulate the bias drift, the results are shown in Figure S10c. It indicates that there is distinguished deterioration using the trained network model of 3 V bias for the testing of bias voltages at 1.5 V, 2 V and 2.5 V.

Figure S10. Impact of component replacement to system performance. (a) replacing driver. (b) replacing EDFA. (c) changing DC bias.

The retraining frequency depends on how fast the components degrade or how often they are replaced. For systems with stable components, retraining might only be required on a seasonal or annual basis, while systems with frequent changes might need more regular updates.

We hope the above discussion can make it clearer. We have added a new part in the Supplementary Section named as “Retraining frequency of the neural network”, shown as follows.

(Supplementary Section 5) When optical or electronic components such as photodetectors, electrical/optical amplifiers, or modulators are exchanged in real network scenario, the learning process of the neural network must typically be repeated. This is because such changes can significantly alter the signal characteristics due to the differentiated device response, rendering the previously trained model ineffective. As optical modulators or other components age, their performance metrics including insertion loss, bandwidth, or nonlinearity may degrade over time. This gradual change requires periodic retraining to adapt to the new system state. If the modulator's bias drifts due to temperature fluctuations or voltage instability, the neural network model may need to be retrained to compensate for the resulting signal distortion. In this case, retraining is needed when the drift exceeds the model's tolerance level. Changes in the noise environment, such as increased crosstalk or interference, might necessitate model updates. The retraining frequency depends on how fast the components degrade or how often they are replaced. For systems with stable components, retraining might only be required on a seasonal or annual basis, while systems with frequent changes might need more regular updates.

We conduct the experiments to analyze the impact of component replacement for neural network. In the experiments, we replace the electrical driver and optical EDFA and evaluate the resulting system performance variations for different data rates. In the case of driver replacement experiment, the neural network is first trained using data collected from the system that employs driver 1 (origin driver). The trained network is then tested using data collected from the system employs driver 1 and replaced driver 2 (replaced driver) of different model. The process is repeated five times, and the average results are presented in Fig.~S10a. The results show that using the trained network model of driver 1 is not suitable for the testing of employing driver 2, there exists obvious performance degradation. The results of training the network using replaced driver 2 and testing with the same replaced driver is also

presented, which is similar to the origin driver situation. For the case of EDFA replacement, the situation is similar with the driver case, as shown in Fig.~S10b. In terms of DC bias, we manually change the modulator DC bias to simulate the bias drift, the results of 400 Gbps PAM-4 are shown in Fig.~S10c. It indicates that there is distinguished deterioration using the trained network model of 3 V bias for the testing of bias voltages at 1.5 V, 2 V and 2.5 V. Furthermore, by adjusting the power after the modulator through variable optical attenuator (VOA), the impact of output optical power variation to the system performance at 400 Gbps PAM-4 is evaluated, the results are shown in Fig.~S11. The results indicate that variations in output intensity have an obvious impact on system performance. Compared to retraining the network at each specific output power, utilizing a model trained at a fixed power level (-13.94 dBm in Fig.~S11a and -15.94 dBm in Fig.~S11b) to evaluate the performance at other power levels results in performance degradation. Meanwhile, as shown in the results, using the model trained at -13.94 dBm and -15.94 dBm to test its corresponding data, it shows favorable performance.

The strategies to predict and minimize the need for retraining neural network-based optical equalization systems could be: 1) Monitoring and diagnostic systems. Use real-time diagnostics to track operating parameters of components such as modulator bias voltage, temperature, aging indicators or component degradation before it impacts performance significantly. Continuously monitor performance metrics like BER, SNR, eye diagram quality, EVM, etc., significant deviations of these performance metrics can indicate the need for retraining. 2) Adapt the model continuously to minimize the need for complete retraining, such as using a small portion of incoming data for periodic updates to the model weights without retraining from scratch. It has been demonstrated that using transfer learning can drastically reduce the training time and training data requirements when changes on the transmission setup occur. 3) Reduce the sources of variation, such as implementing robust thermal management to minimize temperature-induced drift, using feedback circuit loops to maintain modulator bias within the optimal range dynamically, employing higher-quality or more stable components with slower aging or lower drift characteristics.

Comment 25: *"I feel that the energy discussion (including the light source, the modulator, the electrical and the optical amplifiers as well as the AI equalizers and the individual energy for cooling where required) would be of paramount importance, if these devices, as is suggested by the submission, should become the future basis for optical interconnects inside a data center."*

Our reply: Thanks for the suggestions. We fully agree with the reviewer's concerns regarding power consumption and the potential applicability of this approach in future optical interconnect scenarios. For the energy consumption per bit of the optical modulator (excluding systems), we have calculated the E_b of PAM-4 and PAM-8 signals in Comment 19 above. For the experiment system in this study, the power consumption of various components—including the light source, modulator, electrical amplifiers, optical amplifiers, and AI equalizer—can be evaluated in a systematic manner. Below, we provide a detailed discussion of the energy consumption of these components under our experimental conditions. It is noteworthy that the current work remains at the proof-of-concept stage. As device integration and fabrication processes advance, the overall power consumption is expected to be significantly reduced.

The bench-top laser source provides the optical carrier signal with an output power of 13 dBm. The maximum apparent power consumption is 100 VA, with real power consumption not exceeding 100 W.

The main power consumption of modulation part includes the DC bias, TiN heater and drivers. The

recorded voltage and current values allowed us to calculate the actual system power consumption:

DC bias power consumption: Two arms were each given a DC bias of 2.5 V. The currents were 0.077 A and 0.080 A, thus the total current is 0.157 A. The power consumption was: $2.5 \text{ V} \times 0.157 \text{ A} = 0.393 \text{ W}$.

TiN heater power consumption: With one TiN heater driven at 2.29 V and 0.011 A, the corresponding power consumption was: $2.29 \text{ V} \times 0.011 \text{ A} = 0.025 \text{ W}$.

Driver power consumption: To save the port of power sources, two drivers were connected in parallel to the same power supply channel. The measured operating voltage and current of the power channel were 9 V and 0.531 A. Thus, the combined power consumption of drivers was: $9 \text{ V} \times 0.531 \text{ A} = 4.779 \text{ W}$.

A discrete EDFA was placed before the PD to enhance the input optical power. The conventional EDFA with similar gain and noise has a maximum power consumption of about 20 VA (with real power not exceeding 20 W). Here, we set the EDFA power consumption to approximately 20 W.

The AI equalization was performed on a computer with an Intel Core i9-13900HX processor, 32 GB of 5600 MHz memory, and an Nvidia RTX4060 GPU. Power consumption data were obtained using HWINFO [<https://www.hwinfo.com/>] and based on our previous research [*Nat. Commun.*, 14(66), (2023)].

The power consumption of the system with different AI equalizers is shown in new Table S1.

Table S1. Summary of Power Consumption:

Components	Voltage(V)	Current(A)	Power(W)
Laser	N/A	N/A	~100.000
DC bias	2.500	0.157	0.393
TiN heater	2.290	0.011	0.025
Driver	9.000	0.531	4.779
EDFA	N/A	N/A	~20.000
DNN	N/A	N/A	~34.690
bi-GRU	N/A	N/A	~66.570
T-biGRU	N/A	N/A	~67.560
Total Power Consumption			Min~159.887 Max~192.757

Therefore, depending on the chosen type of AI equalizer, the total power consumption of the experimental link ranges from approximately 159.887 W to 192.757 W.

It is important to reiterate that our experiments are still at an early proof-of-concept stage. With future integration of on-chip frequency combs in our previous research [*Nature* 605, 457–463 (2022)], the light source can be changed to integrated on-chip light source. For optical waveguide amplifiers, with integrated erbium-doped amplifiers [*Science* 376, 1309–1313 (2022)], and on-chip semiconductor optical amplifiers (SOAs) [*IEEE J. Sel. Top. Quant. Electron.* 22, 78–88 (2016)], the currently used discrete

EDFA could be replaced by fully integrated photonic devices. By employing frequency combs, the on-chip pump power for frequency comb generation can be as low as about 98 mW [*Nature* 562, 401–405 (2018)]. When using on-chip SOAs, the typical consumption can be around 390 mW [https://www.thorlabs.com/navigation.cfm?guide_id=2104]. Moreover, the drivers can potentially operate at sub-pJ/bit energy levels [*Nat. Electron.* 6(11), 910–921 (2023)]. Taken together, these integrated devices promise lower power consumption than currently measured in our experiment.

For the AI equalizer design, beyond our current implementation on a general-purpose GPU without dedicated complexity optimization, there exist a variety of mature neural network compression and acceleration techniques that can substantially improve energy efficiency. Pruning approaches [*NIPS* 28, 1135–1143 (2015)] remove redundant parameters to reduce memory and input/output overhead, thereby lowering memory and data bus power consumption. Quantization methods [*ECCV* 9908, 525–542 (2016)] replace floating-point logic with lower-bit-width fixed-point or binary arithmetic, fundamentally reducing multiply-accumulate operations and associated data loading energy at the circuit level. Distillation strategies [*ICML* 80, 1607–1616 (2018)] further transfer knowledge from large teacher networks to smaller student models, decreasing overall complexity and thus power usage. In addition to these algorithmic enhancements, leveraging GPU-optimized libraries such as NVIDIA’s cuDNN [*arXiv:1410.0759* (2014)] can improve computational efficiency. Techniques like kernel fusion, as enabled by frameworks such as TVM [*OSDI* 13, 578–594 (2018)], reduce redundant memory accesses lowers both compute and energy requirements. By judiciously combining these hardware and software optimizations with pruning, quantization, and distillation strategies, it is possible to achieve lower-power, higher-efficiency AI equalizers suitable for future implementations.

Meanwhile, we have added a new part “Power consumption” in Supplementary Section 6.

(Supplementary Section 6) For the practical system, in this proof-of-concept demonstration, we evaluated the power consumption of each component involved in the experiment. Although not optimized for energy efficiency, this provides a reference scenario that can be used as a benchmark for future integration and optimization designs. In the experiment, the bench-top laser source provides the optical carrier signal with the output power of 13 dBm. The maximum apparent power consumption of this laser is 100 VA, with real power consumption not exceeding 100 W. For the modulation part, the main power consumption includes the DC bias, TiN heater, and drivers. The recorded voltage and current values in the experiment process allowed us to calculate the actual system power consumption. For DC bias, two modulation arms were given a DC bias of 2.5 V each with the measured currents of 0.077 A and 0.080 A, resulting in a power consumption of 0.393 W. To ensure that the modulator operates at the quadrature point, one TiN heater was driven at 2.290 V and 0.011 A, thereby the corresponding power consumption was 0.025 W. For the electrical amplifiers, two drivers were connected in parallel to the same power supply channel. The measured operating voltage and current were 9 V and 0.531 A, thus the combined power consumption of drivers was 4.779 W. It should be noted that we did not employ additional thermoelectric cooling (TEC) for the modulator due to the intrinsic thermal stability of the Si-SLM. For the optical amplifier, a discrete erbium-doped fiber amplifier (EDFA) was placed before the photodetector (PD) to enhance the input optical power. The conventional discrete EDFA with similar gain and noise has a maximum power consumption of about 20 VA, with real power not exceeding 20 W. For simplicity and comparative purposes, we set the EDFA power consumption to approximately 20 W. For the AI equalizers, referring to the prior measurement work⁶, the power consumption was monitored using HWINFO, ranging from ~35 W for the DNN to ~67 W for

biGRU and T-biGRU implementations.

Summing these contributions, the total system power consumption varied from approximately 159.887 W to 192.757 W, depending on the chosen AI equalizer. And all these values are listed in Table S1. It is important to reiterate that our experiments are still at an early proof-of-concept stage, aimed at demonstrating the potential of Si-SLMs for ultra-high-speed, large-throughput signal transmission. It is anticipated that ongoing advancements, such as on-chip frequency combs⁷, integrated optical amplifiers^{8,9}, and monolithic integration of light sources¹⁰, modulators, and drivers¹¹, will substantially reduce these indicators. Likewise, techniques like network pruning¹², quantization¹³, and distillation¹⁴, as well as hardware-accelerated inference¹⁵, are expected to lower AI equalization power. Overall, as device integration and algorithmic optimization advance, the energy efficiency of Si-SLMs for ultrahigh-speed data center interconnections can be improved dramatically.

Comment 26: “In this case, the data of a transmission experiment with an optical fibre over a distance of at least 300m would be required.”

Our reply: Thanks for the suggestion. We fully agree with the reviewer that the transmission experiment with different fiber distances is quite important to the practical applications. We have conducted the new transmission experiment of different distances (B2B, 100 m, 200 m, 300 m).

The BER performances for PAM-4 and PAM-8 signals under 100m 200m and 300m optical fiber (Corning’s SMF-28) are further measured and the results are shown in new Figure 6 below. Due to the impact of the power fading effect in optical fiber, the BER performance gradually degrades with increasing transmission distance compared to the back-to-back (BTB) scenario for both the cases of PAM-4 and PAM-8. We have already added the experimental results in the main part and constructed the new Figure 6 to describe the results.

Figure 6. BER performance for different transmission distances (a) PAM-4 signals of 280 Gbps, 320 Gbps, 360 Gbps, 400 Gbps for B2B, 100 m, 200 m and 300 m SSMF. (b) PAM-8 signals of 300 Gbps, 330 Gbps, 360 Gbps, 390 Gbps for B2B, 100 m, 200 m and 300 m SSMF. The bi-GRU equalizer is utilized here. For the transmission distance of 300 m, the BERs remain below HD-FEC threshold at 360 Gbps, but deteriorate beyond the HD-FEC threshold at 400 Gbps.

(Main part, line 447) The above experimental results are based on back-to-back (B2B) scenario, and the

transmission penalty for 100 m, 200 m and 300 m standard single-mode fiber (SSMF) is then experimentally assessed for both PAM-4 and PAM-8 signals with the bi-GRU equalizer. Considering the consistency of all channels has been proven to be favorable, the BER penalty is illustrated for one channel (Ch3) at 1550 nm. The transmission results for different SSMF distances are demonstrated in Fig. 6, in which Fig. 6a shows the BERs of PAM-4 (280 Gbps, 320 Gbps, 360 Gbps, 400 Gbps) and Fig. 6b shows the BERs of PAM-8 (300 Gbps, 330 Gbps, 360 Gbps, 390 Gbps). Due to the impact of the power fading effect in optical fiber, the BER performance gradually degrades with increasing transmission distance compared to B2B scenario for both cases of PAM-4 and PAM-8. For the transmission distance of 300 m, the BERs still remain below HD-FEC threshold at 360 Gbps, but deteriorate beyond HD-FEC threshold at 400 Gbps.

Comment 27: *“What would be the latency of such an optical interconnect?”*

Our reply: In optical equalization where a neural network is employed, the latency introduced by the network depends on several factors, primarily including inference latency and data transfer latency.

1) Inference latency refers to the time required for a neural network to process an input signal and generate the corresponding output. The inference latency depends on: a) Network architecture: The type of layers (e.g., convolutional layers, fully connected layers, or recurrent layers like GRUs), the total number of layers, and the size of each layer; b) Number of operations: Each layer in a neural network performs a certain number of mathematical operations. The most common mathematical operations in neural networks are multiply-and-accumulate operations (MACs). The total number of operations can be calculated based on the number of neurons, layer types, and input/output sizes; c) Hardware throughput: The type of processing hardware (e.g., CPU, GPU, FPGA, or ASIC) determines how quickly the operations can be performed. Different hardware platforms have different processing speeds (measured in FLOPs: floating-point operations per second).

The inference latency can be calculated by:

$$L_{\text{inference}} = \text{Total Operations} / \text{Hardware Throughput}$$

Where Total Operations refers to the total number of FLOPs required to process one input, and Hardware Throughput refers to the number of FLOPs the hardware can handle per second.

2) Data transfer latency: The time required to move input data from memory to the processing unit (CPU, GPU, FPGA, ASIC) and the output from the processing unit back to memory or to the next processing stage. This depends on: A) Memory bandwidth: The speed at which data can be read from and written to memory; B) Data size: The size of the data need to be processed.

The data transfer latency can be estimated as:

$$L_{\text{transfer}} = \text{Data Size} / \text{Memory Bandwidth}$$

The total latency L_{total} can be expressed as:

$$L_{\text{total}} = L_{\text{inference}} + L_{\text{transfer}}$$

In our experiment, at the receiver side, the received data of the signal is sampled by a real-time oscilloscope and stored for offline digital signal processing and BER calculation. The data processing was carried out using a Lenovo Legion Y9000P personal computer, equipped with Intel Core™ i9-13900HX processor, 32 GB of RAM with 5600 MHz memory frequency, and GPU Nvidia RTX4060. Latency is defined as the time it takes to process a single symbol to be recovered. The hardware throughput, as specified in the device specifications, is 844.8 GFLOPs for the CPU and 14.56 TFLOPs for the GPU, respectively.

For the implemented DNN algorithm in our work, the inference process requires 3360 FLOPs, which were carefully calculated based on the program.

When processing by CPU, the latency equals to:

$$L_{\text{inference_DNN_CPU}} = 3360 / 844.8 \times 10^9 = 3.97727 \text{ ns}$$

In the case of GPU,

$$L_{\text{inference_DNN_GPU}} = 3360 / 14.56 \times 10^{12} = 0.23077 \text{ ns}$$

For the bi-GRU, the number of operations can be calculated by: $2 \times 2 \times 3 \times n_H(n_E + n_H)$, where n_H represents the number of GRU units that used in one direction, n_E represents the input size of the bi-GRU layer. In our system, $n_H=200$, $n_E=2^{15}-1$, so the required FLOPs of bi-GRU is $2 \times 2 \times 3 \times 200 \times (200 + 2^{15} - 1) = 7.912 \times 10^7$.

When processed by the CPU,

$$L_{\text{inference_Bi-GRU_CPU}} = 7.912 \times 10^7 / 844.8 \times 10^9 = 0.09376 \text{ ms}$$

In the case of GPU,

$$L_{\text{inference_Bi-GRU_GPU}} = 7.912 \times 10^7 / 14.56 \times 10^{12} = 0.00543 \text{ ms}$$

For the T-biGRU model, the total number of operations is calculated using $3 \times 2 \times 3 \times n_H(n_E + n_H)$, By substituting specific values, it equals to $3 \times 2 \times 3 \times 200 \times (200 + 2^{15} - 1) = 1.1868 \times 10^8$.

When processed by the CPU,

$$L_{\text{inference_Bi-GRU_CPU}} = 1.1868 \times 10^8 / 844.8 \times 10^9 = 0.14048 \text{ ms}$$

In the case of GPU,

$$L_{\text{inference_Bi-GRU_GPU}} = 1.1868 \times 10^8 / 14.56 \times 10^{12} = 0.00815 \text{ ms}$$

For the data transfer latency, the memory bandwidth can be calculated by:

Memory Bandwidth = Memory Frequency × Data Transferred per Cycle × Number of Memory Channels, which equals to $5.6 \text{ G} \times 16 \text{ B} = 89.6 \text{ GB/s}$ in our system, the data size is 4.2MB, so:

$$L_{\text{Transfer, DDR5}} = 4.2 \text{ MB} / 89.6 \text{ GB} = 0.04688 \text{ ms}$$

Additionally, when GPU computing is employed, the data will often be transferred through GPU memory rather than standard DDR5. Since GPU memory typically provides much higher bandwidth compared to general-purpose system memory, the data transfer latency between the GPU itself and its memory can be significantly reduced. From the specification, the GPU memory bandwidth is 288 GB/s, so:

$$L_{\text{Transfer, GDDR}} = 4.2 \text{ MB} / 288 \text{ GB} = 0.01458 \text{ ms}$$

Details of the latency information are summarized in Table R1.

Table R1. Details of the latency for different AI equalizer under different hardware

AI Equalizer type	Inference latency	Data transfer latency	Total latency
DNN	CPU: 3.97727 ns	DDR: 0.04688 ms	CPU: 0.04688 ms
	GPU: 0.23077 ns	GDDR: 0.01458 ms	GPU: 0.01458 ms
bi-GRU	CPU: 0.09376 ms	DDR: 0.04688 ms	CPU: 0.14064 ms
	GPU: 0.00543 ms	GDDR: 0.01458 ms	GPU: 0.02001ms
T-biGRU	CPU: 0.14048 ms	DDR: 0.04688 ms	CPU: 0.18736 ms
	GPU: 0.00815 ms	GDDR: 0.01458 ms	GPU: 0.02273 ms

To further reduce latency introduced by neural network inference, several optimization strategies can be employed. (1) Model pruning removes redundant connections and neurons, thereby decreasing the network size and computational complexity [Han, S. et al., ICLR, 2016]. (2) Quantization converts both network weights and activations to low-bit fixed-point types, substantially lowering computation overhead and memory requirements [Jacob, B. et al., CVPR, 2018]. (3) Mixed-precision training and inference selectively use appropriate data precision per layer, thus improving throughput while maintaining sufficient accuracy [Micikevicius, P. et al., ICLR, 2018]. (4) Lightweight architectures, such as

MobileNet or ShuffleNet, employ more efficient operations or reduced width to achieve higher inference speed under the same hardware constraints [Howard, A. G. et al., *arXiv:1704.04861*, 2017]. When combined with high-performance hardware accelerators, these approaches can significantly decrease latency while enhancing real-time signal processing capabilities in optical interconnect systems.

We hope the above discussion could clear the reviewer's concerns.

Response to the report from Referee #3

General comments: *“This paper presents an excellent idea of using AI to compensate for waveform distortion that arises when increasing the modulation frequency with slow-light technology, enabling the use of higher-order modulation formats. The authors demonstrate this concept effectively through a remarkable experimental achievement: achieving a data transmission rate of 400 Gbps per channel across 8-channel WDM on a Si-SLM chip, totaling 3.2 Tbps.*

The technique of enhancing modulation frequency using slow light is an impressive approach previously described by the same authors (Han et al., Sci. Adv. 9, eadi5339 (2023) 20 October 2023). This work extends that concept by incorporating AI, yet the description of AI's specific impact seems limited, making it challenging to assess the level of innovation required by NCOM. If left as it is, Sci. Adv. might be a suitable publication venue. However, for potential acceptance in NCOM, the following points should be addressed.”

Our reply: Thanks for the comments. We appreciate the reviewer’s recognition. And we apologize for not explaining the innovation of our work fully in the initial manuscript. Here, we further summarize the important innovations of our research below.

There exists a high correlation between the slow-light modulators and AI equalizers, which is the starting point of our research. By utilizing the slow-light approach, the bandwidth and modulation efficiency can be improved simultaneously. However, the insufficient linearity is not enhanced, which can be compensated by AI equalization exactly. At the manufacturing level, achieving complete uniformity remains challenging in the doping process, while the PN junction is the pure silicon modulation basis. Simultaneously, although the silicon waveguide grating structure has improved the process stability compared to two-dimensional photonic crystals, unavoidable structure fluctuations still exist. Therefore, silicon slow-light modulators require an adaptive technical approach to minimize the difference caused by process errors. Through AI equalization, the process limitations of the doped-silicon slow-light structure can be eliminated, making the deployment of large-scale slow-light modulator array possible.

We have strengthened the discussion on the relationship between the proposed AI and silicon slow light technology in the manuscript as follows. Since the adaptive characteristics of AI equalization can help mitigate the impact of fabrication sensitivity and nonlinearity in the slow light structure, this approach will further enhance the performance of the silicon slow-light modulator, which is currently the only viable method to achieve bandwidths greater than 100 GHz in silicon photonics. Also, the introduction has been reorganized to better clarify the novelty of our work, particularly emphasizing the significance of achieving 400 Gbps per lane data transmission on a standard silicon photonics platform for the first time. In fact, for ultrahigh-speed transmission around 400 Gbps, the mainstream technical solution nowadays is still heterogeneous integration technology, which requires a substantial and expensive update of the fabrication process. We believe the successful implementation of on-chip single-lane 400 Gbps with pure silicon modulators will greatly boost the industry's confidence in silicon photonics.

(Main part, line 86) Therefore, by using an ANN equalizer, the nonlinear distortion of Si-SLMs can be reduced. At the manufacturing level, achieving complete uniformity remains challenging in the doping process, while the PN junction is the modulation basis for Si-SLMs^{8,9}. Simultaneously, although one-dimensional waveguide grating structure has improved the process stability compared to two-dimensional photonic crystals^{40,42}, the structural fluctuations of doped silicon gratings still exist due to

the fabrication limitations. Therefore, Si-SLMs require an adaptive technical approach at the back end to eliminate the variations caused by process deviations, thus making the deployment of large-scale high-density Si-SLM systems possible.

(Main part, line 242) Meanwhile, the relatively high passband conformity between devices can demonstrate that the fabricated Si-SLMs have a favorable process consistency in general. However, from the spectral results it can be seen that a wavelength shift still occurs, due to the unavoidable fabrication deviations. Although the manufacturing uniformity of Si-SLMs has been improved by introducing one-dimensional CROW compared to two-dimensional photonic crystals, the performance fluctuations between doped silicon grating structures still exist. Therefore, for large-scale Si-SLM systems, the AI equalization at the back end is necessary to minimize device performance fluctuations caused by process errors.

(Main part, line 506) At the manufacturing level, since silicon material does not possess a high and consistent EO coefficient, forming PN junction is the basis for realizing pure silicon modulation, while achieving complete uniformity in doping process is still challenging. Simultaneously, although the CMOS-compatible waveguide grating structure has already improved process stability compared to photonic crystals, the structural fluctuations of doped silicon gratings still exist due to the unavoidable fabrication deviations. Therefore, through AI equalization at the back end, the variations between Si-SLMs caused by process errors can be eliminated, thereby making the deployment of large-scale transmission systems possible.

Moreover, at the AI algorithm level, we proposed a new more powerful AI-equalizer in the revised version of the manuscript. We enhanced the bi-GRU to develop a novel three-layer GRU (T-biGRU) equalizer. By integrating forward, backward, and repeated forward GRU processing innovatively, the T-biGRU equalizer extracts global and local features of the sequence more comprehensively. The new results indicate that by employing T-biGRU, the BERs for 400 Gbps signal can be further improved to smaller than 10^{-3} , which is shown below (new Figure 7, new Figure S8, new Figure S9).

Fig. 7: T-biGRU transmission results. (a) The architecture of T-biGRU equalizer. By integrating forward, backward and repeated forward GRU processing, the T-biGRU model extracts global and local

features of the sequence more comprehensively, thereby further enhancing equalization performance. (b) T-biGRU PAM-4 eye diagram and constellation at 400 Gbps. (c) T-biGRU PAM-8 eye diagram and constellation at 390 Gbps. (d) BER curves of PAM-4 signal with increasing data rates (280 Gbps, 320 Gbps, 360 Gbps and 400 Gbps) of different channels. (e) BER curves of PAM-8 signal with increasing data rates (300 Gbps, 330 Gbps, 360 Gbps and 390 Gbps) of different channels. Ch2, Ch4, Ch6 and Ch8 are shown here, with all BERs lower than 10^{-3} .

(Main part, line 360) Some reports employ bi-GRU to compensate the fiber nonlinear impairment during the long-distance transmission in coherent system^{67,70}, but there still less bi-GRU performance evaluation reports in short-reach IM/DD systems, especially for systems that employ silicon devices with data rates around 400 Gbps in a high-density chip. In this work, we employ the bi-GRU as illustrated in Fig.~4b to promote the transmission performance. Meanwhile, based on bi-GRU, a new three-layer GRU equalizer (T-biGRU) is further proposed and implemented, as shown in Fig.~7a. The T-biGRU model is determined based on the state of three GRU layers. The first GRU layer processes the data in a forward direction, starting from the beginning of the sequence. The second GRU layer processes data in a backward direction to capture reverse temporal dependencies. The third GRU layer, like the first, processes data in a forward direction from the beginning of the sequence (see Methods). The hidden state encapsulates the flow of symbolic information across recurrent time steps, ensuring continuity and context throughout the sequence. The bi-GRU model relies on the states of two GRU layers, whereas the T-biGRU model utilizes the states of three GRU layers. By integrating forward, backward, and repeated forward GRU processing, the T-biGRU model more comprehensively extracts both global and local features of the sequence, thereby further enhancing equalization performance.

(Main part, line 458) Furthermore, the transmission results of employing T-biGRU equalizer (Fig.~7a) are evaluated for both PAM-4 (400 Gbps in Fig.~7b) and PAM-8 (390 Gbps in Fig.~7c). For BER curves with increasing data rates of different channels (Ch2, Ch4, Ch6 and Ch8 are shown for example here), PAM-4 results (280 Gbps, 320 Gbps, 360 Gbps, 400 Gbps) are shown in Fig.~7d, while PAM-8 results (300 Gbps, 330 Gbps, 360 Gbps, 390 Gbps) are shown in Fig.~7e. The results indicate that by employing T-biGRU equalizer, the BER performance for around 400 Gbps transmission of all channels can be improved to be smaller than 10^{-3} , which is slightly better than the bi-GRU case. The T-biGRU results for all eye diagrams, constellations and BERs corresponding to Fig.~7 are demonstrated in Supplementary Section 4. However, it is worth noting that the computational complexity of the T-biGRU is 1.5 times that of the bi-GRU (see Methods), leading to increased multiplications and higher power consumption (Supplementary Section 6). In the equalizer selection, it is important to strike a balance among performance, computational complexity, and power consumption to achieve the desired system performance effectively.

(Supplementary Section 4) GRU networks are made up of GRU cells, which are units that contain a series of gates that can control the flow of information into and out of the cell. The gates can learn to keep relevant information and discard irrelevant information, allowing the cell to remember important information for long periods of time. However, the GRU is less complex than the LSTM, as it has only two types of gates: the reset and update gates. The reset gate is used to handle short-term memory, whereas the update gate is responsible for long-term memory. Models with bidirectional structure are capable of learning information from both preceding and following data when processing the current data. The bi-GRU model comprises two unidirectional GRU layers operating in opposite directions. One GRU processes the input sequence in the forward direction, starting from the beginning, while the other

operates in reverse, processing the sequence from the end toward the start. By combining forward and backward GRU processing, the model incorporates information from both the future and the past to influence its current states. The bi-GRU model relies on the states of two GRUs, whereas the T-biGRU model utilizes the states of three GRUs. By integrating forward, backward, and repeated forward GRU processing, the T-biGRU model more comprehensively extracts both global and local features of the sequence.

(Supplementary Section 4) Furthermore, the T-biGRU transmission results are evaluated for both PAM-4 and PAM-8 signals. The T-biGRU PAM-4 results of eye diagrams, constellations and BERs for Ch2, Ch4, Ch6 and Ch8 are demonstrated in Fig.~S8, from 280 Gbps to 400 Gbps. Meanwhile, the T-biGRU PAM-8 results from 300 Gbps to 390 Gbps are demonstrated in Fig.~S9. The experimental results illustrate that the quality of eye diagrams and constellations is further enhanced, and the BERs for all channels can be improved to be smaller than 10^{-3} , even up to around 400 Gbps, while the high consistency is maintained between different channels.

To further address the raised questions, we made the point-by-point response as follows.

Comment 1: *“Although the paper provides detailed information on Si-SLM, it would be beneficial to clarify how ANN can mitigate nonlinearities in Si-SLM. For example, showing how much the spurious-free dynamic range (SFDR) expands after applying a DNN equalizer to Si-SLM would strengthen the paper.”*

Our reply: Thanks for the questions. In fact, the AI equalizer is utilized at the back end of the system, so it is actually a full-link nonlinear distortion compensation. The role of ANN is to optimize the nonlinear distortion of the transmitted signal caused by factors such as low linearity (SFDR) of the modulator at the back end, rather than directly improving the SFDR indicator of the device. For the nonlinear effects in high-speed optical communication links, the primary sources arise from two main components: the silicon modulator and the radio-frequency (RF) devices.

In terms of the silicon slow-light modulator, since silicon material has a centrosymmetric crystal structure, it does not exhibit a Pockels effect and thus cannot directly support linear electro-optic modulation. Instead, silicon modulators employ plasma dispersion effects by doping the arms of the interferometer to form PN junctions, thereby enabling refractive index tuning [*IEEE J. Quant. Electron.*, 23, 123–129 (1987)]. As the voltage increases, the depletion width within the device does not expand linearly, showing a pronounced nonlinear growth trend. Correspondingly, the refractive index distribution also becomes nonlinear with respect to the applied voltage. During ultrahigh-speed modulation, carriers cannot fully recover to their steady state before the next cycle, causing the accumulation of nonlinear effects. Also, a Mach–Zehnder modulator inherently exhibits a cosine transfer function [*Fundamentals of Photonics (John Wiley & Sons, 2019)*]. At large modulation depths, this transfer function itself introduces nonlinearities, deviating the output from a linear mapping of the input and thus degrading the device’s modulation performance. Moreover, in practical fabrication, doping profiles, waveguide geometries, and interface roughness can lead to spatial variations in local field intensities and non-uniform optical responses. These non-ideal factors perturb the local refractive index distribution, resulting in optical field distortion and mode coupling, and in turn introduce additional nonlinear distortions into the power modulation curve.

In high-speed data transmission links, nonlinear factors also arise from electrical devices [*Microwave*

Engineering, 4th Edition (John Wiley & Sons, 2012)]. Digital-to-analog (DAC) and analog-to-digital (ADC) converters—core components necessary for generating RF signals at the transmitter and sampling signals at the receiver—introduce quantization errors, gain errors, and other issues, which disrupt the strict linear mapping between input and output. To drive modulators, the small signals from arbitrary waveform generators must be amplified by RF amplifiers. Under ultra-high-speed data transmission conditions, the signal spectrum may exceed the 3 dB bandwidth of the amplifier. Although the amplifier may still provide some gain to high-frequency components, the gain is no longer flat and varies significantly with frequency. At this point, the frequency-dependent gain and phase characteristics cause differences in gain amplitude and phase across the spectral components, manifesting as gain ripples and phase mismatches. In an ideal linear system, such distortions could be considered as frequency-selective filtering effects and described using linear filtering theory. However, in practical high-speed amplifiers, factors such as device structure, material properties, parasitic, and non-ideal behavior under large-signal conditions render the frequency-dependent distortions more complex than simple linear filtering. Instead, these frequency-dependent gain and phase variations often couple with amplitude compression, harmonic generation, and other nonlinear distortion mechanisms. On the receiving side, after the optical signal is converted to an electrical current and subsequently to a voltage via a transimpedance amplifier (TIA), operating at or beyond its designed bandwidth causes its gain and phase response to deviate from linearity and stability. Also, the level-dependent skew and noise which causes the differences in optimum sampling time for the different level of the eye, power fading effect of the fiber, receiver side signal-to-signal beating interference are also important parts of the nonlinear distortion source. These electrical and optical nonlinearities will be a serious problem for high-speed and advanced multi-level modulation formats like PAM-4 and PAM-8. Consequently, the signal conversion process deviates from an ideal linear mapping and tends toward nonlinear distortion. High-speed data transmission systems typically include multiple stages. Even if each individual component's nonlinearity is weak, the cumulative effect of multiple cascaded stages leads to significant waveform deviations at the final output.

Within the conventional digital signal processing (DSP) methods, researchers often rely on predefined models or limited approximations to simplify the description of system characteristics. However, when confronted with the highly coupled nonlinear and time-varying effects described above, these methods are prone to model deviations, insufficient compensation, and degraded performance. In contrast, neural network-based equalizers demonstrate stronger adaptability in dealing with complex nonlinear coupling scenarios. Their advantage does not stem from a single mechanism, but rather from the capability of deep neural networks to serve as universal function approximators. They can automatically learn high-dimensional, nonlinear input-output mappings from large-scale training data [*Nature* 521, 436–444 (2015)]. For nonlinear distortions such as the plasma dispersion effect in silicon MZMs, gain ripples under bandwidth limitations, and the gain compression and harmonic generation in electrical devices, traditional equalizers require separate mathematical models for correction. Neural networks, however, do not require explicit analytical models of these processes. During the training phase, the network parameters are iteratively optimized (e.g., via gradient descent). Consequently, when system conditions—such as temperature, biasing, drive voltage, or transmission distance—change, the neural network-based equalizer can be retrained or adapted online with new data conditions, without the need to reconstruct any mathematical models [*PNAS*, 114, 3521–3526 (2017)]. From an engineering implementation perspective, neural network-based equalizers are more feasible thanks to mature deep learning frameworks, such as PyTorch [*NeurIPS* 33, 1 – 12 (2019)] and efficient hardware accelerators

[*Proceedings of the IEEE 96, 879–899 (2008)*]. Neural network methods are now embedded in modern hardware acceleration ecosystems, offering unified programming interfaces and optimization tools. This ecological advantage enables researchers to focus on data and network architecture design, rather than low-level implementation details, thus expediting technology iteration and solution validation.

We fully understand the reviewer’s intention in using SFDR as a quantitative indicator to measure the improvement of neural network-based equalizers over conventional DSP approaches. However, the transmission system environment in this study is more complex, encompassing variations in multiple system parameters that collectively influence overall transmission performance. Given this complexity, we have chosen the bit error rate (BER) as the core performance metric, which is widely accepted in the optical communications field as a terminal quality indicator that more directly reflects the effectiveness of the receiver-side recovery algorithm. In this reply, we have supplemented additional BER data under varying transmission distances (new Figure 6) and input optical powers (new Figure S11). By observing the BER achieved by the AI equalizers, we have demonstrated its capability in handling generalized nonlinear problems, thereby highlighting the value and advantages of AI equalization.

We hope the discussion above can clear the reviewer’s concerns.

Comment 2: *“The output values following the GRU equalizer should be explicitly stated as either discrete or continuous. Comparing them to SD-FEC thresholds may be meaningless if they are discrete.”*

Our reply: Thanks for the comment. For the received signal at the receiver, before sending to the decoder for analog to digital conversion and making symbol decision, it is first equalized by the GRU equalizer, so the output values following the GRU equalizer is continuous and with channel noise added.

Let X be a discrete random variable that represents the sequence of transmitted symbols with an alphabet X consisting of M discrete symbols, i.e. $X = \{x_1, \dots, x_M\}$, and let R be also a discrete random variable that represents the sequence of received samples. An AWGN (additive white gaussian noise) channel has the form: $R = X + Z$, where Z is a complex Gaussian-distributed random variable with zero mean and total variance σ^2 , $Z \sim N(0, \sigma^2)$. Consider a neural network with parameters ϑ , with input R and output $Y = f(R; \vartheta)$, where f represents the neural network and Y is the estimate of the transmitted sequence X . An illustration of this setup is shown in Figure R3-1 (details can be found in [*J. Light. Technol.*,42(20), 7104-7115 (2024)]), where the equalized signal Y is the demapper input and $Q_{X|Y}(\cdot|Y)$ is the soft-information in the form of a posterior distribution that feeds the SD-FEC. A soft-decision provides a finer, more granular indication on whether the incoming signal really is a “1” or a “0” bit compared to hard-decision. Soft-decision use additional soft-decision bits, or “confidence” bits, to indicate how far above or below a received bit is from a threshold. Due to the probabilistic nature of communication systems, bits that sometimes fall very near a threshold, either slightly above or below, can be misinterpreted and defined incorrectly. As demonstrated in references [*J. Light. Technol.*,42(20), 7104-7115 (2024)], [*IEEE J. Sel. Top. Quant. Electron.*,28(4), 1-23 (2022)] and numerous other studies [*IEEE Photon. Technol. Lett.*, 27(4), 387-390 (2015)], [*Opt. Exp.*, 29(7), 11254-11267 (2021)], [*J. Light. Technol.*, 40(8), 2427-2434 (2022)] training neural networks using the mean squared error (MSE) loss function alters the noise distribution, deviating from a Gaussian profile, which degrades SD-FEC performance (generally SD-FEC calculates probabilities based on Gaussian distribution) and results in 'jail window' or 'MSE-grid scatterplot' effect [*J. Light. Technol.*,42(20), 7104-7115 (2024)], [*IEEE J. Sel. Top. Quant. Electron.*, 28(4), 1-23 (2022)]. As noted by the reviewer, SD-FEC becomes ineffective in such

scenarios. In academia, various methods have been proposed to solve the problem, including the use of loss functions such as generalized mutual information (GMI), L^2 regularization, maximum achievable information rate (AIR), MSE-X, and others.

Figure R3-1. Classical transceiver model with a neural network (NN) based nonlinear equalizer in an AWGN channel represented by Z.

In this work, as illustrated in Fig. 4-7, the implementation of the GRU equalizer achieves BER performance that meets the HD-FEC threshold. To ensure a more accurate and rigorous analysis, the results are benchmarked against the HD-FEC threshold in the evaluation of the key findings. Furthermore, to avoid potential misunderstandings, we have deleted all the descriptions and comparisons of SD-FEC threshold in the whole manuscript, including the BERs for all equalizers (DNN, bi-GRU and T-biGRU), as well as all the SD-FEC threshold lines in Figure 3-7.

We hope the above analysis could make it clearer.

Comment 3: *“In real transmission scenarios, transmitter output may need adjustment based on line conditions. If output intensity varies, would DNN equalizers require retraining? If so, indicating the time required for retraining would be informative.”*

Our reply: Thanks for the comments. We fully agree with the reviewer that the transmitter output power needs adjustment based on the system conditions. In real transmission scenarios, if the transmitter output intensity varies due to changes in line conditions (e.g., power fluctuations, modulation depth variations, or environmental factors), a neural network equalizer may require retraining or adaptation. Variations in output intensity can affect the signal-to-noise ratio (SNR), modulation depth, and potentially the signal shape. This can change the distribution of the transmitted symbols, which the neural network was initially trained on. If the model encounters a signal that deviates from the training distribution (e.g., due to lower intensity or clipping), its performance could degrade, as it may not effectively map the received signal to the correct symbols. We have conducted further experiments to verify this.

By adjusting the output power after the modulator through variable optical attenuator (VOA), the impact of transmitter output intensity variation is evaluated, the results are shown in Figure S11. The results indicate that variations in transmitter output intensity have an obvious impact on system performance. Compared to retraining the network at each specific output power, utilizing a model trained at a fixed power level (e.g., -13.94 dBm in Figure S11a and -15.94 dBm in Figure S11b) to evaluate the performance at other power levels results in a degradation in performance. Meanwhile, as shown in the results, using the model trained at -13.94 dBm and -15.94 dBm to test its corresponding data, it shows favorable performance.

Figure S11. Impact of optical power variations to system performance processed. (a) The model trained at a fixed power of -13.94 dBm. (b) The model trained at a fixed power of -15.94 dBm.

On the other hand, continuous or online learning can be used to adapt the neural network in real time to changing conditions without needing retraining. This allows the equalizer to update its parameters incrementally as new data (with adjusted intensity) becomes available. We have already added a new part named as “Retraining frequency of the neural network” in Supplementary Section 5 to demonstrate the retraining requirements including the system component replacements for ANN equalizers.

(Supplementary Section 5) When optical or electronic components such as photodetectors, electrical/optical amplifiers, or modulators are exchanged in real network scenario, the learning process of the neural network must typically be repeated. This is because such changes can significantly alter the signal characteristics due to the differentiated device response, rendering the previously trained model ineffective. As optical modulators or other components age, their performance metrics including insertion loss, bandwidth, or nonlinearity may degrade over time. This gradual change requires periodic retraining to adapt to the new system state. If the modulator's bias drifts due to temperature fluctuations or voltage instability, the neural network model may need to be retrained to compensate for the resulting signal distortion. In this case, retraining is needed when the drift exceeds the model's tolerance level. Changes in the noise environment, such as increased crosstalk or interference, might necessitate model updates. The retraining frequency depends on how fast the components degrade or how often they are replaced. For systems with stable components, retraining might only be required on a seasonal or annual basis, while systems with frequent changes might need more regular updates.

We conduct the experiments to analyze the impact of component replacement for neural network. In the experiments, we replace the electrical driver and optical EDFA and evaluate the resulting system performance variations for different data rates. In the case of driver replacement experiment, the neural network is first trained using data collected from the system that employs driver 1 (origin driver). The trained network is then tested using data collected from the system employs driver 1 and replaced driver 2 (replaced driver) of different model. The process is repeated five times, and the average results are presented in Fig.~S10a. The results show that using the trained network model of driver 1 is not suitable for the testing of employing driver 2, there exists obvious performance degradation. The results of training the network using replaced driver 2 and testing with the same replaced driver is also presented, which is similar to the origin driver situation. For the case of EDFA replacement, the situation is similar with the driver case, as shown in Fig.~S10b. In terms of DC bias, we manually change the

modulator DC bias to simulate the bias drift, the results of 400 Gbps PAM-4 are shown in Fig.~S10c. It indicates that there is distinguished deterioration using the trained network model of 3 V bias for the testing of bias voltages at 1.5 V, 2 V and 2.5 V. Furthermore, by adjusting the power after the modulator through variable optical attenuator (VOA), the impact of output optical power variation to the system performance at 400 Gbps PAM-4 is evaluated, the results are shown in Fig.~S11. The results indicate that variations in output intensity have an obvious impact on system performance. Compared to retraining the network at each specific output power, utilizing a model trained at a fixed power level (-13.94 dBm in Fig.~S11a and -15.94 dBm in Fig.~S11b) to evaluate the performance at other power levels results in performance degradation. Meanwhile, as shown in the results, using the model trained at -13.94 dBm and -15.94 dBm to test its corresponding data, it shows favorable performance.

Response letter

We appreciate the careful review by the reviewers and have modified the manuscript in accordance with their suggestions. Here, we present a point-by-point reply (**in black**) to the reviewers' comments (**in gray**), as well as the action taken (**in red**).

Response to the report from Referee #1

Comment 1: *“You mention an extinction ratio ER of 36 dB, which is a very high value for a Mach-Zehnder modulator and cannot be achieved without a phase shift close to π . I wonder if this value indicates the on/off ratio of the output light during modulation or the maximum on/off ratio of a Mach-Zehnder interferometer with an ideal phase shift given by the thermo-optic effect or similar. The former is impossible, considering the modulation efficiency and the length of the phase shifter. Even in the latter case, this value indicates a very well-fabricated symmetrical interferometer, which is not so easy to achieve. I think this value is just a typo or the authors misunderstood the ER. Please check again and correct it. While I fundamentally believe this paper is worth publishing, such a simple misunderstanding reduces the credibility of this research and the statements in this paper. Please review all the content of this paper and its consistency.”*

Our reply: Thanks for the valuable comments. Here, the extinction ratio (ER) refers to the static ER of the modulator, which is the on/off ratio of the Mach-Zehnder interferometer (MZI) with an ideal phase shift given by TiN heaters (the right side in Figure 2(b)) through the standard thermal tuning mechanism of silicon modulators.

For the MZI architecture, maintaining a high static ER raises high requirements on the process uniformity of the two modulation arms as well as the symmetry of the interferometer. Especially for the slow-light structure, the process uniformity of two arms is even more important. Due to the existence of slow light effect, the waveguide spectra are not flat, and if the passband positions of the two-arm waveguides are offset, the passband width and extinction ratio will be significantly reduced. Taking into account the practical grating process, we controlled the feature size to be within the range that can be stably produced by commercial foundries at the design beginning, thus maximizing the consistency of the grating spectra. Compared to two-dimensional photonic crystal, the CMOS-compatibility and wafer-level production scalability of the designed one-dimensional waveguide gratings have been greatly improved. In fact, even for a complete modulator (the spectra of the two arms have already been combined by interference), the spectra of different devices on each channel are still very close, which can be seen from Figure 2(g) in the main part, so the two arms inside the modulators should be uniform as well. The favorable processability of grating waveguides proves the quality and stability of the process to a certain extent. From this point of view, the fabrication quality of the symmetrical interferometer required by the MZI architecture itself should also be guaranteed to some extent.

In fact, with the improvement of design and manufacturing level, it is common for silicon modulators to have a static ER of more than 30 dB and the indicators could already reach this level several years ago. Here, the device indicator in this work is not a very high value under the current manufacturing process nowadays (high enough is still necessary), and some specially optimized modulators can achieve much higher static ERs than this. We have organized some representative static ERs of pure silicon modulators for reference, including Si-MZMs and Si-MRMs, which are shown in Table R1 below.

Table R1. Comparison of static ERs of silicon modulators

Ref.	Year	Type	Static ER (dB)
Opt. Exp. , 21(20), 23410-23415 (2013)	2013	Si-MZM	35
Opt. Commun. 340, 107-109 (2015)	2015	Si-MZM	30
Chin. Opt. Lett. , 15(4), 042501 (2017)	2017	Si-MZM	30
Opt. Exp. , 25(10), 11254-11264 (2017)	2017	Si-MZM	65
J. Light. Technol. , 41(15), 5059-5066 (2023)	2023	Si-MZM	30
Optica , 12(2), 203-215 (2025)	2025	Si-MZM	35
Optica , 3(6), 622-627 (2016)	2016	Si-MRM	35
Opt. Commun. , 430, 131-138 (2019)	2019	Si-MRM	37
Photon. Res. , 10(4), 1127-1133 (2022)	2022	Si-MRM	30
Opt. Exp. , 30(14), 25672-25684 (2022)	2022	Si-MRM	55
Nat. Commun. , 14(1), 7409 (2023)	2023	Si-MRM	68

We hope the above analysis could make it clearer. To avoid potential misunderstanding, we have specifically emphasized the static ER and the tuning method in the main part. Also, thank you for your kind reminder, we have carefully checked the whole manuscript, to ensure the consistency of the statements and content.

(Main part, line 202) Moreover, by embedding CROW into the modulation arms, the advantages of MZI can be leveraged to maintain a sufficient static extinction ratio (ER) while reducing the device footprint. Through the standard thermal tuning mechanism by TiN heaters, the static ER of the compact modulator is measured to be 36 dB, provided by the resonator-assisted MZI architecture.

Comment 2: “Regarding the driving voltage, 5 V_{pp} is a large value and very power-hungry. Please comment on future prospects for reducing power consumption.”

Our reply: Thanks for the reviewer’s comments. In fact, the slow-light design model is a reconfigurable approach for different application orientations including high-speed or high-efficiency. By adjusting the slow-light structure parameters such as N_p and N_r , especially N_p , the slow-light effect (group index) will increase significantly, which will lead to a narrow optical passband and higher modulation efficiency, reducing the requirements for driving voltages. More details about the performance trend with N_p and N_r can be found in Supplementary Section 2. Here, we summarize three representative groups of N_p and N_r to show the parameter change more intuitively in Table R2 below. For N_p20 N_r20 and N_p40 N_r10 , at almost the same length of 400 periods, although the optical bandwidth drops from 94 GHz to 53 GHz, the group index increases from 6.1 to 21.5, which leads to a leap of efficiency factor about 6 times. In this case, the modulation efficiency of the Si-SLMs will also be enhanced by 6 times, from 0.82 V·cm to 0.13 V·cm (a very high value for silicon modulators), with an expected V_π only around 5 V (the lengths corresponding to the two sets of parameters are almost the same). Therefore, considering that the ~30% V_π swing is basically enough, 1-2 V_{pp} should be available for driving signals in this design, which is

already at the CMOS-compatible level. Although the optical bandwidth will be limited to 53 GHz indeed in this situation, this method can still be applicable to the scenarios that require low drive voltage rather than ultrahigh transmission speed.

Table R2. The parameters for different optical structures

Structure	N_p20 N_r10	N_p20 N_r20	N_p40 N_r10
Optical bandwidth (GHz)	148	95	53
Q factor	1307	2100	3654
Efficiency factor (π/V)	0.013	0.028	0.173
Group index	6.1	6.1	21.5
Modulation length (μm)	124	249	244

For the system level, it is important to reiterate that our experiments are still at an early proof-of-concept stage. The energy consumption of different components in the system can be reduced in future optimizations. With future integration of on-chip frequency combs in our previous research [*Nature* 605, 457 – 463 (2022)], the light source can be changed to integrated on-chip light source. For optical waveguide amplifiers, with integrated erbium-doped amplifiers [*Science* 376, 1309 – 1313 (2022)], and on-chip semiconductor optical amplifiers (SOAs) [*IEEE J. Sel. Top. Quant. Electron.* 22, 78 – 88 (2016)], the currently used discrete EDFA could be replaced by fully integrated photonic devices. For the algorithm level, the introduction of AI methods will indeed increase additional power, and the high energy consumption of AI models is still a recognized common problem in the AI research field that needs to be optimized. For the AI equalizer design, there exist a variety of mature neural network compression and acceleration techniques that can substantially improve energy efficiency. Pruning approaches [*NIPS* 28, 1135 – 1143 (2015)] remove redundant parameters to reduce memory and input/output overhead, thereby lowering memory and data bus power consumption. Quantization methods [*ECCV* 9908, 525 – 542 (2016)] replace floating-point logic with lower-bit-width fixed-point or binary arithmetic, fundamentally reducing multiply-accumulate operations and associated data loading energy at the circuit level. Distillation strategies [*ICML* 80, 1607 – 1616 (2018)] further transfer knowledge from large teacher networks to smaller student models, decreasing overall complexity and thus power usage. In addition to these algorithmic enhancements, leveraging GPU-optimized libraries such as NVIDIA’s cuDNN [*arXiv:1410.0759* (2014)] can improve computational efficiency. Techniques like kernel fusion, as enabled by frameworks such as TVM [*OSDI* 13, 578 – 594 (2018)], reduce redundant memory accesses lowers both compute and energy requirements. By judiciously combining these hardware and software optimizations with pruning, quantization, and distillation strategies, it is possible to achieve lower-power, higher-efficiency AI equalizers suitable for future implementations. Taken together, these optimization methods promise lower power consumption than currently measured in our experiment.

Furthermore, based on the previous power consumption discussion, we evaluated and compared the power consumption of traditional nonlinear algorithm and AI equalizers to balance the relationship between performance and consumption, using the same method as described in the previous response letter (HWINFO [<https://www.hwinfo.com/>]), and based on our previous research [*Nat. Commun.*, 14(66), (2023)]. From the calculation, it can be seen that under the circumstance that the algorithm is

already adopted, the AI equalizers (especially bi-GRU and T-biGRU) have not significantly increased the power consumption per bit. This is because the increase in power consumption also brings about an enhancement in the transmission rate, so that the algorithm energy consumption required to transmit each bit of data is controlled within an acceptable range.

We hope the above analysis could make it clearer. And we have added the new part of discussion for power consumption in Supplementary Section 8.

(Supplementary Section 8) To evaluate the optical energy consumption level of the modulator (excluding systems), we have calculated the energy consumption per bit of the optical modulator for both PAM-4 and PAM-8 transmission. For PAM signals, the energy consumption per bit can be calculated using the equation below^{11,12}:

$$E_b = \frac{CV_{pp}^2}{M^2 \log_2(M)} \sum_{i=0}^{M-1} (M-i) \left(\frac{i}{M-1}\right)^2$$

in which M is the modulation level, C is the junction capacitance and V_{pp} is the driving voltage.

In the device design, the relatively low doping concentration of the PN junction leads to a smaller junction capacitance. Under the bias voltage of 3 V, the designed PN junction model illustrates a junction capacitance of 1.5 pF/cm, which gives the total capacitance of 37.2 fF in a device length of 249 μm . Therefore, under a dual-drive differential signal with a 5-V V_{pp} for each modulation arm, the energy consumption per bit is calculated to be 129.1 fJ/bit for PAM-4 and 66.4 fJ/bit for PAM-8. Although the relatively large driving voltage together with the dual-drive push-pull configuration increase the energy consumption to some extent, the low doping and compact footprint lead to a small junction capacitance, thus controlling the energy consumption to a favorable level.

For the practical system, in this proof-of-concept demonstration, we evaluated the power consumption of each component involved in the experiment. Although not optimized for energy efficiency, this provides a reference scenario that can be used as a benchmark for future integration and optimization designs. In the experiment, the bench-top laser source provides the optical carrier signal with the output power of 13 dBm. The maximum apparent power consumption of this laser is 100 VA, with real power consumption not exceeding 100 W. For the modulation part, the main power consumption includes the DC bias, TiN heater, and drivers. The recorded voltage and current values in the experiment process allowed us to calculate the actual system power consumption. For DC bias, two modulation arms were given a DC bias of 2.5 V each with the measured currents of 0.077 A and 0.080 A, resulting in a power consumption of 0.393 W. To ensure that the modulator operates at the quadrature point, one TiN heater was driven at 2.290 V and 0.011 A, thereby the corresponding power consumption was 0.025 W. For the electrical amplifiers, two drivers were connected in parallel to the same power supply channel. The measured operating voltage and current were 9 V and 0.531 A, thus the combined power consumption of drivers was 4.779 W. It should be noted that we did not employ additional thermoelectric cooling (TEC) for the modulator due to the intrinsic thermal stability of the Si-SLM. For the optical amplifier, a discrete erbium-doped fiber amplifier (EDFA) was placed before the photodetector (PD) to enhance the input optical power. The conventional discrete EDFA with similar gain and noise has a maximum power consumption of about 20 VA, with real power not exceeding 20 W. For simplicity and comparative purposes, we set the EDFA power consumption to approximately 20

W. For the AI equalizers, referring to the prior measurement work¹³, the power consumption was monitored using HWINFO, ranging from ~35 W for the DNN to ~67 W for biGRU and T-biGRU implementations. Summing these contributions, the total system power consumption varied from approximately 159.887 W to 192.757 W, depending on the chosen AI equalizer. And all these values are listed in Table S1.

Furthermore, we evaluated and compared the power consumption of traditional nonlinear algorithm and AI equalizers. By employing VNLE, we achieve a compatible BER below HD-FEC threshold at 130 Gbps PAM-4. Using the same method as described above for AI equalizers, The power consumption was measured to be 42.39 W during the VNLE algorithm running. Here we define the equalizer power consumption per bit *EPCpB* as:

$$EPCpB = \text{Power Consumption (W)} / \text{Data Rate (Gbps)}$$

A smaller *EPCpB* value indicates a lower power consumption per bit. Based on the power consumption of each equalizer, the calculated *EPCpB* for VNLE, DNN, bi-GRU, T-biGRU are shown in Table S2, where the speed selections are all below HD-FEC threshold. From the calculation results, it can be seen that the speed reduction of employing VNLE does not justify the decrease in energy usage. Simultaneously, under the circumstance that the algorithm is already adopted, the AI equalizers (especially bi-GRU and T-biGRU) have not significantly increased *EPCpB* (and even reduce it). This is because the increase in power consumption also brings about a significant increase in the transmission rate, so that the energy consumption of algorithm required to transmit each bit of data is controlled within an acceptable range. In other words, it is worthwhile to use higher total power consumption in exchange for a significant increase in transmission speed based on AI equalizers.

It is important to reiterate that our experiments are still at an early proof-of-concept stage, aimed at demonstrating the potential of Si-SLMs for ultra-high-speed, large-throughput signal transmission. It is anticipated that ongoing advancements, such as on-chip frequency combs¹⁴, integrated optical amplifiers^{15,16}, and monolithic integration of light sources¹⁷, modulators, and drivers¹⁸, will substantially reduce these indicators. Likewise, techniques like network pruning¹⁹, quantization²⁰, and distillation²¹, as well as hardware-accelerated inference²², are expected to lower AI equalization power. Overall, as device integration and algorithmic optimization advance, the energy efficiency of Si-SLMs for ultrahigh-speed data center interconnections can be improved dramatically.

Response to the report from Referee #3

General comments: *“The authors have addressed all of the reviewers' comments. Furthermore, they have proposed T-biGRU, which enhances the performance of the system. From the beginning, this paper demonstrated excellent results in terms of transmission characteristics, and I consider it to be an outstanding work. Therefore, it would be appropriate to submit it to a suitable journal.”*

Our reply: Thanks for the reviewer's recognition. We have made a lot of revisions and additional works based on the reviewers' comments last time and this time. To further address the raised questions, we made the point-by-point response as follows.

Comment 1: *“If the authors aim for publication in Nature Communications (NCOMMS), a more detailed analysis of the ANN equalizer would further enhance the paper. For example, it would be beneficial to specify the extent of variation in the Si-SLM characteristics that the ANN equalizer can compensate for, along with the range of its compensation capabilities.”*

Our reply: Thanks for the valuable comment. The AI equalizer is used at the back end of the system, so it is a full-link nonlinear distortion optimization for both optical and electrical devices. For the modulator nonlinearity, silicon modulators employ plasma dispersion effects by forming PN junctions which is intrinsically nonlinear. During ultrahigh-speed modulation, carriers cannot fully recover to their steady state before the next cycle, causing the accumulation of nonlinear effects. Also, the Mach-Zehnder modulator inherently exhibits a cosine transfer function, which will also introduce nonlinearities, deviating the output from a linear mapping of the input and thus degrading the modulation performance. For the system, the primary electrical nonlinear sources originate from RF components, attributed to the inherent nonlinear behavior of charge transport at the transistor junction which is predominantly observed in the transistor driver. And nonlinear effects such as amplitude-to-phase conversion, in-band and base-band memory effects in amplifiers, the constrained operation range of the analog-to-digital converter introduced nonlinear vibrations cannot be disregarded. Quantitatively varying the characteristics of system components to achieve a specific value of nonlinear change is challenging. However, the distortions suffered by signals at different baud rates can serve as an indicator of the different nonlinear distortions encountered. Based on this consideration, the system performance for different baud rate signals under different AI equalizer network configurations is evaluated, the results are shown in Figure S11.

The BER performance of the DNN equalizer initially improves for all data rates as the number of input neurons increases, as shown in Figure 3(g). However, after reaching a certain threshold, specially 25 neurons for our network, further increases do not yield additional performance gains. Meanwhile, as the number of neurons in the hidden layer increases, the BER performance does not exhibit significant (exponential-order) promotion, as depicted in Fig. 3(h). Here, Figure S11(a) illustrates the performance variation with different numbers of hidden layers. The results indicate that when the number of layers exceeds four, performance saturation occurs. Since a larger network scale does not improve system performance significantly and consumes more computational power, two-hidden-layer structure is employed in the experiment.

For the GRU equalizer, with more GRU units employed in each direction of the equalizer (see Figure 4(b)), the BER will be better, as shown in Figure S11(b). For both the cases of 300 Gbps and 400 Gbps signal, increasing the number of GRU units could effectively reduce BER. This trend remains evident up

to 230 units, beyond which performance saturation occurs, with no significant further improvement. Another dimension is to increase the number of GRU directions in the equalizer. By further extending the T-biGRU to a dual-biGRU, which incorporates two forward and two backward directions, further performance improvements can be achieved as given in Figure S11(c). However, it will result in a power consumption increase. For the same 400 Gbps signal with BERs below HD-FEC threshold, the power consumption of T-biGRU and dual-biGRU under the same hardware configuration is 67.56 W and 73.2 W, respectively. Meanwhile, the power consumption increases from 66.57 W to 67.56 W when transitioning from bi-GRU to T-biGRU, as previously mentioned in the response letter.

Figure S11. System performance under different AI equalizer configurations. (a) number of hidden layers in DNN; (b) number of GRU units in hidden layer; (c) number of GRU directions.

We hope the reasons above can clear the reviewer’s concerns. We have also added the new Supplementary Section 5 to describe the AI equalizer network configurations comprehensively.

(Supplementary Section 5) The AI equalizer is used at the back end of the system, and the distortions suffered by signals at different baud rates can serve as an indicator of the different nonlinear distortions encountered. Based on this consideration, the system performance for different baud rate signals under different AI equalizer network configurations is evaluated.

The core advantage of the GRU algorithm is its ability to effectively capture long-range dependencies in sequential data. For the GRU equalizers, models with bidirectional structure are capable of learning information from both preceding and following data when processing the current data. The first layer is the input layer, the current symbol x_i with its k preceding and k succeeding symbols together are used as the input sequence of the network. In this work, to accurately model temporal dynamics and efficiently process the sequential information, we configured the input layer's symbol count to match the maximum data size captured in each instance. Specifically, in the experiment each capture consists of 32767 signal symbols, corresponding to a total data size of 4.2 MB. Consequently, the input layer's symbol count is set to 32767, with k equal to 16383. When k is set to smaller values, such as 10000 and 5000, the performance is also evaluated, as shown in Fig.S10. The results indicate that these cases exhibit inferior performance compared to $k=16383$, which may be attributed to insufficient sequential information in the data stream. The value of k is determined by the number of symbols the equalizer can process. To retain more sequential information, larger k values are preferred.

The number of neurons and the number of layers are important factors affecting the performance of neural networks. The BER performance of the DNN equalizer initially improves for all data rates as the number of input neurons increases (Fig. 3g). However, after reaching a certain threshold, specially 25 neurons for our network, further increases do not yield additional performance gains. Meanwhile, as

the number of neurons in the hidden layer increases, the BER performance does not exhibit significant (exponential-order) promotion (Fig. 3h). Furthermore, Fig. S11a illustrates the performance variation with different numbers of hidden layers. The results indicate that when the number of layers exceeds four, performance saturation occurs. Since a larger network scale does not improve system performance significantly and consumes more computational power, two-hidden-layer structure is employed in the experiment. For the GRU equalizer, with more GRU units employed in each direction of the equalizer (Fig. 4b), the BER will be better, as shown in Fig. S11b. For both the cases of 300 Gbps and 400 Gbps signal, increasing the number of GRU units could effectively reduce BER. This trend remains evident up to 230 units, beyond which performance saturation occurs, with no significant further improvement. Another dimension is to increase the number of GRU directions in the equalizer. By further extending the T-biGRU to a dual-biGRU, which incorporates two forward and two backward directions, further performance improvements can be achieved as given in Fig. S11c.

Comment 2: *“Additionally, the Bi-GRU NLE incorporates k preceding and k succeeding symbols in its input layer. It would be helpful to clarify how the value of k is determined as a hyperparameter and to provide some degree of physical interpretation for it.”*

Our reply: Thanks for the comment. For the GRU algorithm, models with bidirectional structure are capable of learning information from both preceding and following data when processing the current data. The bi-GRU model comprises two unidirectional GRU layers operating in opposite directions. One GRU processes the input sequence in the forward direction, starting from the beginning, while the other operates in reverse, processing the sequence from the end toward the start. By combining forward and backward GRU processing, the model incorporates information from both the future and the past to influence its current states. For the T-biGRU, the first layer of the T-biGRU equalizer is the input layer, the current symbol x_i with its k preceding and k succeeding symbols together are used as the input sequence of the network, which is the same as the bi-GRU. The T-biGRU model is determined based on the state of three GRU layers. The first GRU layer processes the data in a forward direction, starting from the beginning of the sequence. The second GRU layer processes data in a backward direction, starting from the end of the sequence, to capture reverse temporal dependencies. The third GRU layer, like the first, processes data in a forward direction from the beginning of the sequence. The hidden state encapsulates the flow of symbolic information across recurrent time steps, ensuring continuity and context throughout the sequence. The bi-GRU model relies on the states of two GRUs, whereas the T-biGRU model utilizes the states of three GRUs. By integrating forward, backward, and repeated forward GRU processing, the T-biGRU model more comprehensively extracts both global and local features of the sequence, thereby further enhancing equalization performance. The core advantage of the GRU is its ability to effectively capture long-range dependencies in sequential data. Therefore, in this work, to accurately model temporal dynamics and efficiently process the sequential information, we configured the input layer's symbol count to match the maximum data size captured in each instance. Specifically, in the experiment each capture consists of 32,767 signal symbols, corresponding to a total data size of 4.2 MB. Consequently, the input layer's symbol count is set to 32,767, with k equal to 16,383. When k is set to smaller values, such as 10,000 and 5,000, the performance is also evaluated, as shown in Figure S10 below. The results indicate that these cases exhibit inferior performance compared to $k=16,383$, which may be attributed to the insufficient sequential information in the data stream. The value of k is determined by the number of symbols the equalizer can process. To retain more sequential information, larger k values are preferred. The content about k values is added to the Supplementary Section 5, which

is shown in the response to Comment 1.

Figure S10. System performance of employing bi-GRU and T-biGRU with different k values

Comment 3: *“On the other hand, adding substantial new elements may cause the paper to lose focus, making its core contributions harder to convey. Thus, maintaining simplicity is advisable. Submitting to journals other than NCOMMS is also a valid option worth considering.”*

Our reply: Thanks for the suggestion. In order to support our research with sufficient and detailed works, a lot of new elements have indeed been added to the manuscript. Therefore, to maintain simplicity, we have added nearly all the new content this time to the Supplementary Information, and have carefully organized and arranged the sections, trying our best to present the substantial content clearly and systematically. Meanwhile, since the self-adaptive characteristics of AI equalizers help mitigate the impact of nonlinearity in Si-SLMs, this approach further enhances the performance of the slow light technology, which is currently the only viable method to achieve bandwidths around 100 GHz in silicon photonics. At present, heterogeneous integration (such as LNOI, III-V, etc.) is the main technology route for 400 Gbps per lane, and whether pure silicon modulators can achieve 400 Gbps has been elusive so far. Our solution can achieve the rate leap on a low-cost pure silicon platform without introducing complex processes and materials. On the other aspect, our work is also a reflection of the cross-disciplinary research. These years an increasing number of works utilize AI technology to accelerate the discovery in sciences (AI4S). From a research perspective, the combination of AI and communications is a new research direction, and accelerating traditional silicon photonics to 400 Gbps per lane is a good example of “AI for SiP”. In fact, by demonstrating the capabilities of AI acceleration and its compatibility with silicon photonics, we hope that our work can inspire more “AI for SiP” following-up studies to both the AI and photonic communities in different application orientations (e.g. sensing, imaging, computing, reverse design, etc.). Therefore, a comprehensive journal represented by *NCOMMS* may be more appropriate because of its broad and interdisciplinary audience.

We hope the above discussion could clear the reviewer’s concerns.

Response to the report from Referee #4

General comments: *“This paper titled “Exploring 400 Gbps/λ and beyond with AI-driven silicon photonic slow-light technology” by Han et.al. presents a high-speed silicon modulator with slow-light effects together with a learned equalizer for signal correction. It demonstrates over 400 Gbps per channel transmission with PAM-4. By leveraging AI-based equalization, the system compensates for nonlinear distortions, achieving bit error rates below HD-FEC thresholds.*

The approach presented can offer a scalable path for high-speed optical interconnects. However, despite the general presentation quality, the paper has a critical shortcoming. First of all, the modulator design is similar to prior work; and its performance does not surpass state-of-the-art silicon modulators. Secondly, the AI-optimized equalizer’s novelty over those of existing equalization techniques is not clearly demonstrated. The paper also is missing sufficient detail on training, computational overhead, robustness, and stability of the equalizer in general. After the equalizer’s novelty is clearly demonstrated through an extensive revision, the paper may then be suitable for publication in Nature Communications. Below are my detailed comments:”

Our reply: Thanks for the reviewer’s comments. We appreciate the reviewer’s recognition. And we apologize for not explaining the innovation of our work fully in the initial manuscript. Here, we further summarize the important innovations of our research below.

About the modulator design:

For the modulator design, we adopted the silicon slow-light modulator (Si-SLM) approach based on coupled-resonator optical waveguide (CROW) structure on SOI platform. In fact, nowadays the current modulator designs face the dilemma of choosing between resonant and non-resonant schemes: the conventional silicon Mach-Zehnder modulators (Si-MZMs) are hindered by high-power consumption and large device footprint, the modulation arms are usually a few millimeters long, due to the low modulation efficiency of the depletion-mode phase shifter; the silicon microring modulators (Si-MRMs) hold more compact footprint, but have narrower optical bandwidth and are more sensitive to operating temperature, which requires additional power budget for temperature control. Regarding the above issues, the slow-light approach can improve both bandwidth and modulation efficiency simultaneously, while maintaining a wide optical passband for multi-wavelength applications and high thermal stability. Specific indicator comparisons of our designed Si-SLMs with Si-MZMs and Si-MRMs are given in response to Comment 1 below in detail. From the design perspective, the slow-light device model is a reconfigurable solution with a fairly high degree of design freedom for designing modulators, which means that we can make targeted and effective changes to device performance by adjusting the design. For instance, if the design goal is to push the bandwidth to the limit to achieve ultra-low-cost OOK transmission, fewer resonators are required, and our previous work [*Sci. Adv.* 9(42), eadi5339, 2023] demonstrated the feasibility of using slow light to increase bandwidth significantly to 110 GHz. Here, in this work, for further achieving deeper modulation depth and better ultrahigh-speed multi-level signal separation, we redesigned the structure comprehensively, and focused on designing more resonators to achieve a balance between bandwidth and efficiency. At present, heterogeneous integration (such as LNOI, III-V, etc.) is the main technology route for 400 Gbps per lane, and whether pure silicon modulators can achieve 400 Gbps has been elusive so far. Since the AI equalizer can help compensate for nonlinear distortions caused by insufficient linearity of pure silicon material, this approach further enhances the performance of the slow-light technology, which is currently the only viable method to

achieve around 100 GHz bandwidths in silicon photonics.

About the AI equalizer's novelty in this work:

For the equalizers, as we mentioned in the introduction of the manuscript, signal equalization is essential for the transmission utilizing high-order modulation formats such as PAM-4, PAM-8, and mQAM. In most transmission scenarios, linear equalizers such as feed-forward equalization (FFE) and decision-feedback equalization (DFE) are usually adopted and effectively mitigate linear distortion. However, in the process of compensating nonlinear distortion including nonlinear modulation and chirp characteristics, great challenges have occurred for the above conventional equalization methods. Especially, artificial neural network (ANN) equalizers have been proposed to construct a complex map with nonlinear boundaries between the input and output spaces. ANN equalizers exhibit potency in estimating parameters within a noisy environment and recognizing complex mappings with nonlinear boundaries between input and output data, it views the equalization as multi-class problems and provides significantly better performances than the conventional linear and nonlinear equalizers. By training on large datasets, neural networks can better capture the underlying distribution of the optical signal and learn the subtle relationships between different symbols, this enables them to effectively handle cases where traditional equalizers might fail. Neural networks can learn to directly map the distorted signal back to the transmitted symbols without manually decomposing the problem into individual components. This simplifies the process and often leads to better performance because the network learns an optimal transformation of the received signal, considering all distortions jointly. Compared to conventional equalizers, ANN models the equalization as multi-level problems to mitigate the nonlinear distortions and can provide effective compensation solutions.

Importantly, ANN equalizers are particularly favorable for optical transmission based on Si-SLMs. Although the bandwidth-efficiency trade-off can be optimized effectively by introducing slow-light resonators, the physical principle of Si-SLMs is still the nonlinear plasma dispersion effect which is intrinsically nonlinear. Simultaneously, although the silicon waveguide grating structure has improved the process stability compared to two-dimensional photonic crystals, the unavoidable structure fluctuations of gratings still exist. Therefore, Si-SLMs require an adaptive technical approach to minimize the difference caused by process errors. Through AI equalization at the back end, the process limitations of the doped-silicon slow-light structure can be eliminated on the same wafer, making the deployment of large-scale Si-SLM array possible.

In addition, the AI equalizer is a full-link nonlinear distortion optimization for both optical and electrical devices, not only compensating for the insufficient linearity of the modulator. The optical and electrical nonlinearities will be a problem for multi-level formats like PAM-4 and PAM-8. Compensating for these nonlinear distortions poses a challenge for effective compensation. Digital pre-distortion is one of the frequently employed methods, but pre-compensation algorithms pose complexity in implementation as they necessitate back-propagated signals to determine the nonlinear filter coefficients in the transmitters. Other channel impairments compensation methods such as Volterra equalizer suffered imperfect performance. The system performances are further evaluated for AI equalizers compared to that of other linear and nonlinear equalizers (decision feedback equalizer (DFE) and Volterra nonlinear equalizer (VNLE)) at different baud rates, as shown in Figure S13. The results show that both the VNLE, bi-GRU and T-biGRU outperform the linear DFE equalizer that is based on least mean squares (LMS) algorithm, which means that nonlinear distortion occupies a significant part of the whole signal

distortion. And AI equalizers outperform VNLE, details are given in the reply to Comment 2 below.

In this paper, compared to the traditional DNN and GRU, the employed DNN and GRU in this work are modified and extended to better adapt the high signal modulation level and speed in the IM/DD system that employs Si-SLMs, as we responded in detail last time in the previous response letter, and we have explained the algorithm novelty and added the analysis of DNN in Supplementary Section 3, as well as the experimental results of T-biGRU in main part and Supplementary Section 4.

About the training:

In the experiment, different groups (Data Group 1~5) of data with the same size (more than 30k symbols each, with 4.2MB data size) were captured from the oscilloscope and used separately for training and testing the AI equalizer. The system performance at different training data percentages in one data group, specially 30%, 50%, 80%, and 100%, is evaluated, with the results shown in Figure S12(a). The results shown in the figure are the average BERs of employing other four data group as test data. The results indicate that with 80% and 100% percentage of the training data, the system performance is optima. The results presented in Figures 3 to 7 were obtained using 80% of the training data from a single data group, while BER represents the average results from the remaining data groups. Also, by selecting training data from one data group (Data Group 1) and testing on the remaining data groups (Data Group 2-5), the performance is given in Figure S12(b). The BER results indicate favorable consistency among different data groups.

Figure S12. System performance at different training data percentages and data groups. (a) different training data percentages from one data group. (b) different data groups under the same training data.

(Supplementary Section 5) For the training of different data groups in the experiment, different groups (Data Group 1~5) of data with the same size (more than 30k symbols each, with 4.2 MB data size) were captured from the oscilloscope and used separately for training and testing the AI equalizer. The system performance at different training data percentages in one data group, specially 30%, 50%, 80%, and 100%, is evaluated, with the results shown in Fig.S12a, where the results are the average BERs of employing other four data group as test data (DNN equalizer). The results indicate that with 80% and 100% percentage of the training data, the system performance is optima. For the main part, the results presented in Fig.3 to Fig.7 were obtained using 80% of the training data from a single data group, while BER represents the average results from the remaining data groups. Also, by selecting training data from one data group (Data Group 1) and testing on the remaining data groups (Data Groups 2-5), the performance of different data groups is given in Fig.S12b (bi-GRU equalizer). The BER results indicate

favorable consistency among different data groups.

About the computational overhead:

Consider the four-layer DNN equalizer shown in this paper with n_{in} , n_H , n_{out} nodes in the input, hidden and output layer, respectively. Typically, the output layer has a fixed single node. n_{in} can be regarded as the tap number in digital filter. The primary computational burden in the DNN arises from the error propagation needed to calculate the squared error derivative of each node in all the hidden layers. Compared to the DNN, which employs a *sigmoid* function with two saturation regions, the DNN that employs multi-level *sigmoid* does not incur additional computational requirements. The computational complexity of the DNN equalizer per iteration can be characterized from the program as:

$$C_{DNN}=8n_H^2+7 n_{in} n_H +12 n_H+4 n_{in}+3n_{out}$$

The main computation complexity of one GRU layer can be described as: $C_{GRU}=3n_H(n_E+n_H)$, where n_E refers to the input size of the GRU layer, and n_H represents the number of GRU units that used in the layer. The complexity of bi-GRU and T-biGRU layer can be calculated as: $C_{bi-GRU}=2 \times 3n_H(n_E+n_H)$, $C_{T-biGRU}=3 \times 3n_H(n_E+n_H)$.

(Main part, line 601) Consider the four-layer DNN equalizer with n_{in} , n_H , n_{out} nodes in the input, hidden and output layer, respectively. Typically, the output layer has a fixed single node. Here, n_{in} can be viewed as the tap number in a digital filter. The primary computational burden in the equalizer arises from the error propagation needed to calculate the squared error derivative of each node in all the hidden layers. Compared to the DNN, which employs a *sigmoid* function with two saturation regions, the DNN that employs multi-level *sigmoid* does not incur additional computational requirements. The computational complexity of the DNN equalizer per iteration can be characterized from the program as:

$$C_{DNN}=8n_H^2+7 n_{in} n_H +12 n_H+4 n_{in}+3n_{out}$$

About the robustness:

In real transmission scenarios, if the transmitter output intensity varies due to changes in line conditions (e.g., power fluctuations, modulation depth variations, or environmental factors), a neural network equalizer may require retraining or adaptation. Variations in output intensity can affect the signal-to-noise ratio (SNR), modulation depth, and potentially the signal shape. This can change the distribution of the transmitted symbols, which the neural network was initially trained on. If the model encounters a signal that deviates from the training distribution (e.g., due to lower intensity or clipping), its performance could degrade, as it may not effectively map the received signal to the correct symbols. The results shown previous Figure S11 indicate that variations in transmitter output intensity have an obvious impact on system performance since the system belongs to IM/DD system. Compared to retraining the network at each specific power level, using a model trained at a fixed power level to evaluate performance at different power levels leads to performance degradation. Meanwhile, as shown in the results, using the model trained at a specific power level and testing it with the data at the same power yields favorable performance.

To improve the robustness of neural network-based equalization algorithms involves several key strategies: 1) Data augmentation. By introducing variations in the training dataset to enhance generalization, such as incorporating data from different power levels, signal-to-noise ratios, and nonlinear distortions; 2) Apply dropout or L_2 regularization to prevent the neural network

overestimating and encourage robustness to variations in input signals; 3) Combine neural networks with traditional signal equalization methods to enhance interpretability and robustness. Additionally, training the network with adversarial examples or utilizing transfer learning by pretraining the model on a broader dataset can further enhance its resilience to unexpected distortions. However, this process may require a significant amount of time to obtain sufficient and useful training data.

We first tried to train the DNN network with mixed data that captured at different power -15.87 dBm, -16.75 dBm, -17.78 dBm, -18.70 dBm and -19.77 dBm with the same proportion (2:2:2:2:2), and then test the signal performance of each power, the results are shown in Figure S14. The results indicate that training the network using mixing data from different powers and then testing at each power yields better performance than training at a specific power and then testing at each power (with BER at ~ 0.5 as shown in previous Figure S11). However, compared to retraining the network separately at each power level and then testing at each power, the performance remains inferior.

Figure S14. Comparison between training the DNN with mixed data from different power and retraining at each power separately.

We also measured the performance of the bi-GRU equalizer training at a specific power and then testing at each power, the results are given in Figure S15(a). The results show that there's obvious performance degradation when employing the network trained at a specific power to test the data from other power compared to retraining the equalizer at each power. However, it is still better than the case previous Figure S11, which shows the better robustness of bi-GRU to power variations compared to the DNN algorithm. The results of training the bi-GRU using mixed data captured at different power levels and then testing on individual power levels are also shown in Figure S15(b). By training the model with mixed data from different powers, the overall BER performance can be improved to some extent compared to the training based on one specific fixed power.

Figure S15. Training the bi-GRU (a) at different fixed powers. (b) with mixed data from different power. Retraining at each power separately is also shown.

On the other hand, dropout randomly deactivates a fraction of neurons during training, forcing the network to rely on different subsets of features in each training iteration, during inference (when dropout is turned off), the full network acts as an ensemble of many smaller networks, leading to improved performance and robustness. For tasks like optical signal equalization, where the input data can be affected by noise, dispersion, and nonlinear distortions, dropout helps the network learn features that are less sensitive to these variations. This prevents the model from becoming too dependent on specific neurons and memorizing the training data, improving generalization to new data. By adding dropout layer, we further give a test to the performance, the results are shown in Figure S16(a). The results indicate that incorporating dropout in the network leads to better performance compared to the case without dropout.

Figure S16. Performance of employing (a) dropout (b) L_2 regularization (c) dropout and L_2 regularization to bi-GRU.

L_2 regularization (also known as weight decay) is a technique used to prevent overestimating by penalizing large weights in a neural network. It modifies the loss function by adding a penalty term proportional to the sum of the squared weights, encouraging the model to keep weights small and thus improving generalization. We also tried employing L_2 regularization to the bi-GRU network, the results are shown in Figure S16(b). The improved performance proves the effectiveness of the method.

By applying both dropout and L_2 regularization to the bi-GRU network, the performance, as shown in Figure S16(c), demonstrates superior results compared to using either method individually, indicating further enhanced network robustness to power variations. However, the performance remains inferior compared to retraining the model at each specific power level.

We also attempted to train the network using a mixed dataset comprising different received power levels and bias voltages, followed by performance evaluation under varying received power and DC bias voltages. However, the performance was poor (with a BER at 0.1 level), possibly due to the limited capacity of the small network, which may have been insufficient to simultaneously capture the features associated with both power and bias variations.

What we want to further make clear is that beside transmission speed, power consumption is a critical factor in scenarios like data center, we hope to enhance the device and system performance by employing relatively lite AI-based equalization method within the proposed system, to achieve a balance between performance and power consumption. So, the employed network scale of the AI equalizer in this work is small, only consisting of four layers in total for the DNN and three layers for the

GRU, respectively. Normally, for neural networks, smaller network scale tends to exhibit lower robustness, which has been shown by numerous studies. Under the network configuration shown in this work, the robustness is quite challenging. By employing data mixing, dropout and $L2$ regularization, the robustness can be enhanced to some extent.

Simultaneously, during the experiment period, we captured data on the first day and again on the fifth day, and seventh day. The network was trained using data from the first day and evaluated using data from the fifth and seventh days. The results are presented in Figure R1 below. The performance remains close across different days, demonstrating the equalizer's stability over various training and testing periods.

Figure R1. The performance of the equalizer on different days.

We sincerely appreciate the reviewer's insightful comments regarding the adaptability of the neural network equalizer to dynamic channel variations in practical scenarios and the potential need for retraining. The concerns raised about model robustness and the associated costs in terms of time and resources for retraining are indeed challenges in applying deep learning techniques to signal processing in communication systems. As discussed above, we have employed methods such as data mixing, dropout, and $L2$ regularization to partially mitigate these issues. Furthermore, researchers in machine learning and optical communications are actively exploring more advanced strategies to comprehensively address this challenge. These cutting-edge studies can be broadly categorized into two main directions: firstly, enhancing the inherent robustness of the equalizer to reduce its sensitivity to environmental changes, thereby lowering the frequency of retraining; secondly, decreasing the cost associated with model adaptation to new environments, making necessary training or adjustments more efficient.

Regarding enhancing model robustness, one significant research avenue draws inspiration from advances in adversarial machine learning. For instance, Kim et al. [*IEEE Trans. Wireless Commun.* 21, 3868 – 3880 (2022)] investigated adversarial attacks and defenses for wireless signal classifiers, demonstrating that techniques like randomized smoothing can effectively improve model resilience against input perturbations. Although this study focuses on wireless communications and adversarial samples, its core principle of enhancing robustness, making the model less sensitive to variations through noise injection during training or testing, offers a potential pathway to improve the tolerance of optical communication equalizers to natural channel fluctuations, potentially reducing the need for retraining triggered by minor changes. Another direction is Zero-Shot Learning (ZSL). Lennard et al. [*J. Lightwave Technol.* 1-9 (2025)] proposed a method involving large-scale domain randomization training in simulation, aiming to equip the neural network equalizer with broad channel feature knowledge.

After sufficient offline training, this model can be directly deployed in real experimental links without any online fine-tuning or training. Their experimental results for 30 Gbaud DP-16QAM transmission showed that this ZSL equalizer could match or even outperform conventional DSP methods that require specific tuning, achieving performance above the 7% HD-FEC threshold across various fiber types (SSMF, PSCF, NZDSF, GI-MMF) and transmission frequencies, while utilizing less than 0.08% pilot overhead to infer channel state, potentially eliminating the need for online training altogether.

Alternatively, considering that online adaptation might still be necessary in complex or rapidly changing scenarios, researchers are also focused on significantly reducing the cost of this adaptation process. Meta-Learning emerges as a powerful technique in this regard. The core idea of meta-learning is "learning to learn", aiming to find an excellent model initialization or learning strategy that enables rapid adaptation to new tasks or environments using minimal data and training iterations. For example, Raviv et al. [*IEEE Trans. Wireless Commun.* 22, 6415 – 6431 (2023)] introduced an online predictive meta-learning framework enabling rapid online adaptation for receivers in dynamic channels, achieving performance gains of up to 2.5 dB compared to traditional online training methods, potentially using self-supervised data without requiring new pilots for the adaptation step. Zhang et al. [*J. Lightwave Technol.* 41, 1269-1277 (2023)] utilized meta-learning to optimize the source domain selection in transfer learning; their experiments demonstrated that the unified initialization found through meta-learning significantly reduced the required fine-tuning convergence epochs by up to 96.24% compared to retraining from scratch when adapting to different target tasks (e.g., varying launch powers or transmission distances over 800km), while also providing an additional Q-factor gain of 0.75 dB. Furthermore, research in Few-Shot Learning (FSL) supports the feasibility of low-cost adaptation. Zhang et al. [*Opt. Express* 31, 23183 – 23197 (2023)] showed that multi-task meta-learning could achieve a low OSNR estimation Mean Squared Error (MSE) of approximately 0.18 dB with only 4-shot fine-tuning in optical performance monitoring. Wang et al. [*Opt. Express* 31, 22622 – 22634 (2023)] proposed an FSL-based AffinityNet equalizer for an OAM-MDM system, which outperformed the traditional Volterra equalizer by 1.7-3.3 dB and a CNN equalizer by 0.5-1.4 dB, while reducing computational complexity by 37% and 83%, respectively. These studies, supported by specific performance metrics and complexity data, demonstrate that meta-learning or few-shot learning can substantially decrease the data volume and training time required for model adaptation. As a baseline, conventional transfer learning itself offers significant efficiency gains; for example, Freire et al. [*Signal Processing in Photonic Communications SpM5C.6* (2021)] showed that transfer learning could reduce training epochs by up to 92% or dataset size by 90%.

In summary, the reviewer's concerns regarding retraining are highly relevant to the practical implementation of neural network equalizers. While our current work employs certain techniques to provide a degree of mitigation, advanced machine learning methods offer promising future directions. These include approaches aimed at enhancing intrinsic model robustness, such as concepts from adversarial training and zero-shot learning, as well as strategies focused on drastically reducing adaptation costs, like meta-learning and few-shot learning. Integrating these cutting-edge methodologies, which have already shown quantifiable advantages in relevant studies, into the design of optical communication equalizers is expected to yield intelligent optical systems that possess both high performance and superior adaptability in the future.

We hope the discussion above can clear the reviewer's concerns. We have added a new part in Supplementary Section 7 to describe the robustness of neural network equalizers, and replaced the

previous section “Retraining frequency of the neural network”.

(Supplementary Section 7) In real transmission scenarios, if the transmitter output intensity varies due to changes in link conditions (e.g., power fluctuations, modulation depth variations, or environmental factors), a neural network equalizer may require retraining or adaptation. Variations in output intensity can affect the signal-to-noise ratio (SNR), modulation depth, and potentially the signal shape. This can change the distribution of the transmitted symbols, on which the neural network was initially trained. If the model encounters a signal that deviates from the training distribution (e.g., due to lower intensity or clipping), its performance could degrade, as it may not effectively map the received signal to the correct symbols.

Improving the robustness of the AI equalizers involves several key strategies: 1) Data augmentation. By introducing variations in the training dataset to enhance generalization, such as incorporating data from different power levels, SNRs, and nonlinear distortions; 2) Apply dropout or $L2$ regularization to prevent the neural network overestimating and encourage robustness to variations in input signals; 3) Combine neural networks with traditional signal equalization methods to enhance interpretability and robustness. Additionally, training the network with adversarial examples or utilizing transfer learning by pretraining the model on a broader dataset can further enhance its resilience to unexpected distortions. However, this process may require a significant amount of time to obtain sufficient and useful training data.

We train the DNN network with mixed data that captured at different power -15.87 dBm, -16.75 dBm, -17.78 dBm, -18.70 dBm and -19.77 dBm with the same proportion (2:2:2:2:2), and then test the signal performance of each power, the results for 130 Gbps PAM-4 under DNN are shown in Fig.S14. The results indicate that training the network using mixed data from different powers and then testing at other power yields better performance than training at a specific fixed power and then testing at other power. However, compared to retraining the network separately at each power level and then testing at each corresponding power, the performance remains inferior.

We also measure the bi-GRU performance for 280 Gbps PAM-4 training at a specific fixed power and then testing at other power, the results are given in Fig.S15a. The results show that there is obvious performance degradation when employing the network trained at a specific fixed power to test the data from other power compared to retraining the equalizer at each power separately. However, it is still better than the DNN case with fixed trained power, which shows the better robustness of bi-GRU to power variations. The results of training the bi-GRU using mixed data captured at different power levels and then testing on other power levels are also shown in Fig.S15b. By training the model with mixed data from different powers, the overall BER performance can be improved to some extent compared to the training based on a specific fixed power.

On the other hand, dropout randomly deactivates a fraction of neurons during training, forcing the network to rely on different subsets of features in each training iteration, during inference (when dropout is turned off), the full network acts as an ensemble of many smaller networks, leading to improved performance and robustness. For tasks like optical signal equalization, where the input data can be affected by noise, dispersion, and nonlinear distortions, dropout helps the network learn features that are less sensitive to these variations. This prevents the model from becoming too dependent on specific neurons and memorizing the training data, improving generalization to new data. By adding dropout layer, we further give a test to the performance of employing dropout to bi-GRU for 280 Gbps PAM-4, the results are shown in Fig.S16a. The results indicate that incorporating dropout in

the network leads to better performance compared to the case without dropout.

In the meanwhile, L_2 regularization (also known as weight decay) is a technique used to prevent overestimating by penalizing large weights in a neural network. It modifies the loss function by adding a penalty term proportional to the sum of the squared weights, encouraging the model to keep weights small and thus improving generalization. We also tried employing L_2 regularization to the bi-GRU network for 280 Gbps PAM-4, the results are shown in Fig.S16b, and the improved performance proves the effectiveness of the method.

Furthermore, by applying both dropout and L_2 regularization to the bi-GRU network simultaneously, the performance demonstrates superior results compared to using either method individually, indicating further enhanced network robustness to power variations, as shown in Fig.S16c. However, the performance remains inferior compared to retraining the model at each specific power level separately.

By employing a relatively lite AI equalization method within the proposed system, a balance between performance and power consumption can be achieved. Therefore, the employed network scale of the AI equalizer in this work is small, only consisting of four layers in total for the DNN and three layers for the GRU, respectively. Normally, for neural networks, smaller network scale tends to exhibit lower robustness, which has been shown by numerous studies. Under the network configuration shown in this work, the robustness is still challenging. By employing data mixing, dropout and L_2 regularization, the robustness can be enhanced to some extent.

To further address the raised questions, we made the point-by-point response as follows.

Comment 1: *"This paper really does have two separate parts presented together. A large part of the submission is dedicated to the characterization and performance of the modulator itself. However, the performance of the modulator array is fundamentally separate from the learned equalizer. From the perspective of the modulator alone, a push-pull type, MZI-based slow light modulator is quite common. Even other MZI-based or ring-based modulators can be operate faster (and also more energy efficient) than shown in this paper, with several pure-Si examples reaching above 100 Gbps (some of which are already referenced by the authors):*

- Li, Miaofeng, et al. "Silicon intensity Mach – Zehnder modulator for single lane 100 Gb/s applications." *Photonics Research* 6.2 (2018): 109-116.
- Sun, Jie, et al. "A 128 Gb/s PAM4 silicon microring modulator with integrated thermo-optic resonance tuning." *Journal of Lightwave Technology* 37.1 (2018): 110-115.
- Li, Hao, et al. "A 112 Gb/s PAM4 silicon photonics transmitter with microring modulator and CMOS driver." *Journal of Lightwave Technology* 38.1 (2020): 131-138.

Given these existing results (and the identical structure in ref [39] of the paper), the presented modulator itself does not possess a great novelty in terms of its design/geometry or its performance metrics."

Our reply: Thanks for the comments. We apologize for not explaining clearly the compatibility between Si-SLMs and AI equalizers. There is a strong match and correlation between the two aspects, and this is also the starting point of our research. The AI equalizer is particularly suitable for silicon slow-light modulators for the following reasons:

Nonlinear distortions: Improving the indicators of modulators, especially bandwidth and modulation efficiency, is the device-level guarantee for ultrahigh-speed transmission. By utilizing the slow-light approach, the bandwidth and modulation efficiency can be improved. However, the physical principle is still the nonlinear plasma dispersion effect of pure silicon, and the insufficient linearity is not enhanced, which becomes the bottleneck for high-order transmission. Therefore, the lack of linearity brought by silicon material can be compensated by AI equalization exactly.

Fabrication deviations: Pure silicon modulation is based on the plasma dispersion effect, and it is difficult to achieve complete consistency in the doping process. Also, the introduction of slow-light effect is based on the grating waveguide structure. Compared to the two-dimensional photonic crystal, although the waveguide grating structure has improved the process stability with CMOS compatibility, the unavoidable structure fluctuations of gratings still exist. After introducing AI enhancement, the differences between grating devices can be minimized. Therefore, through the AI equalization at the back end, the process limitations of the slow-light structure can be eliminated on the same wafer.

System design: Based on the points above, the AI equalizer is especially suitable for doped-silicon slow-light modulators for ultrahigh-speed applications. Therefore, the differences between doped-silicon slow-light modulators can be eliminated by AI equalization, making the deployment of large-scale slow-light modulator array possible. The modulator array and the WDM system can maximize the performance advantages by introducing the AI approach.

For specific modulator design, although Si-SLMs have been proposed before indeed, our previous work [Sci. Adv. 9(42), eadi5339, 2023] demonstrated the effectiveness of slow light schemes in achieving ultrahigh bandwidth on silicon platforms, and this work further explored the multi-level rate potential based on the slow-light scheme. However, regarding operation speed, our designed Si-SLMs are still far ahead of other types of modulators. The fundamental indicator for measuring the speed of the device response is the electro-optical (EO) bandwidth, which refers to the 3dB loss of the EO transmission S_{21} response. In this work, the EO bandwidth of Si-SLMs is around 90 GHz, and the EO bandwidths in the above articles listed are 60 GHz, 50 GHz and 50 GHz, respectively. For the data transmission rates, the highest rate of this work is up to 400 Gbps, while these works are around 128 Gbps, 128 Gbps and 112 Gbps in PAM-4 format, respectively. In order to more specifically compare the proposed approach with the indicators of previous studies (including Si-MZMs, Si-MRMs and other Si-SLMs), we list Table 1 for detailed comparison and have some discussions in the main part, which is also shown below.

Table 1. Comparison with representative silicon modulators

Ref.	Type	Operating wavelength (nm)	EO bandwidth (GHz)	Modulation efficiency (V·cm)	Optical passband (nm)	Data rate (Gbps)
32	Si-MZM	C-band	60	1.40	NA	128 PAM-4
14	Si-MZM	C-band	60	NA	NA	200 PAM-4
34	Si-MZM	O-band	47	1.35	NA	225 PAM-8
18	Si-MZM	C-band	67	3.00	NA	360 ASK-8
13	Si-MZM	C-band	65	1.90	NA	224 PAM-4

36	Si-MRM	O-band	50	0.52	~0.3	128 PAM-4
17	Si-MRM	C-band	67	NA	~0.5	200 PAM-4
16	Si-MRM	O-band	60	0.80	0.28	240 PAM-4
19	Si-MRM	C-band	50	0.63	~0.3	330 PAM-8
15	Si-MRM	O-band	48	0.60	0.35	200 PAM-4
43	Si-SLM	C-band	38	0.44	15.0	64 PAM-4
41	Si-SLM	C-band	40	0.51	2.0	90 PAM-4
This work	Si-SLM	C-band	90	0.82	7.0	400 PAM-4

From the comparison, we can see that the bandwidth, modulation efficiency and passband of the Si-SLM designed are remarkable for pure silicon modulators simultaneously. The EO bandwidth of our device exceeds that of previous modulators, which means faster operation speed. Also, the modulation efficiency is improved compared with Si-MZMs, and the optical passband is larger than Si-MRMs. Meanwhile, based on the favorable indicators, the transmission performance of our device is ahead of other Si-SLMs. Also, as a pure silicon scheme, no additional heterogeneous materials are introduced in the fabrication flow, and the advantage of wafer-level production utilizing standard silicon photonic process ensures the feasibility of extensive deployment of this solution.

We hope the discussion above can clear the reviewer's concerns.

Comment 2: *“On the other hand, it can still be argued that coupling the modulator array with a learned receiver backend presents valuable advancements for a broad electronics/optics community. However, even though it is not explicitly mentioned by the authors, the learned equalizer’s main goal here is to mitigate inter-symbol interference caused by various distortion mechanisms in the modulator (as well as through the other components in the transceiver pipeline). Other well-known equalizers (such as DTE or Volterra), or some form of equalization is already used quite frequently in modern high-speed communication networks. Given this fact, the authors should provide detailed analysis on how their learned equalizer compares with other equalization techniques in mitigating ISI for the slow-light modulator presented. Without this comparison, the submission does not clearly demonstrate novelty from the equalizer perspective either, at least in its current state.”*

Our reply: Thanks for the comment. The feed-forward equalizer (FFE) and decision feedback equalizer (DFE) are effective methods for linear impairments compensation and widely used nowadays. The most basic components of FFE is the finite impulse response (FIR) filter, the output of FIR is expressed as:

$$y(k) = \sum_{l=0}^{n-1} h_l x(k-l)$$

where $x(k)$ and $y(k)$ are the input and output signal of FIR at the sampling instant k , respectively. $h = [h_0, h_1, h_2, \dots, h_{n-1}]$ is the array of tap weights, while n is the number of taps.

Unlike FFE, the input of DFE is the signal after decision, the output signal of DFE can be calculated by

$$z(k) = r(k) - \sum_{l=0}^{n-1} h_l \delta(k-l)$$

More details can be found in reference [J. Light. Technol.,36(2), 377-400 (2018)].

Thanks to FFE and DFE, the linear impairments can be efficiently eliminated, however, the residual nonlinear distortions mainly induced by the device and system can severely impact the transmission performance, which cannot be effectively compensated by FFE/DFE. One of the most common algorithms to compensate nonlinear effects is Volterra nonlinear equalizer (VNLE), the output of third-order VNLE can be expressed as [Mathews, V.J. IEEE Signal Process. Mag., 3, 10–26 (1991)]:

$$z(k) = \sum_{l=0}^{n_1-1} h_l x(k-l) + \sum_{l=0}^{n_2-1} \sum_{i=0}^l h_{l,i} x(k-l)x(k-i) + \sum_{l=0}^{n_3-1} \sum_{i=0}^l \sum_{j=0}^i h_{l,i,j} x(k-l)x(k-i)x(k-j)$$

where h_l , $h_{l,i}$, $h_{l,i,j}$ are the tap weights of 1st-order, 2nd-order, 3rd-order kernels, respectively. n_1 , n_2 and n_3 respectively represent the number of taps for the linear part, 2nd-order nonlinear part and 3rd-order nonlinear part. Linear impairments in the system can be eliminated by the first-order kernel of VNLE, while the second- and third-order kernels compensate for nonlinear distortions. VNLE expands the equalization function using polynomial terms to model nonlinear distortions, the computational complexity grows exponentially in VNLE with the order of nonlinearity increases. The neural network is computational model loosely inspired by its biological counterparts and has been proposed to mitigate the nonlinear impairment in optical system. By training on large datasets, neural network can better capture the underlying distribution of the optical signal and learn the subtle relationships between different symbols, this enables them to effectively handle cases where traditional equalizers might fail. Neural networks can learn to directly map the distorted signal back to the transmitted symbols without manually decomposing the problem into individual components, which shows a completely different work philosophy compared to DFE and VNLE that based on filters.

The performances of DFE and VNLE compared to DNN, bi-GRU and T-biGRU for different high-baud PAM-4 and PAM-8 signals are evaluated, the results are shown in Figure S13. The tap weights of DFE are updated by least mean squares (LMS) algorithm and third order VNLE is employed. The results indicate that by employing bi-GRU and T-biGRU compared to DFE and VNLE, there's obvious performance promotion at 300 Gbps, 360 Gbps, 400 Gbps PAM-4 and 330 Gbps, 360 Gbps, 390 Gbps PAM-8 signals, respectively. AI equalizers outperform both DFE and VNLE.

Figure S13. System performance of DFE and VNLE compared to DNN, bi-GRU and T-biGRU. (a) PAM-4

signals. (b) PAM-8 signals.

(Supplementary Section 6) In most transmission scenarios, signal equalization is essential for high-order modulation formats such as PAM-4 and PAM-8. The linear equalizers such as feed-forward equalization (FFE) and decision-feedback equalization (DFE) are effective methods for linear impairments compensation and widely used nowadays. The most basic component of FFE is the finite impulse response (FIR) filter, the output of FIR is expressed as:

$$y(k) = \sum_{l=0}^{n-1} h_l x(k-l)$$

where $x(k)$ and $y(k)$ are the input and output signal of FIR at the sampling instant k , respectively. $h = [h_0, h_1, h_2, \dots, h_{n-1}]$ is the array of tap weights, while n is the number of taps.

Unlike FFE, the input of DFE is the signal after decision, the output signal of DFE can be calculated as:

$$z(k) = r(k) - \sum_{l=0}^{n-1} h_l \delta(k-l)$$

By adopting FFE/DFE, the linear impairments can be eliminated efficiently. However, the residual nonlinear distortions mainly induced by the device and system can severely impact the transmission performance, which cannot be effectively compensated by FFE/DFE. One of the most common algorithms to compensate nonlinear effects is Volterra nonlinear equalizer (VNLE), the output of third-order VNLE can be expressed as:

$$z(k) = \sum_{l=0}^{n_1-1} h_l x(k-l) + \sum_{l=0}^{n_2-1} \sum_{i=0}^l h_{l,i} x(k-l)x(k-i) + \sum_{l=0}^{n_3-1} \sum_{i=0}^l \sum_{j=0}^i h_{l,i,j} x(k-l)x(k-i)x(k-j)$$

where h_l , $h_{l,i}$, $h_{l,i,j}$ are the tap weights of 1st-order, 2nd-order, 3rd-order kernels, respectively. n_1 , n_2 and n_3 respectively represent the number of taps for the linear part, 2nd-order nonlinear part and 3rd-order nonlinear part. Linear impairments in the system can be eliminated by the first-order kernel of VNLE, while the second- and third-order kernels compensate for nonlinear distortions. VNLE expands the equalization function using polynomial terms to model nonlinear distortions, the computational complexity grows exponentially in VNLE with the order of nonlinearity increases. The neural network is computational model loosely inspired by its biological counterparts and has been proposed to mitigate the nonlinear impairment in optical system. By training on large datasets, neural network can better capture the underlying distribution of the optical signal and learn the subtle relationships between different symbols, this enables them to effectively handle cases where traditional equalizers might fail. Neural networks can learn to directly map the distorted signal back to the transmitted symbols without manually decomposing the problem into individual components, which shows a completely different work philosophy compared to DFE and VNLE that based on filters.

To further measure the difference between AI equalizers and traditional algorithms, the performance of DFE and VNLE compared to DNN, bi-GRU and T-biGRU for high-speed PAM-4 and PAM-8 signals of different rates is evaluated, the results are shown in Fig.S13. The tap weights of DFE are updated by least mean squares (LMS) algorithm and third order VNLE is employed. The results indicate that by employing bi-GRU and T-biGRU, compared to DFE and VNLE, there is obvious performance promotion at 300 Gbps, 360 Gbps, 400 Gbps PAM-4 (Fig.S13a) and 330 Gbps, 360 Gbps, 390 Gbps PAM-8 (Fig.S13b)

signals, respectively. From the experimental results, it can be seen that AI equalizers outperform both DFE and VNLE.

Comment 3: “Even more generally, the inner workings of the equalizer should be discussed in more detail. For instance, does the equalizer perform near-identical functions for wavelengths with similar insertion losses through the modulator. Do these adjacent wavelength channels experience similar levels or types of nonlinearities throughout the transceiver? If the network was only trained using a single channel, how well would it generalize to using multiple different channels through transfer learning? These questions would clarify the specific details and capabilities of the equalizer network, as its treatment as a black-box tool does not provide sufficient insight into its capabilities, limitations, or any channel-specific mechanisms.”

Our reply: The modulator possesses a flat passband of 7 nm around 1550 nm, and the passbands are basically consistent between different devices, as shown in Figure 2(g). For WDM Si-SLM system, the specific wavelengths of 8-channels are from 1548 nm to 1555 nm, with the spacing of 1 nm, the insertion loss and experienced nonlinearities are consistent between different channels due to the flat responses for the Si-SLMs. For 400 Gbps PAM-4 signals, the performance for Ch1-Ch8 is relatively close, shown in Figure 4 (d). For the results in the main part, the equalizer training and testing are performed for each channel (Ch1-Ch8) using different captured data groups (as discussed in General Comment) from the same channel. Here, regarding the transfer learning between different channels, we train the equalizer network using data from a single channel (Ch5) and evaluate its performance on other channels (Ch2, Ch4, Ch6, Ch8). The results are shown in Figure R2 below. The results indicate that the performance is similar compared to the results shown in Figure 4 of the manuscript. This consistency is attributed to the approximate channel loss and nonlinearities, as well as the modulator's minimal sensitivity to wavelength variations within the passband.

Figure R2. Training bi-GRU from one channel and evaluating on other channels.

Comment 4: “I fully agree with reviewer #1’s earlier comment that the title with “AI-powered silicon photonic slow light technology” or “AI-driven silicon photonic slow-light technology” (as it is currently written) does not convey the main novelty of the presented work. The slow-light modulator itself does not include any AI-driven or AI-designed elements. As reviewer #2 also noted, this updated title still creates a connection between the modulator and detection through a learned equalizer. This presentation leaves the reader with the impression that the modulator itself is comprised of AI-driven components. In my opinion, even the update to the title still does not resolve this evident potential misunderstanding.”

Our reply: Thanks for the comment. We have further referred to the naming characteristics of related AI and interdisciplinary research fields, and changed the title to “Exploring 400 Gbps/λ and beyond with **AI-accelerated** silicon photonic slow-light technology”. In this work, the function of AI is precisely to accelerate the transmission rate of silicon photonics, so AI-accelerated should be a suitable adjective. We hope by using the term “**AI-accelerated** silicon photonic slow-light technology”, the vision of achieving a significant transmission speed increase by accelerating silicon slow-light chips through AI can be properly expressed. We have also already changed the term “AI-driven” to “AI-accelerated” in the main part and Supplementary Information.

Comment 5: *“The abstract states that the system operates “without additional wavelength adjustment and temperature feedback systems.” However, a DC pad array is still used to control the modulator operating point via a TiN heater, which is a standard thermal tuning mechanism in many silicon modulators. Are the authors referring to a different experimental configuration when making this claim? This requires clarification, as it could mislead readers regarding the tuning requirements of the system as well as its thermal stability.”*

Our reply: Thanks for the comment and suggestion. In fact, the additional temperature feedback system here refers to the thermo-electric cooler (TEC), which is used as the testing platform to provide temperature feedback and maintain a stable testing temperature. The DC pad array (the right side of Figure 2(a)) is adopted here as the standard thermal tuning mechanism to control the operating point of the modulator to be centered at the quadrature point (these are actually two different things indeed), and we also point out the role and function of TiN heater in the main part and Supplementary Section 1. For the thermal stability, thanks to the large passband (7 nm) of the Si-SLMs, the TEC is not required as the operating platform for the chip. In contrast, for Si-MRMs, due to the narrow passband, which brought low thermal stability, Si-MRMs must be placed on a TEC platform when working and extra power budget is required. Considering that this is a description of the test environment rather than an introduction of the performance indicators, and to avoid potential misunderstanding, we have removed this sentence from the abstract. Instead, we directly point out that the chip has a thermal-insensitive structure and high temperature stability in the abstract. Meanwhile, to make clarification, as TEC is the common term in this field, we have changed all the term “temperature feedback systems” to standard “thermo-electric cooler (TEC) platforms” in the main part.

(Main part, line 25) By utilizing the artificial neural network, we achieve a data capacity of 3.2 Tbps based on an 8-channel wavelength-division-multiplexed silicon slow-light modulator chip with a thermal-insensitive structure, leading to an on-chip data-rate density of 1.6 Tb/s/mm².

(Main part, line 103) Meanwhile, the transmission link does not require individual resonant wavelength adjustment and additional thermo-electric cooler (TEC) platforms, thus reducing the system budget.

(Main part, line 232) In practice, the optical passband of the modulator is the reflection of the optical bandwidth, and a wide passband is essential for multi-wavelength applications with high thermal robustness, reducing the requirements for additional TEC operating platforms.

(Main part, line 518) Meanwhile, benefiting from the large flat passband around 1550 nm, the whole links are without individual resonant wavelength adjustment and TEC operating platforms, thus reducing the system budget.

(Main part, line 543) Under precise regulation, our compact thermal-insensitive Si-SLM chip possesses an ultrahigh bandwidth, a remarkable modulation efficiency, a large optical passband simultaneously, leading to an on-chip data-rate density of 1.6 Tb/s/mm², without individual resonant wavelength adjustment requirements and additional TEC operating platforms.

Comment 6: “Transmission is reported as a “per wavelength” basis. But the system’s capability to work without additional thermal tuning is also emphasized. Does this mean it would not be possible to use a tighter spectral spacing than 1nm used (from Fig 2g), corresponding to more than 8 wavelengths spectrally? Are there any potential routes that could enable 50GHz DWDM spacing in such a system? Does this pose any limitations on the spectral density of information that can be encoded using the presented system, or are there any other practical bottlenecks due to spectral broadening?”

Our reply: Thanks for the comment. In fact, the main bottleneck limiting narrower wavelength spacing WDM is indeed the spectral broadening caused by high-speed IM/DD signals. To increase the spectral density further, the frequency spacing between channels needs to be reduced. However, the spectral broadening cannot be ignored for multi-wavelength IM/DD transmission, which will cause spectral overlap, resulting in crosstalk and affecting signal quality, especially when the data rate increases to 400 Gbps per lane. In high-speed IM/DD signal transmission, the signal broadening on spectrum for one side is as follows:

$$\Delta\nu = R_s \times \frac{1}{2} \times (1 + \alpha)$$

in which R_s is the baud rate of the signal, α is the roll-off factor (0.1 in the experiment), $\Delta\nu$ is the signal broadening on one side, and the total width of the signal is $2\Delta\nu$.

For WDM experiments of DNN, the signal rate is up to 224 Gbps PAM-4 (112 Gbaud), the total frequency width for one channel is 123 GHz. The wavelength interval between two adjacent channels is 1 nm (125 GHz), which is basically enough. However, for 400 Gbps (200 Gbaud) PAM-4, the frequency width for one channel is 220 GHz. In the practical WDM experiment of GRU, to reduce the crosstalk caused by signal broadening, we select channel 1,3,5,7 (odd number) as one WDM transmission path, while channel 2,4,6,8 (even number) as the other WDM transmission path. After WDM of each 4 channels, the two WDM paths are then transmitted in parallel to reach 3.2 Tbps capacity, similar to the extensively deployed transceiver scheme (2×FR4) [OFC, M4H.1, (2022)]. Therefore, for the two on-chip channels separated by one (e.g. Ch2 and Ch4), the wavelength interval is 2 nm, and the corresponding frequency range is 250 GHz, which is larger than the signal frequency width 220 GHz. To explain the practical WDM experiment more clearly, we have a detailed description of the GRU experimental system in the corresponding main part.

We hope the above analysis could make it clearer.

Comment 7: “The equalizer’s workload should be reported as an overhead with all relevant time and energy metrics required for training and re-training the system, in case of configuration scenarios like those shown in Figure S10 by the authors. Particularly, it would be useful to see training curves for the learned model, how many iterations or batches were necessary for training, and the energy and time requirements for this process if it were to be performed as an initial calibration step before transmission and detection of real data.”

Our reply: Thanks for the comment. In the experiment, the retraining process follows the same procedure as the initial training. A computer equipped with an Intel Core i9-13900HX processor, 32 GB of 5600 MHz memory, and an Nvidia RTX 4060 GPU is employed. The power and time consumption of the different equalizers are listed in Table R3. In fact, the current work remains at the proof-of-concept stage, and the time consumption will directly depend on the configuration of the GPU hardware. It is important to clarify that these figures represent the overhead using a standard, full retraining procedure on general-purpose hardware, without specific optimizations targeted at accelerating the retraining phase itself. In a practical deployment scenario, where rapid adaptation might be necessary, various established techniques could be employed to significantly reduce this retraining time. For instance, transfer learning offers a path to faster adaptation by fine-tuning only specific parts of an already trained model, requiring less data and time [*Journal of Big data*, 3, 1-40 (2016)]. Another approach is knowledge distillation, where the capabilities of a large, complex model are transferred to a smaller, faster student model which is then easier to retrain [*arXiv:1503.02531v1* (2015)]. Furthermore, direct model optimization techniques can reduce computational load; quantization achieves this by lowering the numerical precision of the model's parameters [*arXiv:2103.13630v3* (2021)], while pruning strategically removes redundant connections or weights from the network [*Proc. Machine learning and systems 2*, 129-146, (2020)]. Finally, executing these optimized models on dedicated AI accelerator hardware [*Proc. IEEE 105*, 2295 – 2329 (2017)], potentially leveraging emerging photonic co-processors [*Science 384*, 202 – 209 (2024)], would provide substantial gains in speed and energy efficiency.

Table R3. Summary of Power and time Consumption in the system.

Equalization Algorithm	Power (W)	Time(s)	Data Rate (Gbps)
DNN	~34.690	62.1367	200
bi-GRU	~66.570	1077	400
T-biGRU	~67.560	1318	400

For 400 Gbps PAM-4 processing, the training curve of bi-GRU equalizer is also presented in Figure R3. Here, the batch size is set to 300, and the number of iterations is 650 with a learning rate of 0.005.

Figure R3. The training curve of bi-GRU for 400 Gbps PAM-4 equalization.

Comment 8: “Similarly, due to such strong emphasis on the capability of the learned equalizer, it would be good if the authors could provide a system performance snapshot both with and without the equalizer in place. This is also related to reviewer #3’s question on the improvement of SFDR (as well as other relevant performance metrics), even if an improved SFDR is not directly targeted by the system implemented. For instance, while clear eye diagrams are observed when the trained equalizer is used

above 200 Gbps, how is the performance affected when no learned equalizer is used at all? At what speed the version of the system with no learned equalizer still achieves BERs compatible with HD-FEC? Can this potential reduction in speed (or eye clarity / BER metrics) justify the decrease in energy usage? This is particularly important as the reported power consumption metrics for the neural network components in Table S1 are larger in comparison to other parts of the presented transceiver.”

Our reply: Thanks for the comment. We sincerely appreciate the reviewer's insightful comments regarding the power consumption associated with neural network equalizers. As shown in the response to Comment 2, the system performance without employing the learned equalizer is further evaluated, including DFE and VNLE. The results show that compared to the learned equalizer DNN, bi-GRU, and T-biGRU, the performance of DFE and VNLE is inferior, particularly for DFE, due to its limited capability in compensating for nonlinear distortions. In contrast, VNLE demonstrates better performance compared to DFE due to its enhanced ability to compensate for nonlinearity. For power consumption, we have previously discussed it at the system level in detail in Supplementary Information, and the power consumption of the system with different AI equalizers is shown in Table S1 in the manuscript. Furthermore, we evaluated power consumption in combination with the algorithm, especially for the comparison between traditional nonlinear equalizers and AI equalizers. By employing VNLE, we achieve a compatible BER with HD-FEC threshold at a data rate of ~130 Gbps PAM-4, as shown in following Figure R4. The power consumption is also measured during the VNLE algorithm running. Consistent with our previous response, the VNLE was performed on a computer equipped with an Intel Core i9-13900HX processor, 32 GB of 5600 MHz memory, and an Nvidia RTX 4060 GPU. Power consumption data were obtained using HWINFO [<https://www.hwinfo.com/>] and based on our previous research [*Nat. Commun.*, 14(66), (2023)]. The measured power consumption of VNLE for ~130 Gbps is 42.39 W. Here we define the equalizer power consumption per bit (*EPCpB*) as:

$$EPCpB = \text{Power Consumption (W)} / \text{Data Rate (Gbps)}$$

A smaller *EPCpB* value indicates a lower power consumption per bit. Based on the power consumption of each equalizer, the calculated *EPCpB* for VNLE, DNN, bi-GRU, T-biGRU are 0.3261, 0.2041, 0.1664, 0.1689, shown in Table S2 below, where the speed selections are all below HD-FEC threshold. From the calculation results, it can be seen that the reduction in speed of employing VNLE does not justify the decrease in energy usage. Simultaneously, it can be found that under the circumstance that the algorithm is already adopted, the AI equalizers (especially bi-GRU and T-biGRU) have not significantly increased *EPCpB* (or even become smaller). This is because the increase in power consumption also brings about an enhancement in the transmission rate, so that the algorithm energy consumption required to transmit each bit of data is controlled within an acceptable range. In other words, it is worthwhile to use higher total power consumption in exchange for an improvement in transmission speed based on AI equalizations.

Figure R4. The performance of VNLE for high-speed PAM-4 signal equalization.

Table S2. Summary of power consumption for different equalizers

Equalizer	Power (W)	Signal format	Speed (Gbps)	BER threshold	EPCpB (nJ/bit)
VNLE	42.39	PAM-4	130	HD-FEC	0.3261
DNN	34.69	PAM-4	170	HD-FEC	0.2041
bi-GRU	66.57	PAM-4	400	HD-FEC	0.1664
T-biGRU	67.56	PAM-4	400	HD-FEC	0.1689

We concur that total power dissipation in electronic hardware presents critical bottlenecks for high-speed systems like the 400 Gbps signals considered here. Meanwhile, photonic computing offers a promising path forward, leveraging the intrinsic speed, low latency, and parallelism of photonics for equalization tasks directly in the optical domain [Nat. Commun., 16(1), 292, (2025)] [Nat. Commun., 15(1), 551, (2024)] [Nature, 632(8024), 280-286, (2024)] [Nat. Photon., 18(12), 1335-1343, (2024)] [Sci. Adv., 10(45), eadp0391 (2024)]. Critically, multiple studies indicate significant potential for lower power consumption compared to electronic DSP, with some providing quantitative estimates [IEEE J. Sel. Top. Quantum Electron. 29, 7400212 (2023)] [Photonics Res. 11, 878–886 (2023)] [Sci. Rep. 12, 4216 (2022)].

Recent advancements demonstrate the effectiveness of various photonic computing approaches for equalization: Integrated PNNs and time-delayed perceptron have successfully performed linear Chromatic Dispersion compensation for formats like OFDM [J. Light. Technol., 43(7), 3034-3040, (2025)], 10 Gbps IMDD over 125 km [Photonics Res. 11, 878–886 (2023)]. Implementations show high-speed operation (e.g., 16 Gbps with BER < 10⁻⁶ [Sci. Rep. 12, 4216 (2022)]) with significantly lower power needs (tens of mW) compared to typical DSP [Photon. Res. 11, 878–886 (2023)].

Photonic Reservoir Computing has effectively addressed nonlinear distortions, outperforming electronic FFE [J. Lightw. Technol. 37, 2232–2239 (2019)], extending transmission reach [J. Lightw. Technol. 37, 2232–2239 (2019)], operating below FEC thresholds for 32 Gbps signals [Opt. Express 29, 30991–30997 (2021)], and achieving substantial BER reduction (3 orders of magnitude for 25 Gbps OOK) via PSO training [J. Lightw. Technol. 41, 5841–5850 (2023)]. Hybrid photonic-electronic NNs have also tackled nonlinearity in ultra-long-haul systems [Nat. Electron. 4, 837–844 (2021)]. For complex WDM scenarios, photonic RNNs have been proposed to simultaneously mitigate intra- and inter-channel distortions. Power analysis for a 1.6 Tbit/s system projects a photonic power consumption of ~12.7 W versus >40 W for electronics, alongside a latency reduction from microseconds to picoseconds [IEEE J.

Sel. Top. Quantum Electron. 29, 7400212 (2023)]. Furthermore, reconfigurable photonic processors have demonstrated functions like MIMO descrambling [ACS Photonics 7, 792–799 (2020)]. Collectively, these studies establish photonic computing as a rapidly advancing field capable of diverse equalization tasks. They demonstrate a viable route to overcoming electronic processing limitations in latency and, significantly, power consumption, as evidenced by quantitative analyses.

Therefore, while our current work utilizes electronic neural networks, facilitating rapid algorithmic development and validation, we fully acknowledge the reviewer's pertinent concerns regarding power consumption for practical deployment. We firmly believe that the effectiveness of the equalization algorithms demonstrated in our manuscript, considered alongside the substantial potential for low-power, high-speed execution offered by evolving photonic computing hardware, validates the long-term relevance of our research direction. Future realization of these or similar algorithms on optimized photonic platforms holds the promise of delivering the energy-efficient, high-performance equalization solutions crucial for next-generation optical networks.

We hope the above analysis could make it clearer. We have also added the new part of power consumption in Supplementary Section 8.

(Supplementary Section 8) Furthermore, we evaluated and compared the power consumption of traditional nonlinear algorithm and AI equalizers. By employing VNLE, we achieve a compatible BER below HD-FEC threshold at 130 Gbps PAM-4. Using the same method as described above for AI equalizers, The power consumption was measured to be 42.39 W during the VNLE algorithm running. Here we define the equalizer power consumption per bit $EPCpB$ as:

$$EPCpB = \text{Power Consumption (W)} / \text{Data Rate (Gbps)}$$

A smaller $EPCpB$ value indicates a lower power consumption per bit. Based on the power consumption of each equalizer, the calculated $EPCpB$ for VNLE, DNN, bi-GRU, T-biGRU are shown in Table S2, where the speed selections are all below HD-FEC threshold. From the calculation results, it can be seen that the speed reduction of employing VNLE does not justify the decrease in energy usage. Simultaneously, under the circumstance that the algorithm is already adopted, the AI equalizers (especially bi-GRU and T-biGRU) have not significantly increased $EPCpB$ (and even reduce it). This is because the increase in power consumption also brings about an enhancement in the transmission rate, so that the energy consumption of algorithm required to transmit each bit of data is controlled within an acceptable range. In other words, it is worthwhile to use higher equalizer power consumption in exchange for an improvement in transmission speed based on AI equalizers.

Comment 9: *“The presented modulator still operates through the physical principle of this nonlinear plasma dispersion, which plays an important role in signal distortion. From the responses by the authors, it seems that the modulator’s 10V total swing is sufficient for about 30% of V_{pi} , which is sufficient in case an equalizer is used. It would be important to quantify what changes to the equalization requirements are necessary if a smaller driving voltage is used, but at slightly lower speeds. Once again, a slight reduction in speed and driving voltage could yield a large decrease in signal distortion, rendering the equalizer unnecessary under certain circumstances. This should be tested for a more complete and comprehensive analysis of the transceiver system.”*

Our reply: Thanks for the comment. In a high-speed optical transmission system, the driving voltage of

the electrical signal plays a critical role in determining the quality and performance of the PAM-4 optical signal. A sufficient driving voltage ensures sufficient signal amplitude for effective modulation while maintaining high SNR and keeps the input voltage within the linear operating range of the MZM, ensuring accurate PAM-4 level transitions. Low driving voltage will result in insufficient modulation depth, causing small eye openings in the PAM-4 signal and reducing the receiver's ability to distinguish between levels. Meanwhile, a high driving voltage can cause voltage overshoot, leading to clipping distortion and the generation of harmonic components, resulting in increased bit errors and a nonuniform eye opening, ultimately degrading system performance.

Here, we made an evaluation on the bi-GRU performance under different driving voltage for 360 Gbps PAM-4 signal. The corresponding BERs are 4.50×10^{-4} , 6.01×10^{-4} , 9.71×10^{-4} and 6.23×10^{-3} for 5 Vpp, 4.6 Vpp, 4.2 Vpp and 3.4 Vpp, respectively. As the driving voltage decreases, the BER degrades and exceeds the HD-FEC threshold at 3.4 Vpp.

In the experiment demonstrated in this work we set the driving voltage at 5 Vpp. The performance of low data rate transmission under different driving voltages is also evaluated using bi-GRU equalizer. For 180 Gbps and 160 Gbps PAM-4 signals, the BERs at a 3.4 Vpp driving voltage remains far below the HD-FEC threshold, achieving values of 7.51×10^{-5} and 1.5×10^{-5} , respectively. The reduction in signal speed can lower the required driving voltage. However, the equalizer remains essential in these cases. Without the equalizer, the BER rapidly exceeds 3.8×10^{-3} , making error correction ineffective.

Meanwhile, for a 100 Gbps OOK signal and 116 Gbps PAM-4 without any equalizer (DSP free), the BER could still remain below the HD-FEC threshold. Details can be found in our previous work [*Han et al., Sci. Adv. 9(42), eadi5339, 2023*]. Compared to OOK, PAM-4 encodes 2 bits per symbol instead of 1, resulting in reduced signal margins, increased noise sensitivity, and more severe inter-symbol interference. As a result, equalization is typically essential in high-speed PAM4 systems as shown in this work to mitigate the very much nonlinearity challenges as we analyzed previously.

We hope the above analysis could make it clearer.

Comment 10: *“The explanation that “The retraining frequency depends on how fast the components degrade or how often they are replaced” does not fully convey the potential limitations of the system. From the authors’ description of the learned equalizer, it is evident that nonlinear distortions are a large part of what the equalizer compensates for. Therefore, any changes these distortions (including those induced by thermal fluctuations or physical displacements in the 100s of meters of fiber itself) would require potential recalibration. These types of changes are quite common in data center application settings. To this end, it would be crucial to quantify how robust the system is to these types of changes, and how much change in these parameters would require retraining of the equalizer. For example, does the model suffer from overfitting to any of these environmental conditions or the equipment used? In contrast, the presented hypothetical changes such as replacement of the driver, EDFA, or changing the DC bias are comparatively rare scenarios. This was also an important part of an earlier comment by reviewer #2, which was not fully addressed in this current revision. Another relevant parameter is this scenario is also the optical power, as indicated by reviewer #3 in their comment regarding output intensity. It would be good to see how much change in this power the system would be able to tolerate without needing recalibration. Currently, the only test performed by the authors is shown for a change of 2dB.”*

Our reply: Thanks for the comment. As we replied to the comment of Reviewer 3, the AI equalizer is used at the back end of the system, so it is a full-link nonlinear distortion optimization for both optical and electrical devices. For the modulator nonlinearity, silicon modulators employ plasma dispersion effects by forming PN junctions which is intrinsically nonlinear. During ultrahigh-speed modulation, carriers cannot fully recover to their steady state before the next cycle, causing the accumulation of nonlinear effects. Also, the Mach-Zehnder modulator inherently exhibits a cosine transfer function, which will also introduce nonlinearities, deviating the output from a linear mapping of the input and thus degrading the modulation performance. For the system, the primary electrical nonlinear sources originate from RF components, attributed to the inherent nonlinear behavior of charge transport at the transistor junction which is predominantly observed in the transistor driver. And nonlinear effects such as amplitude-to-phase conversion, in-band and base-band memory effects in amplifiers, the constrained operation range of the analog-to-digital converter introduced nonlinear vibrations cannot be disregarded. Quantitatively varying the characteristics of system components to achieve a specific value of nonlinear change is quite challenging. However, system performance metrics such as BER, SNR, and eye diagrams can serve as indicators of varying nonlinear distortions, link failures, or physical displacements encountered. At the system level, when performance degrades beyond the required threshold, such as the HD-FEC limit, retraining is typically necessary.

Overfitting in neural networks occurs when the model performs well on the training set but fails to generalize to unseen data, such as the test set. Normally, the test set should be collected under the same conditions as the training set to evaluate the model's generalization. If the test set is obtained under different conditions (e.g., varying power levels, noise levels, or distortions), and the model performs poorly, it may not necessarily be due to overfitting. Instead, the issue could be caused by domain shift - a mismatch between the training and testing data distributions. Overfitting specifically refers to a model memorizing training data patterns rather than learning generalizable features. If performance drops significantly, the model lacks robustness to varying conditions rather than suffering from overfitting.

As given in the response to the General Comment, by employing data mixing, dropout and $L2$ regularization, the robustness can be enhanced to some extent. Other potential methods for enhancing the robustness of the equalizer are also analyzed. Details of the robustness analysis can be found in the response to the General Comment.

By mixing training data from different received power could make the performance better than trained at a specific power. Since the BER remains within the same exponential magnitude for a power variation of up to 2 dB, so 2 dB change in the power we regard the system is able to tolerate without needing recalibration. Normally, for neural networks, smaller network scale tends to exhibit lower robustness, which has been shown by numerous studies. Increasing the size of the neural network can further enhance robustness, however, it also increases power consumption. In this work, we try to employ relatively lite AI-based equalization methods to achieve a balance between performance and power consumption. The employed network scale of the AI equalizer in this work is small, only consisting of four layers in total for the DNN and three layers for the GRU, respectively, and making robustness still remains a relatively challenging task.

Comment 11: *"The reported insertion loss of the modulator is over 10 dB; and the majority of this is attributed to the modulation arms. In comparison to existing MZI-type modulators, this is relatively high*

and can be an important practical limitation. I suspect a large part of this is due to the free-carrier absorption loss, and some potentially due to mode conversion into and out of the Bragg waveguide. Can the authors quantify the different factors contributing to this loss? Are there any specific design modifications or fabrication improvements that could be used to reduce these losses?”

Our reply: Thanks for the comment. Indeed, the relatively high loss of Si-SLMs is still a recognized problem in the field that needs to be optimized further. Here, the insertion loss of the Si-SLM is measured to be 10.5 dB, of which 9.1 dB from the modulation arms. Similarly, in [Optica, 11(9), 1212-1219 (2024)], the insertion loss of the total device is approximately 10 dB and around 8 dB comes from the modulation arms.

For the loss quantification, the Q factor can be used to describe the relationship between the energy storage and loss of the resonant cavity, which can comprehensively characterize the loss of the optical resonator and be decomposed into the contribution of different loss terms. For the mid-gap mode, the Q factor is consisted by,

$$Q^{-1} = Q_{loading}^{-1} + Q_{propagating}^{-1}$$

in which $Q_{loading}$ represents the energy leakage at both ends of the slow light waveguide, while $Q_{propagating}$ counts for the energy dissipated due to scattering loss and material absorption when the light propagates through the waveguide. Here, from the simulation, we estimate $Q \sim 2100$, $Q_{propagating} \sim 5000$ and $Q_{loading} \sim 3600$ for the current design parameters ($Np20$ $Nr20$). Moreover, $Q_{loading}$ can be understood from the proportion between the total energy W stored in the waveguide and the energy flux escaping from its ends. Therefore, according to the relationship between Q factor and loss factor at around 1550 nm in silicon waveguides, the loss per unit length corresponds to $Q / Q_{propagating} / Q_{loading}$ can be calculated as around 234 dB/cm, 98 dB/cm, 136 dB/cm in theory, respectively.

At the manufacturing level, due to the limitations of actual process and materials, the practical experimental loss will be higher than theoretical value indeed, which can be improved further by optimizing the fabrication process, such as reducing the side roughness, so as to enhance the device performance further. For the device design, as a proof-of-concept design, the transition region that adiabatically connects the conventional waveguide and the slow-light waveguide has not been optimized yet. Therefore, there exists some abrupt change of group velocities between the conventional waveguide mode to the slow-light mode, thus will bring some additional reflection loss. In the subsequent optimization, a gradual taper can be specially designed for the transition between modes, thereby reducing unnecessary losses.

We hope the above analysis could make it clearer.

Comment 12: *“In Fig 3e, there is a significant difference between the BERs measured for different channels, especially at lower data rates. Does this have anything to do with the wavelength dependence of the modulator response? Or are there any other reasons behind this difference?”*

Our reply: Thanks for the comment. The specific wavelengths of 8-channels are from 1548 nm to 1555 nm, with the spacing of 1 nm, the insertion losses are relatively consistent between different channels due to the flat responses across the 7 nm for the Si-SLMs. Actually, the BERs measured for different channels are still close, such as for 130 Gbps signal, the BERs for Ch2 to Ch8 are about 1.9×10^{-3} to 4.3×10^{-3} , which are still on the same order of magnitude. However, the slight performance fluctuation

could be found, which is mainly due to the nonflattened gain spectrum of the pre-amplifier and different spectral responses of PDs at receiving end. By optimizing the link comprehensively, the slight fluctuation between different channels can be reduced further.

Response letter

We appreciate the careful review by the reviewers and have modified the manuscript in accordance with their suggestions. Here, we present a point-by-point reply (**in black**) to the reviewers' comments (**in gray**), as well as the action taken (**in red**).

Response to the report from Referee #1

Comment 1: *"In Supplementary Section 8, the bit energy consumption E_b of the modulator is discussed based on the equation in the first paragraph. However, this equation neglects the power consumption in the termination resistors. As discussed in Kawahara, et al., Optica vol. 11, no. 9, pp. 1212-121, 2024, when the electrodes in the Mach-Zehnder modulator are terminated by impedance-matched resistors, the total power consumption of the modulator is determined by that at the resistors, regardless of the amount of charge and discharge at the junction capacitance of the phase shifters. This paper says that the electrodes were terminated by resistors to avoid the RF reflection, although its value is not written. In such a case, E_b should be estimated from that at the resistors, which might be much larger than that denoted in this supplement."*

Our reply: Thanks for the reviewer's valuable comments. As for the previous " E_b " in Supplementary Section 8, we focus specifically on the energy consumption of the PN junction. To avoid potential misunderstanding, we have clarified that in the equation calculation, the term refers exclusively to the energy consumption of the PN junctions, as expressed in Equation (S4). Accordingly, we have revised the term from " E_b " to " E_{bj} ". We also explain that the system power consumption includes contributions from the PN junctions, termination resistors, and TiN heaters, as detailed in Supplementary Section 8.

We fully agree with the reviewer that the power dissipation of termination resistors should be considered for practical power estimates, as emphasized in the referenced paper [Kawahara, et al., Optica vol. 11, no. 9, pp. 1212-121, 2024]. That is a very rigorous reference, which takes terminal-induced power dissipation seriously into account, rather than focusing solely on the minimal PN junction power consumption. To make our power consumption calculation more rigorous, we have included the power dissipation induced by termination resistors in the overall system power calculation, which comprises both RF dissipation and the DC component due to bias control.

To accurately assess practical power consumption, we use actual power outputs measured from all equipment ports (including RF drivers, DC bias supplies, and TiN heaters), and combine these values with the power consumption of the laser, EDFA, and equalizer, as summarized in Table S1. For DC bias, two modulation arms were given a DC bias of 2.5 V each with the measured currents of 0.077 A and 0.080 A, resulting in a power consumption of 0.393 W. For RF dissipation, the measured operating voltage and current were 9 V and 0.531 A, thus the combined power consumption of drivers was 4.779 W. The total power of the RF driver and DC bias includes all current driving power used in our scenarios. It is also worth noting from Table S1, this driving power is minimal compared to the equalizer power, which is the more dominant factor in our case.

Additionally, for the TiN impedance-matched resistors, we have included the practical measurement results, showing an on-chip resistance of approximately 55 Ω , to provide a clearer estimation of our devices. This information has been added to Supplementary Section 1.

We hope the above analysis could make it clearer. And again, thanks for your recognition of our work. Your insightful and valuable comments have helped us a lot to improve our manuscript further.

(Supplementary Section 1) Meanwhile, on-chip termination resistors are integrated at the remote end of RF electrodes, with a measured resistance of around 55Ω , to achieve better impedance matching and further reduce microwave reflection.

(Supplementary Section 8) To evaluate the optical energy consumption level of the modulation arms of Si-SLMs (excluding systems), we have calculated the energy consumption per bit of the depletion-mode PN junction modulation (E_{bj}) specifically for both PAM-4 and PAM-8 transmission first.

(Supplementary Section 8) By reading the actual voltage and current values directly from all equipment ports, the calculated power consumption includes contributions from the PN junctions, termination resistors, and TiN heaters of the modulator. For DC bias, two modulation arms were given a DC bias of 2.5 V each with the measured currents of 0.077 A and 0.080 A, resulting in a power consumption of 0.393 W. To ensure that the modulator operates at the quadrature point, one TiN heater was driven at 2.290 V and 0.011 A, thereby the corresponding power consumption was 0.025 W. For the electrical amplifiers, two drivers were connected in parallel to the same power supply channel. The measured operating voltage and current were 9 V and 0.531 A, thus the combined power consumption of drivers was 4.779 W.

Response to the report from Referee #3

General comments: *"I would like to express my appreciation to the authors for their thoughtful and thorough responses to the reviewer comments, including those I previously raised. It is clear that they have made significant efforts in revising the manuscript, providing additional experimental data, expanding technical explanations, and clarifying key points. These efforts are sincerely acknowledged.*

The integration of an AI-based equalizer into a silicon photonic system to achieve 400 Gbps transmission is an impressive technical achievement. The direction of combining machine learning with silicon photonics is timely and has strong potential for impact in both fields.

At the same time, I still have some remaining questions, particularly regarding how deeply the manuscript explains the internal behavior of the proposed equalizer and to what extent the method could be applied more broadly across different photonic systems. While the results are clearly presented and well supported, I feel that additional discussion on the underlying mechanisms and general design principles could help further highlight the conceptual contribution of the work.

That said, I also acknowledge that some of these questions may stem from my own limited understanding of certain aspects of the system or model. If so, I appreciate the authors' patience and would welcome any additional clarification they might be able to provide.

I would be happy to see further elaboration on these points if feasible, and in the meantime, I defer to the editorial team regarding the manuscript's alignment with the scope and expectations of Nature Communications."

Our reply: Thanks for your recognition. Your insightful comments have been helpful for us in revising our manuscript. Based on the reviewers' comments, we have made improvements in our work, and we really appreciate that. To further address the questions raised, we made the corresponding response in detail as follows:

For the underlying mechanism of AI equalizers, the artificial neural network (ANN) equalizers construct a complex map with nonlinear boundaries between the input and output spaces. They exhibit potency in estimating parameters within a noisy environment and recognizing complex mappings with nonlinear boundaries between input and output data, it views the equalization as multi-class problems and provides better performances than the conventional equalizers. By training on large datasets, neural networks can better capture the underlying distribution of the signal and learn the subtle relationships between different symbols. Neural networks can learn to directly map the distorted signal back to the transmitted symbols without manually decomposing the problem into individual components. This simplifies the process and often leads to better performance because the network learns an optimal transformation of the received signal, considering all distortions jointly.

For the working architecture of DNN and GRU equalizers in this work, detailed explanation is given in line 282-294 and line 335-371 of main part, and Supplementary Section 3-5. In the Methods section of the manuscript, the mathematical derivations are also provided for both DNN and GRU cases in detail.

GRUs utilize gating mechanisms to selectively retain or discard information, allowing them to effectively remember signal context, which is an essential capability for mitigating inter-symbol interference (ISI). This architecture provides a parameter-efficient means of capturing temporal dependencies. Below are general design principles for developing a GRU-based optical equalizer:

1. Input window. Input typically includes k preceding and k succeeding symbols around the current sample. The value of k is a hyperparameter and relates to channel memory length. To retain more sequential information, larger k values are preferred.
2. Network architecture. One or more layers of GRU units can be configured, and multi-directional processing can be employed to capture both past and future contextual information within the sequence. Employing a relatively lite architecture is recommended to achieve an optimal balance between performance and power consumption.
3. Loss function. Select an appropriate loss function based on the output structure. Mean squared error (MSE) or cross-entropy loss can be used depending on the nature of the network's output. The network is trained to minimize the discrepancy between the predicted symbols and the actual transmitted symbols by optimizing a predefined loss function.
4. Training strategy. A large labeled dataset of received symbols and corresponding transmitted symbols can be used to train the equalizer. Training the network with adversarial examples or utilizing transfer learning by pretraining the model on a broader dataset can enhance its resilience to unexpected distortions.
5. Evaluation metrics. Determine the equalization performance evaluation metrics such as BERs. These metrics are crucial for assessing the effectiveness of the equalizer in mitigating signal impairments.
6. Hardware considerations. Ensure that the model is deployable under given hardware constraints, keeping in mind the computational and memory overhead during both training and inference stages.

Regarding the broader applicability of our proposed AI equalization methodology beyond the specific context of Si-SLMs, we believe that the core advantages of the employed AI equalizers lie fundamentally in their demonstrated capacity to address complex temporal dependencies such as ISI, and to effectively mitigate significant nonlinear distortions. These signal impairments are not unique to Si-SLM systems indeed but are rather ubiquitous challenges in other high-speed optical communication links, including conventional silicon modulators or electro-absorption modulators (EAMs). While system-specific optimization and validation remain crucial avenues for future investigation, the underlying mechanisms of the AI approach suggest considerable potential beyond Si-SLM systems.

We hope the above discussion could clear the reviewer's concerns. We have also added the new discussion content in Supplementary Section 5 to describe the underlying mechanisms and general design principles of the network equalizers further comprehensively. And thanks again for your recognition of our work.

(Supplementary Section 5) GRUs utilize gating mechanisms to selectively retain or discard information, allowing them to effectively remember signal context, which is an essential capability for mitigating inter-symbol interference (ISI) in high-speed PAM-4 and PAM-8 transmissions. This architecture provides a parameter-efficient means of capturing temporal dependencies. Compared to feedforward DNNs, bidirectional or multidirectional GRU models can leverage both past and future context, thereby enhancing symbol prediction accuracy. Here, the general design principles for developing a GRU-based optical equalizer are summarized. For the input window, which provides temporal information the GRU needs to model ISI, the input typically includes k preceding and k succeeding symbols around the current sample. The value of k is a hyperparameter and relates to channel memory length, and larger k values

are preferred to retain more sequential information. In the network architecture, one or more layers of GRU units can be configured, and multi-directional processing can be employed to capture both past and future contextual information within the sequence. Employing a relatively lite architecture is recommended to achieve an optimal balance between performance and power consumption. Also, it is important to select an appropriate loss function based on the output structure. Typically, mean squared error (MSE) or cross-entropy loss can be used depending on the properties of the network output. The network is trained to minimize the discrepancy between the predicted symbols and the actual transmitted symbols by optimizing a predefined loss function. For the training strategy, a large labeled dataset of received symbols and corresponding transmitted symbols can be used to train the equalizer. By introducing variations in the training dataset, the generalization of the equalizer can be enhanced. Meanwhile, determining equalization performance evaluation metrics such as BER is necessary for assessing the effectiveness of the equalizer in mitigating signal impairments, and ensure that the model is deployable under given hardware constraints while recording the computational and memory overhead during both training and inference stages.

Response to the report from Referee #4

General comments: “I appreciate the effort that the authors have put in to revise their work. They have now addressed all of my comments in detail. The new supplementary details on how the implemented equalizer compares with existing methods are clear. It is also now much clear why the implemented AI equalizer is particularly suitable for silicon slow-light modulators. The new title is more suitable for conveying the impact of the presented work. The provided quantitative details on the energy consumption, training of the equalizer, modulator loss, and driving voltage also provide better context and improve the reproducibility of the work. I now believe the work is suitable for publication at Nature Communications.”

Our reply: Thanks for the reviewer’s valuable comments and support for publication of our work. The insightful comments have been helpful for allowing us to clarify and further improve our manuscript. We appreciate the reviewer’s recognition of our work.

Response letter

We appreciate the careful review by the reviewers and have modified the manuscript in accordance with their suggestions. Here, we present a point-by-point reply (in black) to the reviewers' comments (in gray), as well as the action taken (in red).

Response to the report from Referee #1

Comment 1: *"The revised paper still discusses the power consumption and bit energy consumption values at the p-n junction of the phase shifter, but this only misleads the reader.*

The performance of the phase shifter should be evaluated only by $V_{\pi} L$, not by power consumption. An efficient phase shifter reduces V_{π} and/or shortens L . Some power is consumed by the parasitic junction resistance R_{pn} in the phase shifter, but it is negligible compared to the power consumption at the termination resistance R_L , which is given by $2 \times (V_{\pi}/2)^2/R_L$ in the case of push-pull driving. In an ideal situation, R_{pn} is almost zero, but this power consumption still remains.

Therefore, the power consumption and corresponding bit energy consumption values evaluated for the p-n junction are meaningless. Many previous papers claiming these consumption values are very misleading. This paper should stop doing that."

Our reply: Thanks for the reviewer's valuable comments. Sorry for the misleading and we initially adopted this calculation method based on some of the previous references in the area. In that view, the modulation energy consumption is regarded as the charging energy of the PN junction, while the power dissipation of the terminal resistance was attributed to the driving electrical link. We fully agree with the reviewer that the power dissipation of termination resistors should be included in the discussion of total energy consumption of the modulator. Many previous works did not pay enough attention to this point indeed. The paper last time, [Kawahara, et al., *Optica* vol. 11, no. 9, pp. 1212-121, 2024], provides a remarkable exemplification for terminal-induced power dissipation research and the calculation method listed above is the rigorous approach to conduct this calculation. For push-pull driving, the power dissipation (P_T) at the termination resistors (R_T) is given by $P_T = 2 \times \frac{(V_{pp}/2)^2}{R_T}$. Based on this, the P_T in this work is calculated to be 0.454 W, and the corresponding energy consumption per bit of the modulator (E_b) is 1.135 pJ/bit at 400 Gbps. Although the high V_{pp} value results in a power increase, the enhanced transmission speed allows the E_b of the modulator to remain at an acceptable level. Compared to the previous energy consumption of only PN junction, which is ~100 fJ/bit, the E_b including termination resistors (> 1pJ/bit) is much larger indeed, which is more accurate and more consistent with the actual situation. We have already added this updated calculation approach into Supplementary Section 8. Also, to avoid potential misunderstanding, we have deleted the original calculation Equation (S4) for PN junction and the corresponding discussion part in Supplementary Section 8 completely.

Regarding practical system power consumption, we comprehensively describe the energy consumption of each key component of the entire system, as we discussed in the previous responses. To accurately assess practical power consumption, we collect power outputs measured from all equipment ports (including RF drivers, DC bias supplies, and TiN heaters), and combine these values with the power of the laser (which can be replaced by on-chip lasers), EDFA, and equalizer, as summarized in Table S1. For the energy consumption of AI equalizers, we evaluated the power of traditional nonlinear algorithm

(VNLE) and AI equalizers (DNN, bi-GRU and T-biGRU) further in Table S2, to balance the relationship between performance and consumption. The *EPCpBs* (equalizer power consumption per bit) are compared, which demonstrate that the AI equalizers have not increased *EPCpB* (and in some cases, even reduce it) compared to VNLE. This result proves the feasibility of the proposed solutions in terms of equalizer power consumption.

We hope the above analysis could make it clearer. And again, thanks for your insightful and valuable comments that have helped us a lot to improve our manuscript further.

(Supplementary Section 8) To evaluate the power consumption of Si-SLMs (excluding systems), we have calculated the energy consumption per bit (E_b) of Si-SLMs first, which is dominated by the termination resistors^[11]. For push-pull driving, the power dissipation (P_T) at the termination resistors (R_T) is given by

$P_T = 2 \times \frac{(\frac{V_{pp}}{2})^2}{R_T}$. Based on this, the P_T is calculated to be 0.454 W, and the corresponding E_b is 1.135 pJ/bit at 400 Gbps. Although the relatively high V_{pp} value results in a power increase, the enhanced transmission speed allows the E_b of Si-SLMs to remain at an acceptable level.

Response to the report from Referee #3

General comments: *"I sincerely appreciate the authors' thoughtful and comprehensive responses to my previous comments. All of my concerns have been carefully addressed, with substantial improvements made to both the main text and supplementary materials. In particular, the enhanced explanations regarding the internal workings of the AI-based equalizer, the underlying design principles, and the potential applicability to broader photonic systems are clear and informative.*

These clarifications significantly strengthen the conceptual and technical contributions of the manuscript. I believe the work is now well-suited for publication in Nature Communications."

Our reply: We appreciate the reviewer's support for the publication of our work. Your insightful and valuable comments have been helpful for us to further improve our manuscript. Thanks again for your recognition of our work.